# Medial hypothalamic MC3R signalling regulates energy rheostasis in adult mice

Ingrid Camila Possa-Paranhos[1], Jared Butts[1,2], Emma Pyszka[1], Christina Nelson[1], Samuel Congdon[1], Dajin Cho[1,2] (iD) and Patrick Sweeney[1,2] (iD)

[1]*Department of Molecular and Integrative Physiology, University of Illinois Urbana-Champaign, Urbana, IL, USA*
[2]*University of Illinois Urbana-Champaign Neuroscience Program, Urbana, IL, USA*

Handling Editors: David Wyllie & Valentina Mosienko

The peer review history is available in the Supporting Information section of this article (https://doi.org/10.1113/JP286699#support-information-section).

**Abstract figure legend** Model depicting role of DMH MC3R signaling in energy rheostasis.

**Ingrid Possa Paranhos** obtained her BSc in Biomedical Science and MSc in Molecular and Cellular Biology at the Universidade Federal do Estado do Rio de Janeiro (Brazil). She then obtained her second master's degree and started her PhD in Molecular and Integrative Physiology at the University of Illinois Urbana-Champaign (USA). Her research is mainly focused on the neuroscience and genetic aspects that regulate food intake and body weight in mammals.

This article was first published as a preprint. Possa-Paranhos IC, Butts J, Pyszka E, Nelson C, Cho D, Sweeney P. 2024. Neuroanatomical dissection of the MC3R circuitry regulating energy rheostasis. bioRxiv. https://doi.org/10.1101/2024.04.22.590573

**Abstract** Although mammals resist both acute weight loss and weight gain, the neural circuitry mediating bi-directional defense against weight change is incompletely understood. Global constitutive deletion of the melanocortin-3-receptor (MC3R) impairs the behavioural response to both anorexic and orexigenic stimuli, with MC3R knockout mice demonstrating increased weight gain following anabolic challenges and increased weight loss following anorexic challenges (i.e. impaired energy rheostasis). However, the brain regions mediating this phenotype are not well understood. Here, we utilized MC3R floxed mice and viral injections of Cre-recombinase to selectively delete MC3R from the medial hypothalamus (MH) in adult mice. Behavioural assays were performed on these animals to test the role of MC3R in MH in the acute response to orexigenic and anorexic challenges. Complementary chemogenetic approaches were used in MC3R-Cre mice to localize and characterize the specific medial hypothalamic brain regions mediating the role of MC3R in energy homeostasis. Finally, we performed RNAscope *in situ* hybridization to map changes in the mRNA expression of MC3R, pro-opiomelanocortin and agouti-related peptide following energy rheostatic challenges, as well as to characterize the MC3R expressing cells in dorsal MH. Our results demonstrate that MC3R deletion in MH increases feeding and weight gain following high-fat diet feeding, and enhances the anorexic effects of semaglutide, in a sexually dimorphic manner. Furthermore, although the arcuate nucleus exerts an important role in MC3R-mediated effects on energy homeostasis, viral deletion in the dorsal MH also resulted in altered energy rheostasis, indicating that brain regions outside of the arcuate nucleus also contribute to the role of MC3R in energy rheostasis. Together, these results demonstrate that MC3R-mediated effects on energy rheostasis result from the loss of MC3R signalling in medial hypothalamic neurons and suggest an important role for dorsal-MH MC3R signalling in energy rheostasis.

(Received 8 April 2024; accepted after revision 11 November 2024; first published online 25 December 2024)
**Corresponding author** P. Sweeney: Department of Molecular and Integrative Physiology, University of Illinois Urbana-Champaign, Urbana, IL 61801, USA. Email: sweenp@illinois.edu

**Key points**

- Melanocortin-3-receptor (MC3R) signalling regulates energy rheostasis in adult mice.
- Medial hypothalamus regulates energy rheostasis in adult mice.
- Energy rheostatic stimuli alter mRNA levels of agouti-related peptide, pro-opiomelanocortin and MC3R.
- Dorsal-medial hypothalamus (DMH) MC3R neurons increase locomotion and energy expenditure.
- MC3R cell types in DMH are sexually dimorphic.

## Introduction

In mammals, body weight is remarkably stable in the short term (i.e. over the course of days) because humans and rodents will actively resist acute weight gain or weight loss. For example, in response to energy deprivation, changes in feeding, energy expenditure and neuroendocrine circuits occur to prevent severe weight loss (Gulick, 1995; Jen & Hansen, 1984; Keys, 1946; Leibel, 2008; Leibel et al., 1995; Ravussin et al., 2018; Sims & Horton, 1968; Woods et al., 1985). Although adaptive in preventing starvation, these changes probably contribute to the difficulty in maintaining weight loss in the context of dieting. Conversely, humans and rodents also defend against acute weight gain (Ranea-Robles et al., 2022; Ravussin et al., 2018; White et al., 2010). For example, in humans, periods of excessive weight gain, such as holiday-induced weight gain, are typically followed by weight loss and a return to prior body weight (Yanovski et al., 2000). Consistently, forced overfeeding in rodents and primates (over 1–2 weeks) via intragastric infusions is followed by drastic anorexia until body weight returns to the levels observed prior to overfeeding (Ravussin et al., 2018; White et al., 2010). Although the mechanisms preventing excessive weight loss are relatively well understood, the mechanisms preventing excessive weight gain are largely unknown. Furthermore, because acute weight gain or weight loss are typically

studied in isolation, it remains unclear how neural circuits adapt in animals to resist both orexigenic and anorexic challenges.

The leptin-melanocortin pathway is among the most well-studied circuitry regulating feeding and body weight. This pathway is essential for regulating feeding and energy expenditure and mutations in genes related to the leptin-melanocortin circuit are the most common cause of monogenic obesity in humans (Sweeney et al., 2023). The melanocortin system is composed of pro-opiomelanocortin (POMC) and agouti-related peptide (AgRP) neurons in the arcuate nucleus (ARC) of the hypothalamus, which produce the endogenous agonist [$\alpha$-melanocyte-stimulating hormone ($\alpha$-MSH)] and antagonist (AgRP) for the central melanocortin receptors (MC3R and MC4R) (Cone, 2005; Sternson & Atasoy, 2014; Sweeney et al., 2023). AgRP neurons are activated by caloric deprivation (Liu et al., 2012; Takahashi & Cone, 2005; Yang et al., 2011), resulting in the release of the melanocortin receptor antagonist/inverse agonist AgRP (in addition to the inhibitory neurotransmitter GABA and neuropeptide Y) (Krashes et al., 2013; Sweeney et al., 2023) to stimulate feeding and reduce energy expenditure. Conversely, POMC neurons are activated by caloric sufficiency (Quarta et al., 2021) and release $\alpha$-MSH, an agonist at the melanocortin-4 receptor (MC4R) and melanocortin-3 receptor (MC3R) (Sweeney et al., 2023), to suppress feeding and increase energy expenditure.

AgRP and POMC neurons mediate their effects on feeding and energy expenditure by projecting to multiple downstream brain regions (Sternson & Atasoy, 2014; Sweeney et al., 2023), where they act on secondary neurons containing MC3R and/or MC4R (in addition to non-MC3R and non-MC4R expressing neurons via GABA and/or neuropeptide Y). MC4R is considered to be the primary receptor mediating satiety downstream of POMC neurons (Krashes et al., 2016), whereas the role of MC3R is energy homeostasis is less well understood (Butler et al., 2000; Chen et al., 2000). Consistent with this interpretation, rodents and humans with mutations in *MC4R* are hyperphagic and obese (Fan et al., 1997; Farooqi et al., 2000, 2003; Huszar et al., 1997), while *MC3R* knockout (KO) mice exhibit minor late onset obesity (Butler et al., 2000; Chen et al., 2000), without noticeable hyperphagia when provided *ad libitum* access to regular chow diet. Recently, a case of homozygous loss-of function of MC3R was reported in humans, with this patient displaying obesity, increased fat mass, reduced lean mass and a delayed puberty phenotype such as those previously characterized in MC3R KO mice (Lam et al., 2021), suggesting that the function of MC3R may be conserved across species.

Emerging evidence suggests that MC3R exerts a unique role in energy homeostasis compared to the other components of the central melanocortin system (i.e. POMC, AgRP and MC4R). MC3R KO mice exhibit an impaired response to both acute weight loss and weight gain, which was termed altered energy rheostasis (Ghamari-Langroudi et al., 2018; Marks & Cone, 2003; Renquist et al., 2012). Although MC3R KO mice consume normal amounts of food on a regular chow diet, these mice are hyperphagic and gain excessive weight upon access to a high-fat diet (HFD) or surgical ovariectomy (Butler et al., 2000; Ghamari-Langroudi et al., 2018). Conversely, MC3R KO mice are hypersensitive to various anorexic challenges, including stress-related anorexia (Sweeney et al., 2021) (restraint stress and social-isolation-induced anorexia), diverse forms of pharmacological anorexia (Dahir et al., 2024; Sweeney et al., 2021) and physiological anorexia (tumour-associated anorexia) (Marks & Cone, 2003; Marks et al., 2003). Together, these findings suggest that MC3R controls 'energy rheostasis', or the magnitude of metabolic responses in both the positive and negative direction following anabolic or catabolic stimuli (Ghamari-Langroudi et al., 2018). Thus, in contrast to other mutations in the leptin-melanocortin pathway, which result in uncontrolled hyperphagia and weight gain regardless of the dietary condition, deletion of MC3R results in an exaggerated response in opposing directions to orexigenic or anorexic stimuli.

Although prior work implicates MC3R in excessive responses to both orexigenic and anorexic stimuli (energy rheostasis), this interpretation is almost entirely based on behavioural data in which the MC3R has been deleted globally from development. Constitutive deletion of MC3R results in a multitude of secondary metabolic and neuroendocrine abnormalities, such as increased fat mass and leptin levels, and impaired levels of corticosterone and thyroid hormone (Butler et al., 2000; Chen et al., 2000; Ghamari-Langroudi et al., 2018; Renquist et al., 2012), which confound the interpretation of behavioural phenotypes in MC3R KO mice because these changes may alter behavioural phenotypes independent of MC3R action. Thus, it remains unclear whether central MC3R signalling regulates energy rheostasis or whether the previously reported energy rheostasis phenotype is a result of secondary abnormalities observed in constitutive MC3R KO mice. Furthermore, prior studies indicate that global gene deletions can result in developmental compensation, such that a limited behavioural phenotype is observed following constitutive deletion. For example, despite well-established and critical roles in energy homeostasis, constitutive deletion of important metabolic receptors and signals such as glucagon-like peptide-1 receptor (GLP1R) (Ayala et al., 2010), ghrelin (Sun et al., 2003) and ghrelin receptor (Sun et al., 2008) result in limited metabolic phenotypes. Therefore, the relative importance of central MC3R signalling in adult animals cannot be ascertained solely from global KO mouse models.

Recent work demonstrates that MC3R is widely expressed throughout the brain, with dense MC3R expression observed throughout most brain regions (Bedenbaugh et al., 2022). Of interest to the role of MC3R in energy rheostasis, particularly strong MC3R expression is observed in medial hypothalamic structures [ARC, ventral-medial hypothalamus (VMH) and dorsal-medial hypothalamus (DMH)] with well-established roles in energy homeostasis (Bedenbaugh et al., 2022). However, given the dense brain-wide expression of MC3R, the specific brain regions and neural circuits mediating the role of MC3R in energy rheostasis are largely unknown. In the present study, we utilized a viral deletion approach to specifically delete the MC3R within select medial hypothalamus (MH) regions in adult mice. Our findings indicate that adult-specific deletion of MC3R throughout the MH recapitulates many of the energy rheostasis phenotype observed in MC3R KO mice. Unexpectedly, despite a critical role for MC3R in the ARC in feeding (Dahir et al., 2024; Ghamari-Langroudi et al., 2018; Renquist et al., 2012; Sweeney et al., 2021), we find that the ability of MC3R to control energy rheostasis does not require the ARC. Furthermore, we identify the DMH as an important site for MC3R-mediated changes in energy rheostasis and characterize a specific role for DMH MC3R neurons in controlling energy expenditure and locomotion.

## Methods

### Animals

All animal experiments were approved by the University of Illinois Institutional Animal Care and Use (IACUC) committee. The experiments were performed on littermate mice (6–16 weeks old) that were approximately matched for age and sex between experimental and control groups. MC3R-floxed mice were generated previously by Cyagen using CRISPR-Cas9 gene editing approaches as described in previous studies (Cho et al., 2023; Ghamari-Langroudi et al., 2018; Sweeney et al., 2021). Heterozygous MC3R-floxed mice were bred together to generate littermate homozygous flox/flox and wild-type (WT) mice that were used for experiments. MC3R-Cre mice were previously described (Ghamari-Langroudi et al., 2018). Tdtomato mice were bred as homozygous mice and crossed to MC3R-Cre heterozygous mice to generate MC3R-Cre/tdtomato transgenic lines (Bedenbaugh et al., 2022; Cho et al., 2023). All mice were initially caged in groups of two to five mice per cage of the same sex until food intake experiments started, when they were single caged for at least 3 days prior to starting experiments. Mice were housed under a 12:12 h light/dark photocycle (lights on 07.00 h) in temperature- (20–21°C) and humidity-controlled cages.

All mice had *ad libitum* access to regular chow food and water, unless otherwise noted.

### Viral vectors

For MC3R deletion studies Cre-expressing adeno-associated viral vectors (AAV2-hsyn-mcherry-cre; (UNC GTC Vector #AV6445B) were injected in both WT and MC3R floxed homozygous littermate mice. For chemogenetic activation of DMH and VMH MC3R neurons, the control group (MC3R-Cre positive mice) was unilaterally injected with AAV5-hsyn-DIO-mcherry (Addgene, Watertown, MA, USA; #50459). Experimental mice were injected with AAV5-hsyn-DIO-hM3D(Gq)-mcherry (Addgene; #44361).

### Stereotaxic viral injections

Mice were anaesthetized in an isoflurane chamber and placed in a stereotaxic frame (Kopf, Tujunga, CA, USA) with constant flow of isoflurane. Viral injection co-ordinates for the MC3R-flox MH-KO injections were: anterior/posterior (A/P), −1.35 and −1.8 mm (from bregma); medial/lateral (M/L), −0.40 and +0.40 mm; dorsal/ventral (D/V), −5.65, −5.75 and −5.80 mm (from surface of the brain). In each A/P co-ordinate, three injections of viral vectors were delivered in three D/V co-ordinates, and 75, 100 and 75 nL of virus was injected, respectively, in each D/V site at a rate of 50 nL min$^{-1}$. Viral injection co-ordinates for the MC3R-flox dMH-KO and the MC3R-cre chemogenetics experiments were: A/P, −1.7 mm (from the bregma); M/L, −0.30 mm; D/V, −5.10 mm (from surface of the brain). The viral vectors were delivered at a rate of 10 nL min$^{-1}$, with a total of 60 nL per side. After the injection, the glass pipette was left in the more dorsal co-ordinate for an additional 5 min to prevent leakage of virus from the targeted brain region. Mice were administered 5 mg kg$^{-1}$ carprofen subcutaneously after surgery, had their body weights measured and were returned to their home cages. For all experiments, there was a 2 week period post-surgery before starting experiments to allow time for viral expression and recovery from surgery.

### Indirect calorimetry and locomotion measurements

For chemogenetic experiments and one cohort of dMH MC3R floxed experiments, to measure the energy expenditure and x-ambulatory (locomotion), the male mice were placed into Comprehensive Laboratory Animal Monitoring System (CLAMS; Columbus Instruments, Columbus, OH, USA) cages. Mice were habituated to the CLAMS cages for 24 h before beginning behavioural measurements. Food intake, locomotion and energy expenditure were automatically calculated every 15 min

by the CLAMS system. Energy expenditure was calculated as the average per hour and the *x*-ambulatory as the sum of the values each hour.

## Chemogenetic experiments

All the mice used in this assay were MC3R-cre positive and the stereotaxic viral injections were of either cre-dependent mcherry (control group) or cre-dependent hM3Dq-Cherry viral vectors (experimental group). These experiments were performed in the CLAMS system (Columbus Instruments) as described above. On the first day after habituation to the CLAMS, half of the mice were administered saline (200 µL, I.P.) and the other half received 1 mg kg$^{-1}$ CNO (Enzo, Farmingdale, NY, USA; catalog. no. BML-NS 105-0025, dissolved in 200 µL of saline, I.P.) 1 h before the start of the dark cycle (18.00 h). On the following day the groups were swapped so that all mice received both saline and CNO. The measurements were analysed in the first hour of the dark cycle (i.e. 2 h after the saline or CNO administration).

Validation of the technique was performed by administrating half of the control and experimental groups with saline (0.9% NaCL, 200 µL, I.P.) and the other half with CNO (1 mg kg$^{-1}$, 200 µL, i.p). Perfusion was performed 90 min after injections and sectioning and viral location was performed as described in the sections below 'Perfusion, Sectioning and viral location assessment' and 'Immunohistochemistry and cfos quantification'.

## Perfusion, sectioning and viral location assessment

Following experiments all mice were perfused with 10% formalin followed by dissection of the brain. The brains were transferred to a 10% formalin solution for 24 h, followed by 24 h in 10%, 20% and 30% sucrose solutions in 1× phosphate-buffered saline (PBS). Brain slices containing the hypothalamus were then obtained by sectioning on a cryostat (Leica, Wetzlar, Germany; CM3050S) at 40 µm thickness. These sections were then placed into 24-well plates containing 500 µL of 1× ultra-pure PBS, mounted on Superfrost glass slides (Thermo Fisher Scientific, Waltham, MA, USA), and imaged with confocal microscopy (Zeiss, Oberkochen, Germany; LSM700 microscope, Z-stack and tile scan of whole hypothalamus). For RNAscope experiments, sections were mounted directly onto glass slides (20 µm) and the RNAscope experiment was performed as described in the section below on 'RNAscope *in situ* hybridization and mRNA quantification'.

## Immunohistochemistry and cfos quantification

The 40 µm brain sections were placed into 24-well plates containing 500 µL of blocking buffer (100 mL of Ultrapure 1× PBS, 2 g of bovine serum albumin and 100 µL of Tween 20) and placed on a shaker at room temperature and allowed to sit for 2 h. Next, a master mix of the rabbit cFos (dilution 1:1000; 9F6 Rabbit mAb; Cell Signaling Technology, Danvers, MA, USA), rabbit AgRP (dilution 1:1000; catalog. no. H003-57; Phoenix Pharmaceuticals, Mannheim, Germany) or rabbit POMC (dilution 1:1000; catalog. no. H029-30; Phoenix Pharmaceuticals) primary antibody in blocking buffer was prepared. Then, 500 µL of this master mix was added to each well-containing brain sections and placed onto a shaker at 4°C overnight. The primary antibody mixture was replaced by 500 µL of ultra-pure 1× PBS and placed on a shaker at room temperature for 10 min and this step was repeated twice more. The secondary antibodies [goat anti-rabbit IgG (H+L) cross-adsorbed secondary antibody, Alexa Fluor 488; Thermo Fisher Scientific] were prepared in blocking buffer at a concentration of 1:500 and then 500 µL of the secondary was added to each well and placed on a shaker, at room temperature, and incubated for 2 h. Following three 10 min wash steps with 1× ultra-pure PBS, the sections were mounted on Superfrost glass slides (Thermo Fisher Scientific) and analysis of the images were performed using a LSM700 or LSM900 confocal microscope (Z-stack, 20×).

For the chemogenetic experiments, two sections with mcherry expression in DMH for each mouse were used to count the mcherry and cfos signals, and the average of those measurements was calculated for each mouse and used in the statistical analysis. The cfos quantification was performed using ImageJ/Fiji software (https://fiji.sc) and the total number of DMH cells expressing MC3R (labelled with mcherry viral vectors) was counted for the entire section and the percentage of these cells co-expressing with cfos was quantified for the statistical analysis.

## RNAscope *in situ* hybridization and mRNA quantification

RNAscope analysis of viral KO efficiency was performed on WT and MC3R-flox mice, all injected with AVV2-hsyn-mcherry-cre. The animals were perfused as described above, but using 4% paraformaldehyde instead of 10% formalin according to ACD RNAscope recommended protocols, and 20 µm sections were directly mounted onto Superfrost glass slides for RNAscope *in situ* hybridization. RNAscope multiplex fluorescent *in situ* hybridization, version 2, was used according to the protocol described in the kit. MC3R mRNA expression was visualized using the probe Mm-Mc3r-C2 probe (Ref: 412541-C2). Images were obtained via confocal microscopy (Z-stack and tile scan of whole hypothalamus). The mRNA count (Fig. 1*C*–*E* and *G*–*I*) was performed using Fiji/ImageJ software, considering the soma [stained with

4′,6-diamidino-2-phenylindole (DAPI)] containing at least one transcript of each probe as a positive cell in the entirely of the MH.

For MC4R, LEPR, VGAT, VGlut2, AgRP and POMC, the probes utilized were: Mm-Mc4r-C3 (Ref: 319181-C3), Mm-Lepr-C3 (Ref: 402731-C3), Mm-Slc32a1 (Ref: 319191-C1), Mm-Slc17a6 (Ref: 319171-C1), Mm-Agrp-C2 (Ref: 400711-C2) and Mm-Pomc-C3 (Ref: 314081-C3), respectively. These experiments were performed on C57/BL6J WT mice

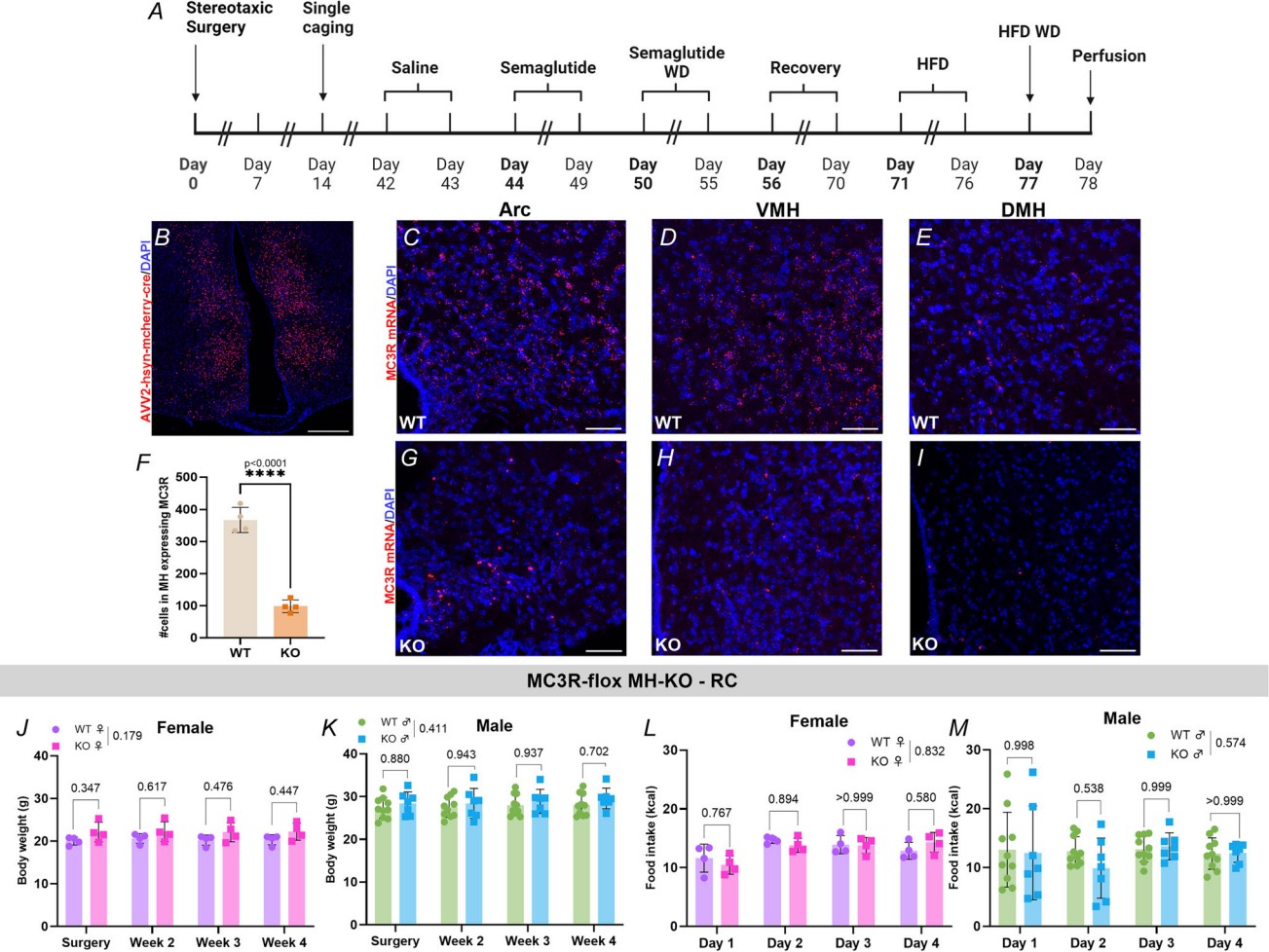

**Figure 1. Adult hypothalamic deletion of MC3R does not alter food intake or body weight in regular chow *ad libitum* fed conditions**

*A*, timeline of experimental protocol for experiments in Fig. 1. Created with Biorender. *B*, viral location example of a MC3R-flox mouse with AVV2-hsyn-mcherry-cre in medial hypothalamus (MH). *C–E*, RNAscope images of example WT mouse injected with AVV2-hsyn-mcherry-cre and with MC3R mRNA in red in the ARC (*C*), ventromedial hypothalamus (VMH; *D*) and dorsomedial hypothalamus (DMH; *E*). *F*, number of cells expressing MC3R mRNA in the medial hypothalamus in WT and MC3R MH-KO mice (unpaired *t* test, $F_{3,3} = 3.957$ $n = 4$ mice per group, three sections per mouse). *G–I*, RNAscope images of an example KO mouse injected with AVV2-hsyn-mcherry-cre and with MC3R mRNA in red in the ARC (*G*), VMH (*H*) and DMH (*I*). *J* and *K*, WT and MC3R-MH KO body weight from surgery until 4 weeks post-surgery in female mice (*J*) and male mice (*K*). (*J*, two-way ANOVA and Šídák's test *post hoc* analysis repeated measures, main effect of genotype $F_{1,6} = 1.390$, time $F_{1.264,7.582} = 2.243$ and interaction $F_{3,18} = 1.214$, $n = 4$ mice per group; *K*, two-way ANOVA and Šídák's test *post hoc* analysis repeated measures, main effect of genotype $F_{1,15} = 0.013$, time $F_{1.818,27.26} = 6.831$ and interaction $F_{3,45} = 0.8371$, $n = 10$ mice for WT group and 7 mice for MH MC3R floxed group). *L* and *M*, food intake for 4 days, 2 weeks after surgery in female mice (*L*) and male mice (*M*) (*L*, two-way ANOVA and Šídák's test *post hoc* analysis repeated measures, main effect of genotype $F_{1,6} = 0.049$, time $F_{3,18} = 10.85$ and interaction $F_{3,18} = 1.755$, $n = 4$ mice per group; *M*, two-way ANOVA and Šídák's test *post hoc* analysis repeated measures, main effect of genotype $F_{1,15} = 0.013$, time $F_{3,15} = 0.7351$ and interaction $F_{3,45} = 0.5873$, $n = 10$ mice for WT group and 7 mice for MH MC3R floxed group). Data points represent individual mice. Error bars represent the SD. Scale bar = 300 μm in (*B*), 50 μm in (*C*) to (*E*) and 50 μm in (*G*) to (*I*). [Colour figure can be viewed at wileyonlinelibrary.com]

following exposure to the energy rheostatic challenges described in the Results section (i.e. exposed to HFD for 5 days, 5 days of semaglutide treatment, or *ad libitum* regular chow fed). Following behavioural perturbations, mice were perfused and tissue sections containing the MH were obtained as previously described. Following RNAscope protocols, confocal images were taken using LSM700 or LS 900 microscope and Zen software (Zeiss) (Z-stack and 20× zoom) and the cells were counted in the entire DMH, VMH or ARC regions (Figs 4, 5, 14 and 15) using Fiji/ImageJ software. We analysed three sections for each combination of probes for each mouse, and the average of those sections per mouse was used to establish the number of cells which expressed the mRNA for statistical analysis. The colocalization cell count was performed considering the soma (stained with DAPI) containing at least one transcript of each probe as a positive cell. For the mRNA pixels measurement (i.e. MC3R expression), using Fiji/Image J, a region of interest was created covering entirely DMH, VMH or ARC sections in duplicates or triplicates for each animal, and the signal was measured using the 'measure' software feature. The average of the signal value for each animal was used to determine its signal intensity and used for statistical analysis.

### Feeding behaviour assays

**Body weight and food intake measurements.**  Two weeks after the stereotaxic surgery, which is the time given for the mice to recover from surgery and for viral expression to occur, we started daily body weight and food intake measurements. For the body weight measurements, we single-housed the individual mice and the measurements were performed at the same hour every day. At the same time, a previously measured amount of food was given to the mice and the mice were placed in a new clean cage, aiming to avoid errors with crumbles of food on the bottom of the cage that could interfere with the food intake values. Food intake was measured by weighing the food (either regular chow or HFD, depending on the experiment) in the hopper and subtracting from the food weight given on the previous day.

**Caloric restriction studies.**  We measured the *ad libitum* food intake and body weight for the mice as described in above in the section on 'Body weight and food intake measurements' prior to the caloric restriction. The average food intake was used as a baseline (4.3 g) and the animals were given, for 8 consecutive days, 70% of the amount of regular chow that was normally consumed in 24 h (3 g) and their body weight was measured daily. After the caloric restriction period, the animals received *ad libitum*

access to food and their food intake and body weight were measured as described above.

**Diet-induced obesity (DIO).**  WT and MC3R-floxed mice had access to *ad libitum* HFD (Research Diet Inc., New Brunswick, NJ, USA; D12492) for 7 weeks prior to surgery to induce obesity. After the DIO period, we performed stereotaxic surgical injections as described in the section above on 'Stereotaxic viral injections' to delete the MC3R in dMH. The mice were continuously provided with a HFD throughout the entire experiment and food intake was measured as described above.

**Semaglutide administration.**  We administered the WT and MC3R-KO mice with saline solution s.c ($200 \mu L$) for 2 days to acclimate animals to being handled and to the subcutaneous injections. On the days they were given saline, their body weight and food intake were measured as described in the section above on 'Body weight and food intake measurements'. Semaglutide (Peptide Sciences, Henderson, NV, USA; CAS# 2023788-19-2) was administered at $100 \mu g kg^{-1}$ daily to the mice s.c. for 5 days and body weight and food intake was measured daily, as described above.

**Statistical analysis.** The animals were regrouped according to the viral location. For the MC3R-flox MH-KO animals, a bilateral viral expression in DMH, VMH and ARC was necessary for the mice to be included in the data analysis. For the MC3R-flox dMH KO mice, only mice with bilateral viral expression in the DMH were used. However, it was observed that unilateral expression of the virus was sufficient to show the same phenotype on the MC3R-flox dMH KO chronic HFD and indirect calorimetry and locomotion cohorts, which led to the inclusion of those animal in the data analysis of those experiments. For the chemogenetic assays, the mice had unilateral expression and were regrouped after viral location. All the specific viral spread of each animal is further outlined in Figs 3, 8, 10, 12 and 13. Specific statistical tests are further outlined where appropriate; when using two-way ANOVA, we utilized Šídák's test as *post hoc* analysis to compare the control and experimental columns between time-points. Normally distributed data, with equal variance between the groups, were analysed with parametric tests, whereas not normally distributed data or data with unequal variance between the groups were analysed with non-parametric tests. Data were analysed using Prism, version 10 (GraphPad Software Inc., San Diego, CA, USA) and are shown as the mean $\pm$ SD.

## Results

### Deletion of MC3R in the MH does not alter regular chow feeding

Although MC3R expression is observed throughout the brain, particularly strong expression is observed in MH regions (including the ARC, VMH and DMH) (Bedenbaugh et al., 2022). Given the established role of MH regions in energy homeostasis, we first aimed to determine the role of MH MC3R signalling in energy homeostasis in adult mice. To selectively delete MC3R in the MH of male and female mice, we injected adeno-associated virus (AAV) expressing Cre recombinase into the MH in mice containing loxp sites flanking the entire exon encoding MC3R (MC3R floxed mice) (Figs 1*A*, *B* and 2). To control for any potential off-target effects of Cre expression, WT littermate mice were also injected with the identical Cre expressing virus. First, to validate successful MC3R deletion, we performed

RNAscope *in situ* hybridization analysis to quantify MC3R mRNA expression in the MH in MC3R floxed and WT mice injected with AAV-Cre virus. Consistent with prior reports (Bedenbaugh et al., 2022), we identified dense MC3R mRNA expression throughout the MH in WT mice injected with AAV-Cre (Fig. 1*C–E*). By contrast, AAV-Cre injections in MC3R floxed mice markedly reduced MC3R mRNA expression throughout the MH, indicating successful viral-mediated deletion of MC3R (Fig. 1*F–I*).

Given the previous literature implicating MC3R in energy rheostasis (Ghamari-Langroudi et al., 2018), we designed an experimental protocol to assess the role of MC3R signalling in MH in mediating acute responses to anorexic and orexigenic stimuli in adult mice. First, we measured feeding behaviour and body weight change in MH MC3R KO and WT mice provided *ad libitum* access to a regular chow diet (Fig. 1*A*). Deletion of MC3R in the MH did not alter body weight in male or female mice

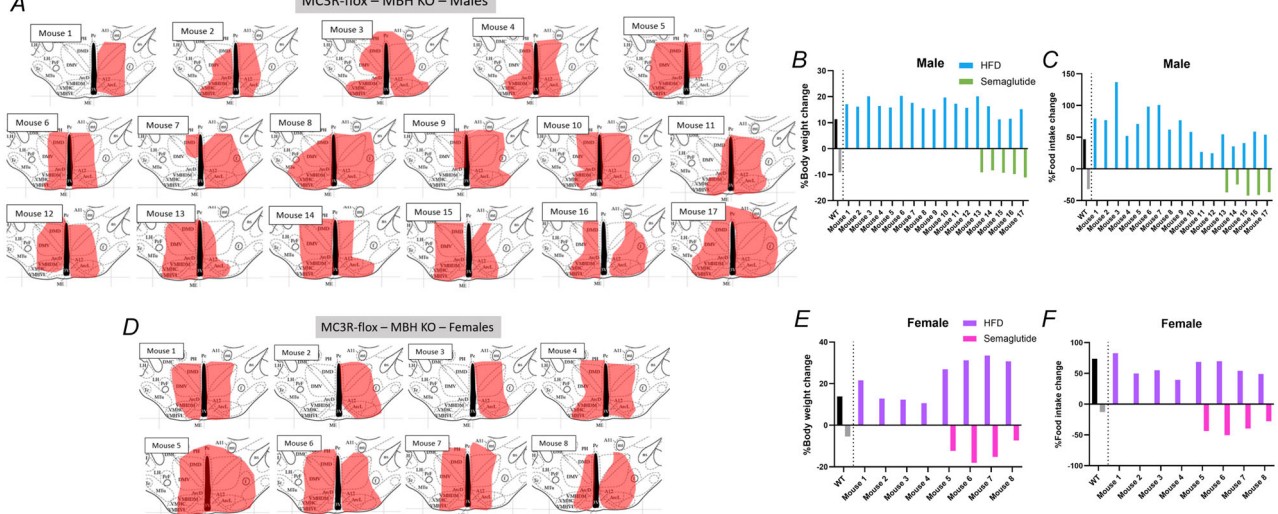

**Figure 2. Viral expression of MH injections in MC3R floxed mice**
*A*, schematics of the viral expression of all the male MC3R-flox MH KO mice used in the experiments in Figs 1 and 2. Mouse 1 to Mouse 10 were included in the chronic HFD experiments shown in Fig. 2*I–L* Adapted from Allen Brain Atlas – Mouse Brain, using the reference of bregma −1.94 mm. Mice without semaglutide results are from a cohort where only HFD experiment was performed. *B*, individual percentage difference of body weight of male mice in the last day of HFD compared to the last day of regular chow, and on the last day of semaglutide treatment compared to last day of saline injection (prior to semaglutide treatment). The WT is the average of the percentage body weight difference for all the animals in that group. *C*, individual percentage difference of the average daily food intake of male mice during HFD administration compared to the last day of regular chow, and average daily food intake during semaglutide treatment compared to last day of saline injection. The WT is the average of the percentage food intake difference for all the animals in that group. *D*, schematics of the viral expression of all the female MC3R-flox MH KO mice used in the experiments in Figs 1 and 2. Mouse 1 to Mouse 4 were included in the chronic HFD experiments shown in Fig. 2*I* and *K*. Adapted from Allen Brain Atlas – Mouse Brain, using the reference of bregma −1.94 mm. *E*, individual percentage difference of body weight of female mice in the last day of HFD compared to the last day of regular chow, and after semaglutide treatment compared to the last day of saline injection. The WT is the average of the percentage body weight difference for all the animals in that group. *F*, individual percentage difference of the average daily food intake of female mice during HFD compared to the last day of regular chow, and average daily food intake during semaglutide treatment compared to last day of saline injection. The WT is the average of the percentage food intake difference for all the animals in that group. [Colour figure can be viewed at wileyonlinelibrary.com]

(Fig. 1*J* and *K*) and had no effect on food intake in mice provided *ad libitum* access to a regular chow diet (Fig. 1*L* and *M*). Thus, similar to global developmental deletion of MC3R, adult medial hypothalamic deletion of MC3R does not affect feeding or body weight in basal conditions (Ghamari-Langroudi et al., 2018).

## MH deletion of MC3R increases anorexic response to semaglutide

Despite normal *ad libitum* chow intake, MC3R KO mice are hypersensitive to a remarkably diverse array of anorexic stimuli including stress-related anorexia (restraint stress and social isolation) (Sweeney et al., 2021), diverse forms of pharmacological anorexia (Dahir et al., 2024; Ghamari-Langroudi et al., 2018; Sweeney et al., 2021) (i.e. administration of GLP1R agonists, leptin, MC4R agonists, peptide YY and cholecystokinin) and physiological anorexia (i.e. tumour cachexia) (Marks & Cone, 2003; Marks et al., 2003). Furthermore, recent reports demonstrate that deletion or inhibition of MC3R markedly enhances the anorexic effect of semaglutide, without altering the incretin or aversive effects associated with GLP1R stimulation (Dahir et al., 2024). Therefore, we next tested whether medial hypothalamic MC3Rs mediate the enhanced anorexia associated with MC3R inhibition. Following daily habituation to saline injections, we administered semaglutide (0.1 mg kg$^{-1}$, s.c.) daily for 5 days to male and female WT and MH MC3R KO mice. Female MC3R MH-KO mice lost significantly more weight than WT littermates (Fig. 3*A*) and ate significantly less during semaglutide treatment (Fig. 3*C*). Following semaglutide treatment, female MC3R MH-KO mice maintained a significantly lower body weight for 3 days after the withdrawal, although food intake levels were not significantly different between WT and MH MC3R KO mice following the cessation of semaglutide treatment (Fig. 3*A* and *C*). By contrast, no differences in body weight or feeding were observed between male WT mice and male MH MC3R KO mice during semaglutide administration or following the cessation of semaglutide treatment (Fig. 3*B* and *D*).

## MH deletion of MC3R increases body weight gain on a HFD

In addition to enhanced responsivity to anorexic stimuli, prior studies indicate that MC3R KO mice gain more weight on a HFD (Ghamari-Langroudi et al., 2018). Therefore, following washout and recovery from semaglutide treatment (Fig. 1*A*), we next tested whether MH KO of MC3R increases feeding and body weight

when mice are provided acute access to a palatable HFD. During *ad libitum* access to HFD, female MH MC3R KO mice consumed a more HFD than littermate control animals, although no difference in body weight was observed between female WT and MH MC3R KO mice (Fig. 3*E* and *G*). Male MH MC3R KO mice gained significantly more weight starting on the third day of access to HFD until the end of the experiment (Fig. 3*F*), which was accompanied by increased HFD food intake (Fig. 3*H*). Thus, MH deletion of MC3R increases body weight gain in male mice and food intake in both male and female mice fed a HFD, recapitulating the phenotype previously observed in male mice with global deletion of MC3R (Ghamari-Langroudi et al., 2018).

Recent reports demonstrate that acute access to HFD rapidly alters the activity of hypothalamic AgRP neurons, resulting in a de-valuation of the regular chow diet, such that acute anorexia develops when animals are switched back from an HFD to a regular chow diet (Beutler et al., 2020; Mazzone et al., 2020). Because AgRP neurons produce the endogenous antagonist for the MC3R (AgRP), and prior work indicates that MC3R KO mice are hypersensitive to anorexic stimuli, we next measured the feeding and weight loss response in WT and MH MC3R KO mice after switching the diet back to a regular chow diet. Both female WT and MH MC3R KO mice exhibited a similar anorexic response upon switching to a regular chow diet, losing a similar amount of weight following the switch to regular chow (Fig. 3*H*) and equivalently reducing the kilocalories ingested after the transition to regular chow diet (Fig. 3*GF*). However, male MH MC3R KO mice lost less body weight (Fig. 3) and had a reduced anorexic response compared to WT mice following the transition to a regular chow diet (Fig. 3*H*).

Although the primary focus of this study was to characterize the role of MC3R in regulating the acute response to anorexic and orexigenic challenges, such as acute HFD exposure, in new cohorts of mice, we next tested whether deletion of MC3R regulates weight gain and feeding following a more prolonged period of HFD feeding. Following 30 days of HFD feeding, female MH MC3R KO mice did not show any significant differences in food intake or body weight main effects, although a trend towards increased feeding was observed in female MH MC3R KO mice (Fig. 3*I* and *K*). By contrast, male MH MC3R KO mice consumed significantly more food than WT littermate mice and gained more weight than WT littermate mice following 1 month of HFD exposure (Fig. 3*J* and *L*). Thus, deletion of MC3R in the MH increases feeding and body weight gain following more prolonged exposure to a HFD in males, suggesting that deletion of MC3R may increase susceptibility to diet-induced obesity in a sexually dimorphic manner.

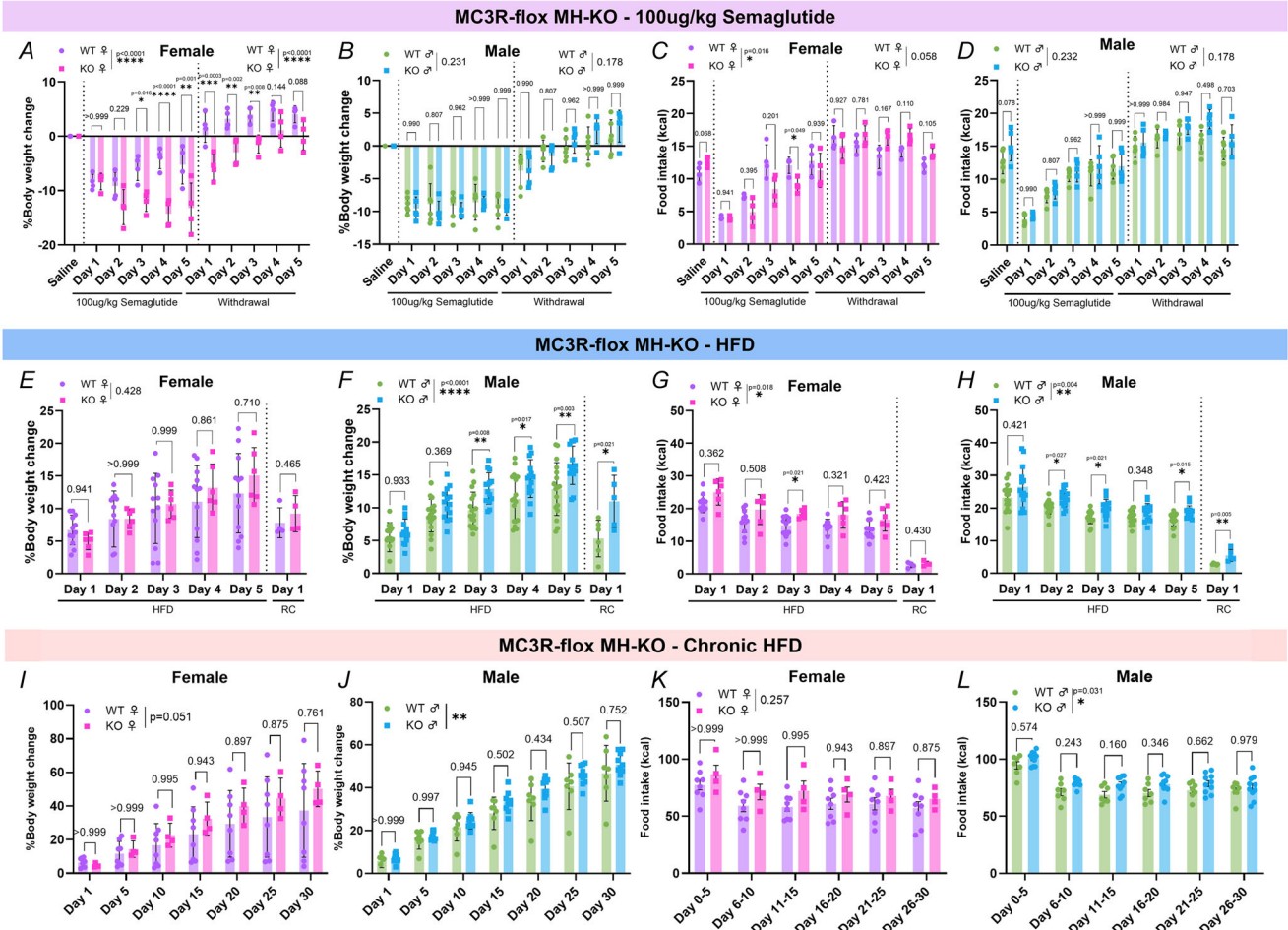

**Figure 3. Adult hypothalamic deletion of MC3R alters energy rheostasis**

*A* and *B*, daily percentage body weight change compared to baseline saline injections during daily 100 μg kg$^{-1}$ injections of semaglutide in female mice (*A*) and male mice (*B*) (*A*, baseline-corrected followed by two-way repeated measures ANOVA and Šídák's test *post hoc* analysis for semaglutide treatment period, main effect of genotype $F_{1,6}$ = 42.83, time $F_{2.624,15.74}$ = 20.17 and interaction $F_{10,60}$ = 5.759; and withdrawal period, main effect of genotype $F_{1,6}$ = 56.38, time $F_{2.286,1372}$ = 6.009 and interaction $F_{5,30}$ = 3.196, *n* = 4 mice per group; *B*, baseline-corrected repeated measures two-way ANOVA and Šídák's test *post hoc* analysis for semaglutide treatment period, main effect of genotype $F_{1,9}$ = 1.468, time $F_{2.862,25.76}$ = 48.74 and interaction $F_{5,45}$ = 0.2074 and withdrawal period, main effect of genotype $F_{1,9}$ = 5.854, time $F_{3.007,27.06}$ = 0.6336 and interaction $F_{5,45}$ = 0.0461, *n* = 10 mice in WT group and 7 mice in MH KO group). *C* and *D*, 24-h daily food intake during semaglutide administration days and withdrawal in female mice (*C*) and male mice (*D*) (*C*, two-way ANOVA and Šídák's test *post hoc* analysis repeated measures, main effect of genotype $F_{1,6}$ = 7.340, time $F_{1.781,10.69}$ = 38.34 and interaction $F_{5,30}$ = 3.904 for semaglutide period and withdrawal period, main effect of genotype $F_{1,6}$ = 5.462, time $F_{3.196,19.18}$ = 7.418 and interaction $F_{5,30}$ = 3.438, *n* = 4 mice per group; *D*, two-way ANOVA and Šídák's test *post hoc* analysis repeated measures, main effect for semaglutide period of genotype $F_{1,9}$ = 1.320, time $F_{2.652,23.87}$ = 105.5 and interaction $F_{5,45}$ = 1.491, and withdrawal period of genotype $F_{1,9}$ = 3.886, time $F_{2.191,19.72}$ = 8.042 and interaction $F_{4,36}$ = 2.828, *n* = 10 mice for WT group and 7 mice for MH KO group). *E* and *F*, percentage body weight change, comparing WT to MC3R-flox MH-KO mice with *ad libitum* access to high-fat diet (HFD) and following the transition to regular chow (RC) in female mice (*E*) and male mice (*F*) [*E*, two-way ANOVA and Šídák's test *post hoc* analysis repeated measures, main effect of genotype $F_{1,16}$ = 0.6335, time $F_{1.63,26.13}$ = 7.988 and interaction $F_{4,64}$ = 0.6512 for HFD period and unpaired *t* test $F_{3,3}$ = 1.437 for RC period; *n* = 12 mice for WT and 8 mice for MH MC3R KO group during HFD, *n* = 4 mice per group during RC period; *F*, two-way ANOVA and Šídák's test *post hoc* analysis repeated measures, main effect of genotype $F_{1,29}$ = 30.02, time $F_{1.761,51.07}$ = 41.86 and interaction $F_{4,116}$ = 1.313 for HFD and unpaired *t* test $F_{4,5}$ = 2.005 for regular chow period, *n* = 17 mice for WT group and 14 for MH MC3R KO group during HFD, *n* = 6 mice for WT group and 5 for MH MC3R KO group during RC period. Some of the mice in HFD period in (*E*) and (*F*) were continued on HFD for a longer period (see *I* and *J*) and were thus not switched to RC]. *G* and *H*, HFD and change to RC food intake in WT and MC3R-flox MH-KO mice in female mice (*G*) and male mice (*H*) (*G*, two-way ANOVA and Šídák's test *post hoc* analysis repeated measures, main effect of genotype $F_{1,16}$ = 6.931, time $F_{4,64}$ = 42.09 and interaction

$F_{4,64} = 0.0837$ for HFD period and unpaired *t* test $F_{3,3} = 1.266$ for regular chow period, $n = 12$ mice for WT and 8 for MH MC3R KO group during HFD, $n = 4$ mice per group during RC period; *H*, two-way ANOVA and Šídák's test *post hoc* analysis repeated measures, main effect of genotype $F_{1,29} = 9.947$, time $F_{1.878,54.45} = 51.69$ and interaction $F_{1,116} = 0.5463$ for HFD period and unpaired *t* test $F_{4,59} = 65.94$ for regular chow period, $n = 17$ mice for WT group and 14 for MH MC3R KO group during HFD, $n = 6$ mice for WT group and 5 mice for MH MC3R KO group during RC period). *I* and *J*, percentage body weight change, comparing WT to MC3R-flox MH-KO with chronic *ad libitum* access to HFD in female mice (*I*) and male mice (*J*) (*I*, two-way ANOVA and Šídák's test *post hoc* analysis repeated measures, main effect of genotype $F_{1,10} = 3.937$, time $F_{1.094,10.94} = 8.600$ and interaction $F_{6,60} = 0.3007$ $n = 8$ mice for WT group and 4 mice for MH MC3R KO group; *J*, two-way ANOVA and Šídák's test *post hoc* analysis repeated measures, main effect of genotype $F_{1,15} = 9.935$, time $F_{1.871,28.06} = 96.14$ and interaction $F_{6,90} = 0.3431$, $n = 7$ mice for WT group and 10 for MH MC3R KO group). *K* and *L*, food intake measurements of WT and MC3R-flox MH-KO mice during chronic *ad libitum* access to HFD in female mice (*K*) and male mice (*L*) (*K*, two-way ANOVA and Šídák's test *post hoc* analysis, repeated measures, main effect of genotype $F_{1,10} = 1.447$, time $F_{3.030,30.30} = 34.50$ and interaction $F_{5,50} = 1.137$ $n = 8$ mice for WT group and 4 mice for MH MC3R KO group; *L*, two-way ANOVA and Šídák's test *post hoc* analysis repeated measures, main effect of genotype $F_{1,15} = 5.659$, time $F_{3.792,56.88} = 48.38$ and interaction $F_{5,75} = 0.5900$, $n = 7$ mice for WT group and 10 mice for MH MC3R KO group). Data points represent individual mice. Error bars represent the SD. [Colour figure can be viewed at wileyonlinelibrary.com]

## Energy rheostatic challenges alter mRNA expression of melanocortin circuitry in a sexually dimorphic manner

MC3R signalling is bi-directionally regulated by the endogenous melanocortin receptor agonist $\alpha$-MSH (produced by the POMC peptide in POMC neurons), which stimulates MC3R, and the endogenous MC3R antagonist AgRP, which inhibits the MC3R. MC3R expression is sexually dimorphic in mice, with multiple hypothalamic regions exhibiting differing levels of MC3R expression between male and female animals (Bedenbaugh et al., 2022). Furthermore, prior studies (Cho et al., 2023; Dahir et al., 2024; Gui et al., 2023; Sweeney et al., 2021), as well as the results shown here, demonstrate differing effects of MC3R deletion on feeding, emotional behaviour and motivation in male and female mice. Thus, to determine how the melanocortin system responds to energy rheostatic challenges, as well as how this response may differ between male and female animals, we next utilized RNAscope *in situ* hybridization to quantify the mRNA expression of AgRP, POMC and MC3R in the MH in both male and female mice in the context of energy rheostatic challenges. WT male or female mice were either provided *ad libitum* access to a regular chow diet, HFD for 5 days or 5 days of semaglutide treatment, and then processed for *in situ* hybridization analysis of AgRP, POMC and MC3R in MH. Because fasting increases mRNA expression of AgRP (Liu et al., 2012; Makimura et al., 2003), we also quantified AgRP and POMC expression in a separate cohort of male mice that were fasted for 16 h overnight to validate the sensitivity of RNAscope analysis for detecting transcriptional changes in AgRP and POMC. As expected, fasting significantly increased mRNA expression of AgRP, and non-significantly reduced POMC expression (Fig. 4*D*, *F*, *J* and *L*). In male mice, no significant difference in POMC expression was observed following semaglutide treatment or HFD administration (Fig. 4*A*–*C* and *E*).

By contrast to male mice, acute exposure to semaglutide reduced mRNA expression of POMC in female animals, whereas HFD consumption did not alter POMC levels in females (Fig. 4*M*–*P*). AgRP mRNA levels were drastically decreased following HFD administration in male mice, an effect that was not observed in female animals (Fig. 4*G*–*K* and *Q*–*T*). By contrast, semaglutide administration did not alter AgRP mRNA expression in male or female mice (Fig. 4*K* and *T*).

Because MC3R signalling regulates the acute anorexic and orexigenic response to semaglutide and HFD, and MC3R directly responds to changes in AgRP and POMC peptide levels, we next tested whether the mRNA expression of MC3R was altered within the MH in male and female mice (ARC, VMH and DMH) following acute semaglutide or HFD administration. Both HFD and semaglutide administration had no effect on mRNA expression of MC3R within the ARC in male or female mice (Fig. 5*A*–*H*). In the DMH, semaglutide administration reduced MC3R mRNA expression in male mice, an effect that was not observed in female animals. HFD administration did not alter MC3R mRNA expression in the DMH in male or female mice (Fig. 5*Q*–*X*). Within the VMH, semaglutide reduced MC3R mRNA expression in female mice, an effect that did not occur in male mice. However, HFD feeding did not alter MC3R expression in the VMH in male or female animals (Fig. 5*I*–*P*). Thus, acute semaglutide administration alters the mRNA expression of MC3R in medial hypothalamic nuclei in a sexually dimorphic manner.

## Role of MC3R in positive energy rheostasis does not require an ARC

The MH is comprised of multiple subregions (ARC, VMH and DMH) that each have critical roles in energy homeostasis (Andermann & Lowell, 2017). Importantly, MC3R

is expressed throughout the entire MH, with especially high expression observed in the ARC (Bedenbaugh et al., 2022). Most prior work indicates that the primary site of action for MC3R-mediated effects on feeding is the ARC of the hypothalamus, where MC3R acts to both control the activity level of AgRP neurons and regulate presynaptic release from AgRP neuron terminals (Ghamari-Langroudi et al., 2018; Gui et al.,

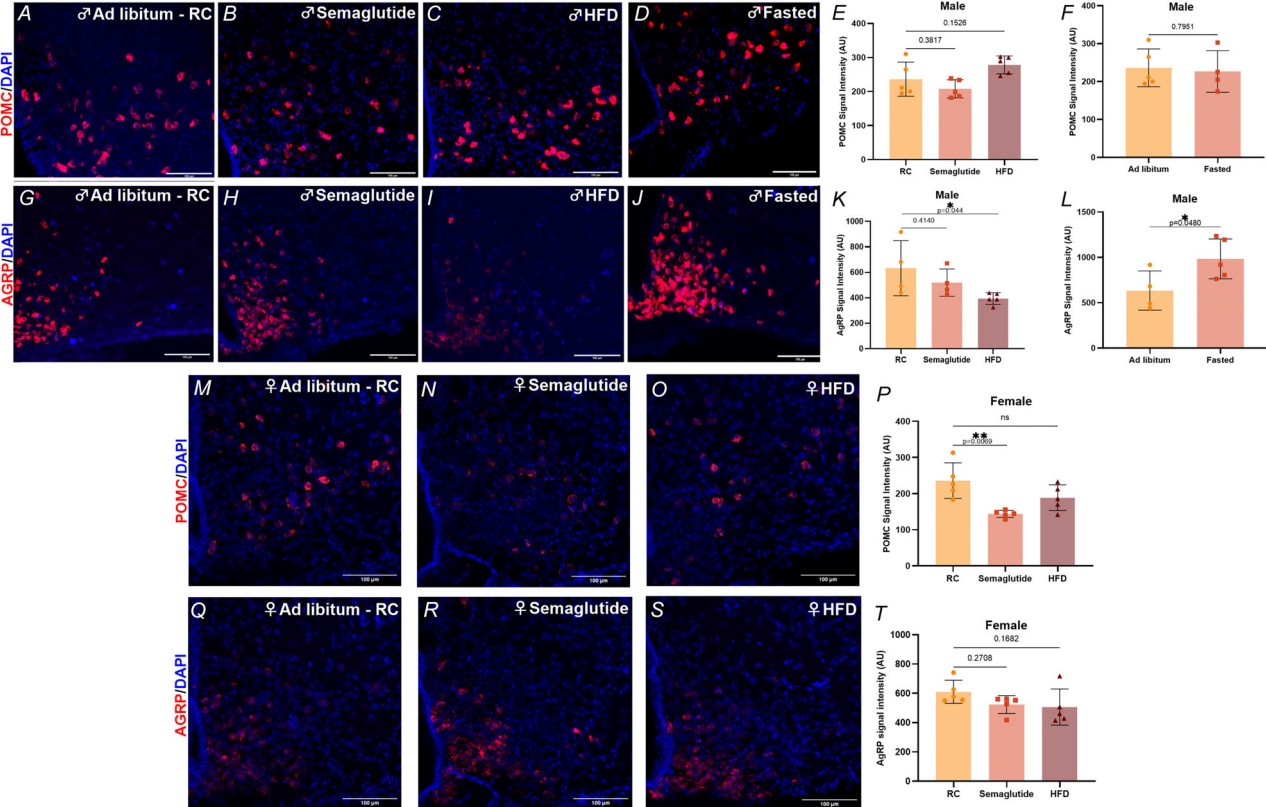

**Figure 4. Semaglutide treatment lowers expression of POMC in female mice, whereas HFD lowers AgRP expression in male mice**

*A–D*, confocal image of male POMC mRNA expression (red) with DAPI (blue) of a mouse with *ad libitum* access to regular chow (RC; *A*), after 5 days of semaglutide administration (*B*), after 5 days of high-fat diet (HFD) access (*C*) and after 16 h of fasting (*D*). Scale bars = 100 μm. *E*, comparison between the intensity of male POMC mRNA signal during *ad libitum*, semaglutide administration for 5 days, and access to HFD for 5 days (*n* = 5 mice per group, average from 3 sections per mouse, one-way ANOVA and Brown–Forsythe *post hoc* test, $F_{2,12}$ = 0.5289). *F*, comparison between the intensity of male POMC mRNA signal during *ad libitum* and 16 h of fasting (*n* = 5 mice *ad libitum*, *n* = 4 mice fasted, average from 3 sections per mouse, unpaired *t* test $F_{3,4}$ = 1.211). *G–J*, confocal image of AgRP mRNA expression (red) with DAPI (blue) of a mouse with *ad libitum* access to RC (*G*), after 5 days of semaglutide administration (*H*), after 5 days of HFD access (*I*) and after 24 h of fasting (*J*). Scale bar = 100 μm. *K*, comparison of the intensity of AgRP mRNA signal during *ad libitum* regular chow fed conditions *vs.* semaglutide administration for 5 days or access to HFD for 5 days (*n* = 4 mice RC, *n* = 4 mice semaglutide, *n* = 5 HFD, 3 sections per mouse one-way ANOVA and Brown–Forsythe *post hoc* test, $F_{2,10}$ = 3.609. *L*, comparison between the intensity of AgRP mRNA signal during *ad libitum* regular chow fed conditions and 16 h of fasting (*n* = 4 mice *ad libitum*, *n* = 5 mice fasted, average from 3 sections per mouse, unpaired *t* test $F_{4,3}$ = 1.029). *M–O*, confocal image of female POMC mRNA expression (red) with DAPI (blue) of a mouse with *ad libitum* access to RC (*M*), after 5 days of semaglutide administration (*N*) and after 5 days of HFD access (*O*). Scale bar = 50 μm. *P*, comparison of the intensity of female POMC mRNA signal during *ad libitum* fed conditions with regular chow, semaglutide administration for 5 days, and access to HFD for 5 days (*n* = 5 mice per group, average from 3 sections per mouse, one-way ANOVA and Brown–Forsythe *post hoc* test, $F_{2,12}$ = 1.877). *Q–S*, confocal image of female AgRP mRNA expression (red) of a mouse with *ad libitum* access to RC (*Q*), after 5 days of semaglutide administration (*R*) and after 5 days of HFD access (*S*). Scale bar = 50 μm. *T*, comparison of the intensity of female AgRP mRNA signal during *ad libitum*, semaglutide administration for 5 days, and access to HFD for 5 days (*n* = 5 mice per group, average from 3 sections per mouse, one-way ANOVA and Brown–Forsythe *post hoc* test, $F_{2,12}$ = 1.844). Data points represent the average of the signal intensity from all the sections of each individual animal. Error bars represent the SD. [Colour figure can be viewed at wileyonlinelibrary.com]

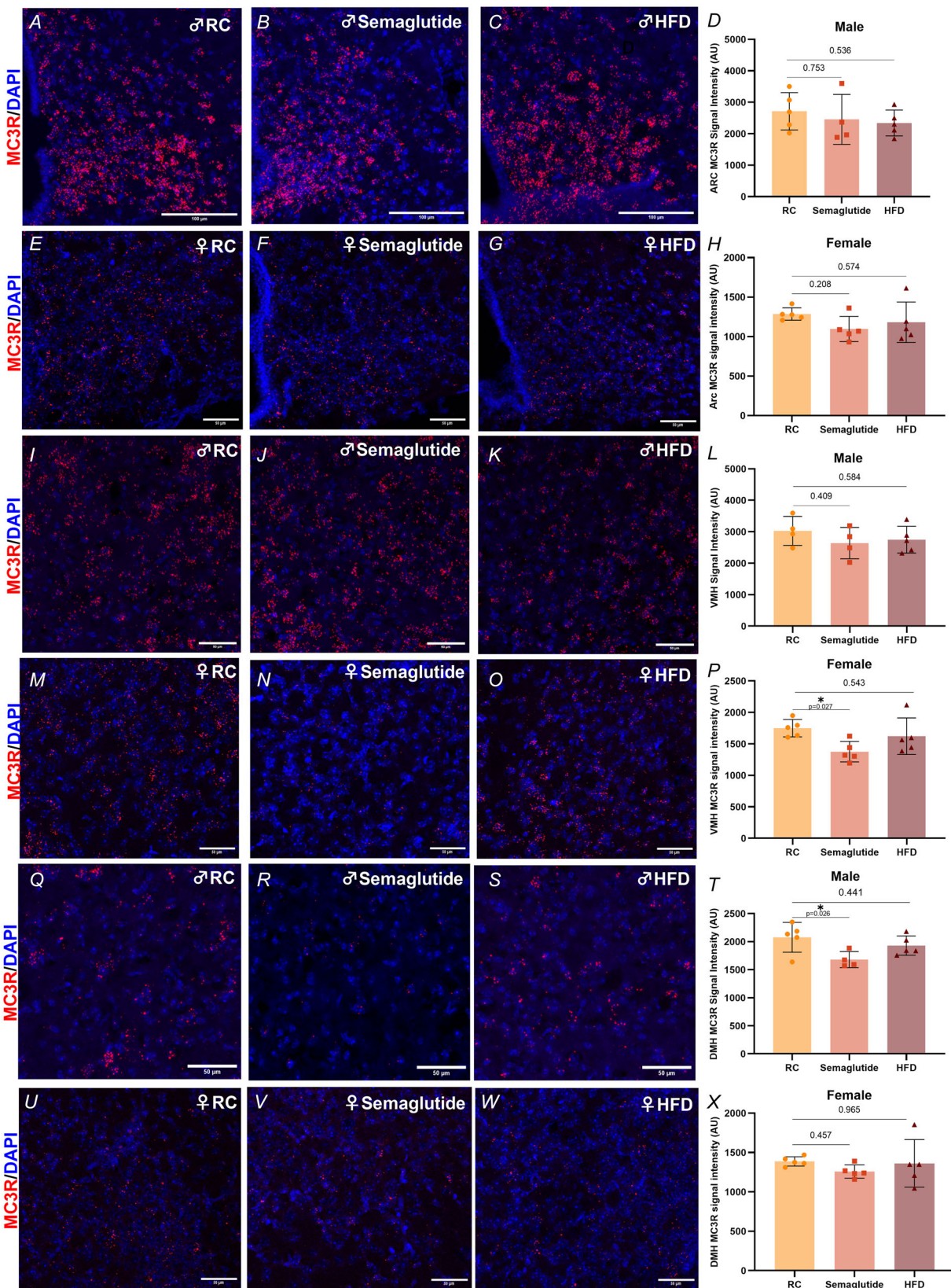

**Figure 5. Semaglutide treatment lowers MC3R mRNA expression in DMH in male mice and VMH in female mice**

*A–C*, confocal images of male MC3R mRNA (red) expression in ARC with DAPI (blue) of a mouse with *ad libitum* access to RC (*A*), after 5 days of semaglutide administration (*B*), after 5 days of HFD access (*C*). Scale bar = 100 μm.

*D*, comparison of the intensity of male MC3R mRNA signal in ARC during *ad libitum*, semaglutide administration for 5 days, and access to HFD for 5 days ($n$ = 5 mice RC, $n$ = 4 mice semaglutide, $n$ = 5 mice HFD, average from 3 sections per mouse, one-way ANOVA and Brown–Forsythe *post hoc* test, $F_{2,11}$ = 0.4235). *E–G*, confocal image of female MC3R mRNA (red) expression in ARC with DAPI (blue) of a mouse with *ad libitum* access to RC (*E*), after 5 days of semaglutide administration (*F*) and after 5 days of HFD access (*G*). Scale bar = 50 μm. *H*, comparison of the intensity of female MC3R mRNA signal in ARC during *ad libitum* regular chow fed conditions, semaglutide administration for 5 days, and access to HFD for 5 days ($n$ = 5 mice per group, average from 3 sections per mouse, one-way ANOVA and Brown–Forsythe *post hoc* test, $F_{2,12}$ = 0.8073). *I–K*, confocal image of male MC3R mRNA (red) expression in VMH with DAPI (blue) of a mouse with *ad libitum* access (*I*), after 5 days of semaglutide administration (*J*) and after 5 days of HFD access (*K*). Scale bar = 50 μm. *L*, comparison of the intensity of male MC3R mRNA signal in VMH during *ad libitum*, semaglutide administration for 5 days and access to HFD for 5 days ($n$ = 4 mice RC, $n$ = 4 mice semaglutide, $n$ = 5 mice HFD, average from 3 sections per mouse, one-way ANOVA and Brown–Forsythe *post hoc* test, $F_{2,10}$ = 0.0954). *M–O*, confocal image of female MC3R mRNA (red) expression in VMH with DAPI (blue) of a mouse with *ad libitum* access to regular chow (*M*), after 5 days of semaglutide administration (*N*), and after 5 days of HFD access (*O*). Scale bar = 50 μm. *P*, comparison of the intensity of female MC3R mRNA signal in VMH during *ad libitum* regular chow fed conditions, semaglutide administration for 5 days and access to HFD for 5 days ($n$ = 5 mice per group, average from 3 sections per mouse, one-way ANOVA and Brown–Forsythe *post hoc* test, $F_{2,12}$ = 0.3578). *Q–S*, confocal image of male MC3R mRNA (red) expression in DMH with DAPI (blue) of a mouse with *ad libitum* access to regular chow (*Q*), after 5 days of semaglutide administration (*R*), and after 5 days of HFD access (*S*). Scale bar = 50 μm. *T*, comparison of the intensity of MC3R mRNA signal in DMH during *ad libitum* regular chow fed conditions, semaglutide administration for 5 days, and access to HFD for 5 days ($n$ = 5 mice RC, $n$ = 4 mice semaglutide, $n$ = 5 mice HFD, average from 3 sections per mouse, one-way ANOVA and Brown–Forsythe *post hoc* test, $F_{2,11}$ = 0.1955). *U–W*, confocal image of female MC3R mRNA (red) expression in DMH with DAPI (blue) of a mouse with *ad libitum* access to regular chow food (*U*), after 5 days of semaglutide administration (*V*) and after 5 days of HFD access (*W*). Scale bar = 50 μm. *X*, comparison of the intensity of female MC3R mRNA signal in DMH during *ad libitum* regular chow fed conditions, semaglutide administration for 5 days, and access to HFD for 5 days ($n$ = 5 mice per group, average from 3 sections per mouse, one-way ANOVA and Brown–Forsythe *post hoc* test, $F_{2,12}$ = 1.943). Data points represent the average of the signal intensity from all the sections of each individual animal. Error bars represent the SD. [Colour figure can be viewed at wileyonlinelibrary.com]

2023). However, MC3R expression is also observed in the VMH (Bedenbaugh et al., 2022; Begriche et al., 2011; Sutton et al., 2021) and DMH (Bedenbaugh et al., 2022), comprising brain regions that are critically involved in feeding and energy homeostasis and where the function of MC3R is less well understood. Furthermore, MC3R mRNA expression is reduced in these regions following semaglutide treatment (Fig. 5*P* and *T*), suggesting that DMH and/or VMH may also contribute to the role of MC3R in energy rheostasis. We therefore next examined whether MC3R signalling in the DMH contributes to energy rheostasis. In new cohorts of mice, the AAV-Cre virus was targeted more dorsally in the MH in WT and MC3R floxed mice to determine whether the role of MH MC3R signalling in energy rheostasis requires the ARC (Figs 6*A*, *B* and 7). Viral expression was primarily localized to the DMH, with more sparse expression also observed in the VMH and posterior hypothalamus in some mice (subsequently referred to as dMH deletion of MC3R, Figs 6*B* and 8). However, no viral expression was observed in the ARC or lateral hypothalamus (Fig. 7). As previously described (Fig. 6*A*), we next characterized the response of dMH MC3R KO and WT mice to acute anorexic and orexigenic challenges (Figs 6*A* and 8). As observed with MH deletion of MC3R, deletion of MC3R in the dMH did not alter body weight or food intake in male or female mice on a regular chow *ad libitum* diet (Fig. 6*C–F*). Because MC3R KO mice have a defective orexigenic response to both fasting and caloric restriction (Ghamari-Langroudi et al., 2018), we first tested whether MC3R signalling in dMH also regulates weight regain following caloric restriction. During 70% caloric restriction, both WT and dMH MC3R KO mice lost similar amounts of weight (Fig. 6*G* and *H*). Following re-feeding in WT and dMH MC3R KO mice, both groups re-gained weight at a similar pace (Fig. 6*G* and *H*). Thus, in contrast to MC3R signalling in arcuate AgRP neurons, dMH MC3R signalling does not regulate weight regain following caloric restriction (Ghamari-Langroudi et al., 2018).

Following recovery from caloric restriction, we next tested whether dMH deletion of MC3R alters the anorexic response to anorexigenic stimuli by administering semaglutide or vehicle to WT and dMH MC3R KO mice. During semaglutide administration, female dMH MC3R KO mice lost slightly more weight than WT littermates, although food intake was similar between female WT and dMH MC3R KO mice during semaglutide treatment. Following the cessation of semaglutide treatment, female dMH MC3R KO mice maintained a slightly lower body weight than WT littermates with no significant difference in food intake between female WT and dMH MC3R KO mice during this period (Fig. 8*A* and *C*). During semaglutide treatment, male dMH MC3R KO mice lost a similar amount of weight as WT littermates but consumed significantly less food than WT mice during semaglutide treatment (Fig. 8*B* and *D*). Following the cessation of semaglutide treatment, WT and dMH MC3R KO male

mice re-gained weight at a similar pace and consumed similar amounts of food (Fig. 8*B* and *D*).

Following recovery from semaglutide treatment (Fig. 6*A*), we next tested whether dMH deletion of MC3R alters feeding and body weight when mice are provided a HFD. Female dMH MC3R KO mice gained more weight on a HFD than WT littermates, although food intake was similar between WT and dMH MC3R KO mice during HFD administration (Fig. 8*E* and *G*). Similar to female mice, male MC3R dMH KO mice had a significant increase in body weight following HFD

administration and consumed more HFD than WT littermate mice (Fig. 8*F* and *H*). To determine whether dMH MC3R deletion alters the anorexic response associated with switching HFD-fed mice to a regular chow diet, we measured feeding and body weight after changing HFD-fed mice to a regular chow diet. In female mice, dMH KO and WT mice demonstrated a similar anorexic and weight loss response following the switch to a regular chow diet. However, male dMH MC3R KO mice lost significantly less weight than WT littermate mice following the switch to a regular chow diet, although food

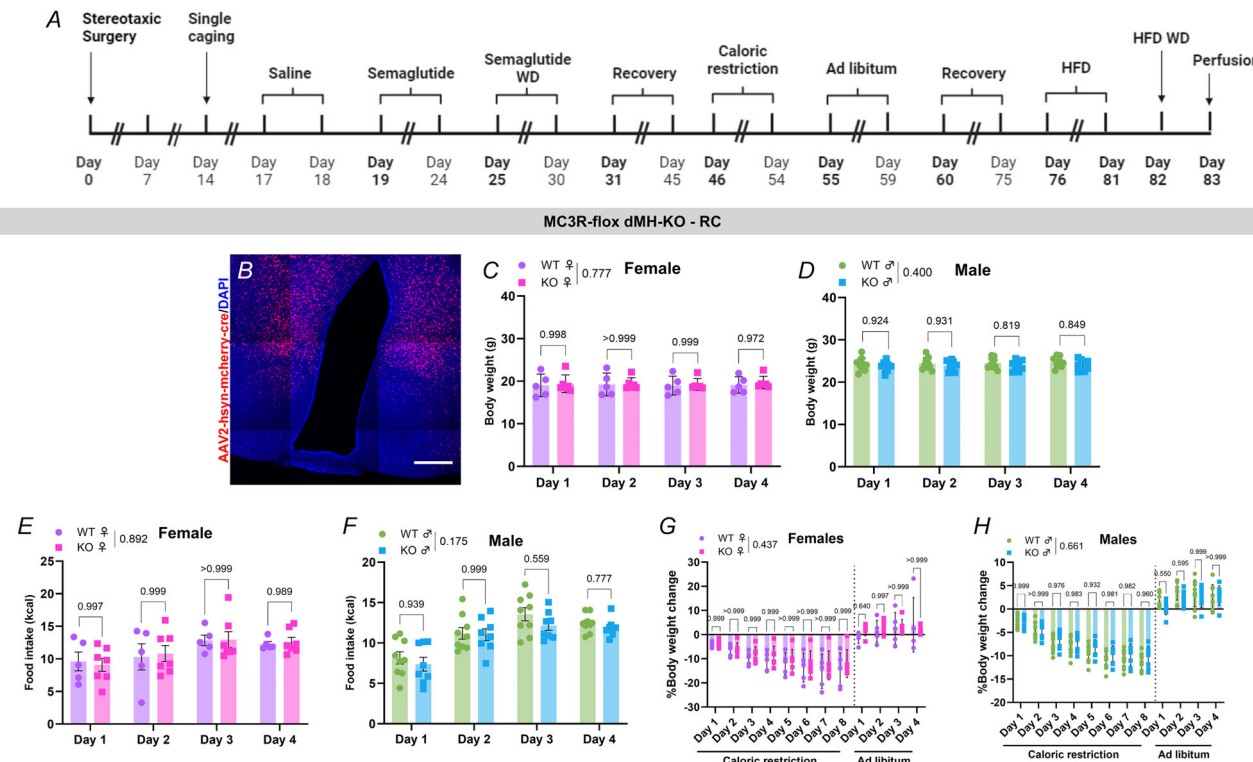

**Figure 6. Deletion of MC3R in DMH does not alter food intake or body weight on regular chow diet**
*A*, timeline of experimental protocol for experiments in Figs 6 and 7. Created with Biorender. *B*, example image of a MC3R-flox mouse with AVV2-hsyn-mcherry-cre in dMH. Scale bar = 300 μm. *C* and *D*, body weights with *ad libitum* access to regular chow 2 weeks after surgery in female mice (*C*) and male mice (*D*) (*C*, two-way ANOVA and Šídák's test *post hoc* analysis repeated measures, main effect of genotype $F_{1,9} = 0.085$, time $F_{1.1991,17.92} = 0.8354$ and interaction $F_{3,27} = 0.4361$, $n = 5$ mice for WT group and 7 mice for dMH KO group; *D*, two-way ANOVA and Šídák's test *post hoc* analysis repeated measures, main effect of genotype $F_{1,15} = 0.7518$, time $F_{1.428,21.42} = 3.207$ and interaction $F_{3,45} = 0.0852$, $n = 9$ mice for WT group and 8 mice for dMH KO group). *E* and *F*, regular chow food intake 2 weeks after surgery in female mice (*E*) and male mice (*F*) (*E*, two-way ANOVA and Šídák's test *post hoc* analysis, repeated measures, main effect of genotype $F_{1,10} = 0.0194$, time $F_{1.681,16.81} = 3.768$ and interaction $F_{3.30} = 0.0712$, $n = 5$ mice for WT group and 7 mice for dMH KO group; *F*, two-way ANOVA and Šídák's test *post hoc* analysis, repeated measures, main effect of genotype $F_{1,15} = 2.032$, time $F_{1.528,22.92} = 23.03$ and interaction $F_{3.45} = 0.2987$ $n = 9$ mice for WT group and 8 mice for dMH KO group). *G* and *H*, percentage body weight change during caloric restriction and after *ad libitum* access to food, using the body weight from last day of *ad libitum* access to regular chow as the baseline in female mice (*G*) and male mice (*H*) (*G*, baseline-corrected followed by repeated measures two-way ANOVA and Šídák's test *post hoc* analysis, main effect of genotype $F_{1,9} = 0.661$, time $F_{1.305,11.74} = 27.35$ and interaction $F_{11,99} = 0.3192$, $n = 5$ mice for WT group and 7 mice for dMH KO group, *H*, baseline-corrected followed by repeated measures two-way ANOVA and Šídák's test *post hoc* analysis, main effect $F_{1,15} = 0.199$, time $F_{1.637,24.55} = 250.4$ and interaction $F_{11,165} = 2.382$, $n = 9$ mice for WT group and 8 mice for dMH KO group). Data points represent individual mice, and error bars represent the SD. [Colour figure can be viewed at wileyonlinelibrary.com]

intake levels were similar in both groups of mice (Fig. 8*E* and *H*). Thus, although the ARC exerts an important role in energy rheostasis, additional regions outside the ARC (i.e. DMH area) are also capable of regulating energy rheostasis.

## dMH MC3R signalling selectively regulates the acquisition of weight gain on a HFD

Our previous results demonstrate that dMH deletion of MC3R increases weight gain following a HFD (Fig. 8*E* and *F*). However, it is unclear whether deletion of MC3R specifically regulates the initial acquisition of weight gain on a HFD, or whether deletion of MC3R amplifies weight gain in mice that already exhibit diet-induced obesity. To test this, in new cohorts of mice, we fed WT littermate and MC3R floxed mice with a HFD for 6 weeks to induce obesity (Fig. 9*A*). Importantly, no difference in initial weight gain or feeding was detected between WT and MC3R floxed mice prior to AAV injections, indicating that the presence of the floxed allele does not alter the baseline response to high-fat diet. Further, both groups of mice developed obesity at a similar rate (Fig. 9*B* and *C*). After 6 weeks of HFD, we injected AAV-Cre virus into

the dMH in WT and MC3R floxed mice and continued to measure feeding and body weight gain for three additional weeks. Similar to prior dMH targeted viral injections, AAV expression was primarily localized to the DMH regions, with sparse expression observed in the nearby posterior hypothalamus (Fig. 10*A* and *D*). By contrast to the initial response to novel HFD, dMH deletion of MC3R did not alter feeding or body weight gain when mice were accustomed to HFD and had already developed diet-induced obesity (Fig. 9*B–E*). Thus, MC3R signalling in dMH regulates the initial acquisition of weight gain in response to HFD but does not amplify weight gain following diet-induced obesity.

## MC3R deletion in dMH enhances weight loss to semaglutide in male diet-induced obese mice

Semaglutide has demonstrated remarkable efficiency in producing weight loss in the context of diet-induced obesity in humans and pre-clinical rodent models (Christou et al., 2019; Gabery et al., 2020). Therefore, since our previous experiments were performed in lean regular chow fed mice, we next tested whether dMH deletion of MC3R contributes to the anorexic response to

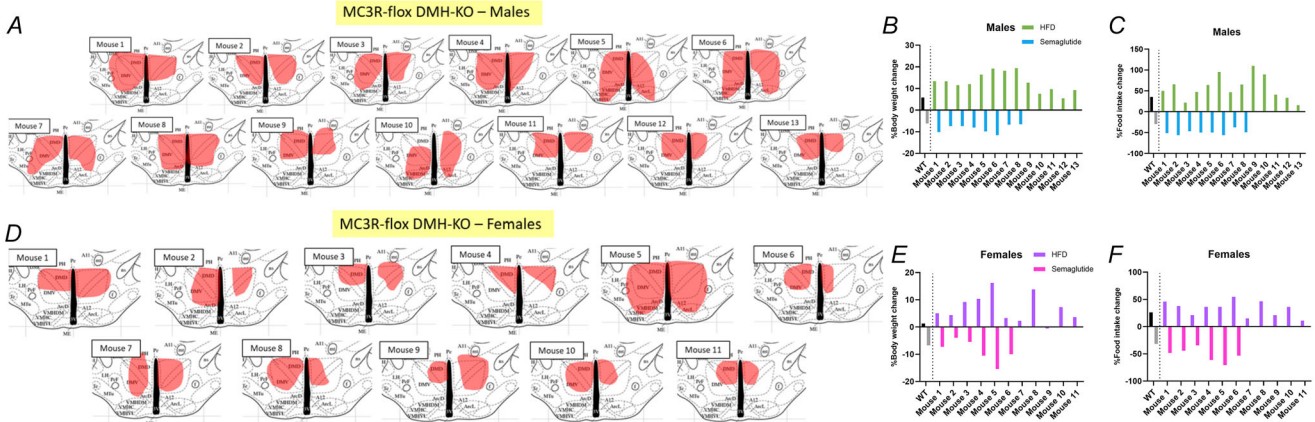

**Figure 7. Viral expression in dMH in MC3R floxed mice injected with AAV-Cre virus**
*A*, schematics of the viral expression of all the male MC3R-flox dMH KO mice used in the experiments in Figs 6 and 7. Adapted from Paxinos and Franklin's the Mouse Brain in Stereotaxic Coordinates, using the reference of bregma −1.94 mm. Mice without semaglutide results are from a cohort where only HFD experiment was performed. *B*, individual percentage difference of body weight of male mice in the last day of HFD compared to the last day of regular chow, and after semaglutide treatment compared to the last day of saline injection. The WT is the average of the percentage body weight difference for all the animals in that group. *C*, individual percentage difference of the average daily food intake of male mice during HFD compared to the last day of regular chow, and during semaglutide treatment compared to last day of saline injection. The WT is the average of the percentage food intake difference for all the animals in that group. *D*, schematics of the viral expression of all the female MC3R-flox dMH KO mice used in the experiments in Figs 6 and 7. Adapted from Paxinos and Franklin's the Mouse Brain in Stereotaxic Coordinates, using the reference of bregma −1.94 mm. *E*, individual percentage difference of body weight of female mice in the last day of HFD compared to the last day of regular chow, and after semaglutide treatment compared to last day of saline injection. The WT is the average of the percentage body weight difference for all the animals in that group. *F*, individual percentage difference of the average daily food intake of female mice during HFD compared to the last day of regular chow, and during semaglutide treatment compared to last day of saline injection. The WT is the average of the percentage food intake difference for all the animals in that group. [Colour figure can be viewed at wileyonlinelibrary.com]

semaglutide in diet-induced obese mice (Figs 9*A* and 10). Male dMH MC3R KO mice displayed an enhanced anorexic and weight loss response following semaglutide administration, relative to WT littermate mice (Fig. 9*G* and *I*). However, dMH MC3R deletion in female mice did not alter the anorexic or weight loss response to

semaglutide (Fig. 9*F* and *H*). Following cessation of semaglutide treatment, the female mice did not show any feeding or body weight differences compared to WT mice (Fig. 9*F* and *H*). By contrast, male dMH MC3R-KO mice re-gained weight at a slower pace than WT littermates (Fig. 9*G* and *I*). Thus, dMH deletion of MC3R enhances

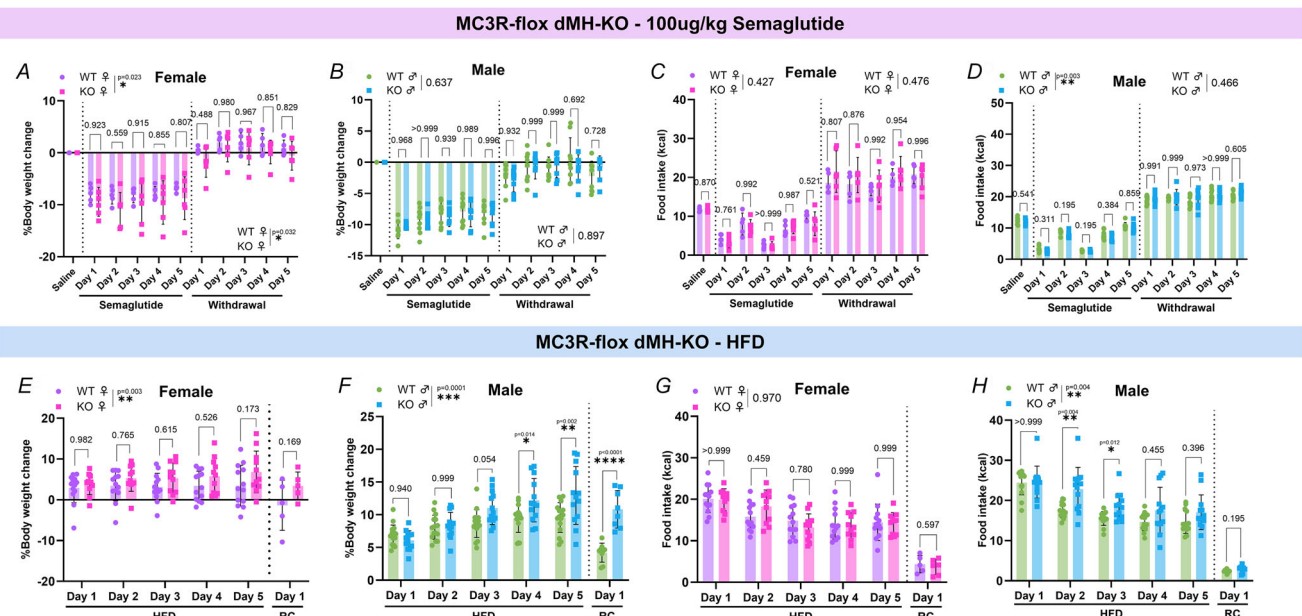

**Figure 8. dMH MC3R signalling regulates energy rheostasis**

*A* and *B*, daily percentage body weight change using saline as the baseline after 100 μg kg$^{-1}$ of semaglutide administration in female mice (*A*) and male mice (*B*) (*A*, baseline-corrected followed by repeated measures two-way ANOVA and Šídák's test *post hoc* analysis, main effect for semaglutide period: $F_{1,9} = 5.566$, time $F_{5,45} = 19.46$ and interaction $F_{5,45} = 0.2817$, and withdrawal period: $F_{1,9} = 4.867$, time $F_{5,45} = 1.726$ and interaction $F_{5,45} = 0.3296$, $n = 5$ mice in WT group and 6 mice in dMH KO group; *B*, baseline-corrected followed by repeated measures two-way ANOVA and Šídák's test *post hoc* analysis, main effect for semaglutide period of genotype $F_{1,15} = 0.2245$, time $F_{3.413,51.19} = 97.18$ and interaction $F_{5,75} = 0.3302$ and withdrawal period of genotype $F_{1,15} = 0.0169$, time $F_{3.005,45.08} = 3.326$ and interaction $F_{5,75} = 0.9344$, $n = 9$ WT mice and 8 dMH KO mice). *C* and *D*, 24 h food intake after daily semaglutide administration in WT and dMH MC3R KO mice (*C*, two-way ANOVA and Šídák's test *post hoc* analysis repeated measures, main effect for semaglutide period of genotype $F_{1,9} = 0.5840$, time $F_{2.610,23.49} = 80.93$ and interaction $F_{5,45} = 1.751$ and withdrawal period of genotype $F_{1,9} = 0.5518$, time $F_{2.952,26.56} = 4.745$ and interaction $F_{4,36} = 1.256$, $n = 5$ mice in WT group and 6 mice in dMH KO group; *D*, two-way ANOVA and Šídák's test *post hoc* analysis, repeated measures, main effect for semaglutide period of genotype $F_{1,15} = 16.86$, time $F_{2.679,40.19} = 287.5$ and interaction $F_{5,75} = 0.4670$ and withdrawal period of genotype $F_{1,15} = 0.5582$, time $F_{2.662,39.93} = 7.415$ and interaction $F_{4,60} = 0.4430$, $n = 9$ WT mice and 8 dMH KO mice). *E* and *F*, percentage of body weight change during access to *ad libitum* HFD, using the last day of regular chow as the baseline in female mice (*E*) and male mice (*F*) (*E*, baseline-corrected followed by repeated measures two-way ANOVA and Šídák's test *post hoc* analysis, main effect for HFD period of genotype $F_{1,22} = 8.998$, time $F_{2.151,47.33} = 0.6565$ and interaction $F_{4,88} = 0.3021$ and unpaired *t* test for regular chow period of genotype $F_{4,4} = 3.283$, $n = 13$ WT mice and 11 dMH KO mice with HFD, $n = 5$ mice per group with RC; *F*, baseline-corrected followed by repeated measures two-way ANOVA and Šídák's test *post hoc* analysis, main effect for HFD period of genotype $F_{1,27} = 15.60$, time $F_{1.857,50.14} = 18.62$ and interaction $F_{4,108} = 3.632$ and unpaired *t* test for regular chow period: $F_{7,8} = 3.913$, $n = 16$ WT mice and 13 dMH KO mice with HFD, $n = 9$ WT mice and 8 dMH KO mice with RC). *G* and *H*, daily food intake during HFD access in female mice (*G*) and male mice (*H*) (*G*, two-way ANOVA and Šídák's test *post hoc* analysis repeated measures, main effect for HFD period of genotype $F_{1,22} = 0.0014$, time $F_{2.650,58.30} = 16.83$ and interaction $F_{4,88} = 1.376$ and unpaired *t* test for regular chow period: $F_{4,5} = 1.236$, $n = 13$ WT mice and 11 dMH KO mice with HFD, $n = 5$ mice per group with RC; *H*, two-way ANOVA and Šídák's test *post hoc* analysis, repeated measures, main effect for HFD period of genotype $F_{1,25} = 9.915$, time $F_{1.974,49.36} = 335.18$ and interaction $F_{4,100} = 2.200$ and unpaired *t* test for regular chow period: $F_{7,8} = 5.673$, $n = 16$ WT mice and 13 dMH KO mice with HFD, $n = 9$ WT mice and 8 dMH KO mice with RC). Data points represent individual mice and error bars represent the SD. [Colour figure can be viewed at wileyonlinelibrary.com]

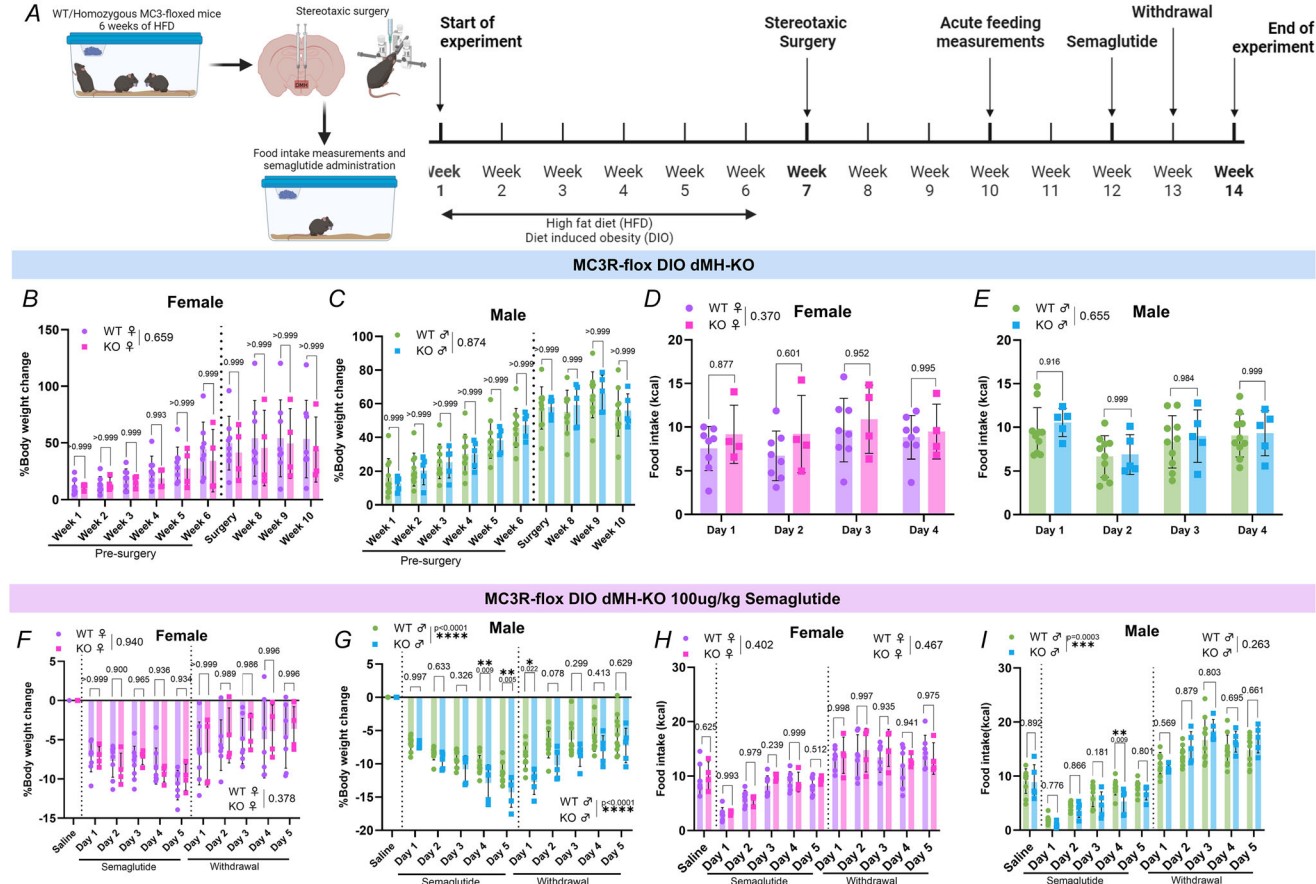

**Figure 9. Adult dMH deletion of MC3R selectively regulates the acquisition of body weight on high-fat diet (HFD)**

*A*, timeline of experimental protocol for experiments in Figs 9 and 10. Created with Biorender. *B* and *C*, percentage of body weight change during HFD, before and after stereotaxic surgery injecting AVV2-hsyn-mcherry-cre in dMH, using their body weight before the access to *ad libitum* HFD as baseline in female mice (*B*) and male mice (*C*) (*B*, baseline-corrected followed by repeated measures two-way ANOVA and Šídák's test *post hoc* analysis, main effect of genotype $F_{1,10} = 0.206$, time $F_{1.128,11.28} = 16.89$ and interaction $F_{9,90} = 0.2399$, $n = 8$ WT mice and 4 dMH KO mice; *C*, baseline-corrected followed by repeated measures two-way ANOVA and Šídák's test *post hoc* analysis, main effect of genotype $F_{1,13} = 0.0028$, time $F_{1.140,14.81} = 319.4$ and interaction $F_{9,117} = 0.9580$, $n = 10$ WT mice and 5 dMH KO mice). *D* and *E*, daily HFD food intake of DIO mice 2 weeks after stereotaxic surgery in female mice (*D*) and male mice (*E*) (*D*, two-way ANOVA and Šídák's test *post hoc* analysis repeated measures, main effect of genotype $F_{1,10} = 0.8813$, time $F_{3,30} = 2.567$ and interaction $F_{3,30} = 0.3708$, $n = 8$ WT mice and 4 dMH KO mice; *E*, two-way ANOVA and Šídák's test *post hoc* analysis repeated measures, main effect of genotype $F_{1,13} = 0.209$, time $F_{3,39} = 10.65$ and interaction $F_{3,39} = 0.2037$, $n = 10$ WT mice and 5 dMH KO mice). *F* and *G*, daily percentage of body weight change using saline as the baseline of DIO mice administered daily with 100 μg kg$^{-1}$ of semaglutide in female mice (*F*) and male mice (*G*) (*F*, baseline-corrected followed by repeated measures two-way ANOVA and Šídák's test *post hoc* analysis, main effect for semaglutide period of genotype $F_{1,10} = 0.0056$, time $F_{3,30} = 45.23$ and interaction $F_{3,30} = 0.6289$ and withdrawal period of genotype $F_{1,10} = 0.7921$, time $F_{3,30} = 4.325$ and interaction $F_{3,30} = 0.0742$, $n = 8$ WT mice and 4 dMH KO mice; *G*, baseline-corrected followed by repeated measures two-way ANOVA and Šídák's test *post hoc* analysis, main effect for semaglutide period of genotype $F_{1,13} = 20.96$, time $F_{5,65} = 151.2$ and interaction $F_{5,65} = 2.440$ and withdrawal period of genotype $F_{1,13} = 21.17$, time $F_{5,65} = 41.10$ and interaction $F_{5,65} = 1.246$, $n = 10$ WT mice and 5 dMH KO mice). *H* and *I*, daily HFD food intake of DIO mice during semaglutide administration in female mice (*H*) and male mice (*I*) (*H*, two-way ANOVA and Šídák's test *post hoc* analysis repeated measures, main effect for semaglutide treatment of genotype $F_{1,10} = 0.856$, time $F_{2.839,28.39} = 28.60$ and interaction $F_{5,50} = 0.6799$ and withdrawal period of genotype $F_{1,10} = 0.575$, time $F_{2.839,28.39} = 0.7946$ and interaction $F_{5,50} = 0.3505$, $n = 8$ WT mice and 4 dMH KO mice; *I*, two-way ANOVA and Šídák's test *post hoc* analysis repeated measures for semaglutide period of genotype $F_{1,13} = 3.162$, time $F_{3.132,40.72} = 53.38$ and interaction $F_{5,65} = 1.408$ and withdrawal period of genotype $F_{1,13} = 1.367$, time $F_{2.453,31.89} = 13.33$ and interaction $F_{4,52} = 1.214$, $n = 10$ WT mice and 5 dMH KO mice). Data points represent individual mice and error bars represent the SD. [Colour figure can be viewed at wileyonlinelibrary.com]

the anorexic response to semaglutide in both lean and obese male mice.

## Activation of DMH MC3R neurons acutely alters energy homeostasis

MC3R neurons in the ARC exert an orexigenic effect on feeding, primarily by regulating the activity of AgRP neurons (Gui et al., 2023). However, the role of MC3R neurons in medial hypothalamic populations outside of the ARC has been less well characterized. Given that MC3R deletion in the dorsal portions of the MH (i.e. VMH and DMH) altered energy rheostasis (Fig. 7), we next utilized immunohistochemical approaches to characterize putative inputs from AgRP and POMC neurons to VMH and DMH MC3R neurons (Fig. 11). Immunohistochemical assays indicate strong AgRP and POMC neuronal projections co-localizing with MC3R expressing cells in DMH (Fig. 11*A–C*) and ARC (Fig. 11*G–I*), but not VMH (Fig. 11*D–F*). These neuroanatomical results are consistent with functional melanocortin signalling in DMH MC3R neurons, although further electrophysiological assays are required to confirm functional connectivity. To determine the

function of VMH and DMH MC3R neurons in energy homeostasis, we next targeted the chemogenetic activator hM3Dq or control mCherry expressing virus to MC3R containing neurons in the DMH or the VMH in two separate cohorts of mice. Comprehensive metabolic profiling was performed on these mice following acute stimulation of MC3R neurons in VMH or DMH (Fig. 12*A–C*). Administration of the DREADD agonist CNO (1 mg kg$^{-1}$, I.P.) increased cfos expression in hM3Dq expressing neurons, but not mice containing a control mCherry fluorescent protein (Fig. 12*D–G* and *K*), indicating that CNO successfully increased the activity of MC3R neurons. CNO administration to mice expressing hM3Dq in VMH MC3R neurons did not alter locomotion (Fig. 12*H*), energy expenditure (Fig. 12*I*) or food intake (Fig. 12*J*). Similar null results were obtained in mice expressing control fluorescent protein in VMH MC3R neurons. By contrast to the activation of VMH MC3R neurons, acute activation of DMH MC3R neurons significantly increased locomotion and energy expenditure (Fig. 12*L* and *M*), without altering food intake (Fig. 12*N*). These effects were not the result of potential off-target effects of CNO because no difference in locomotion, energy expenditure or feeding

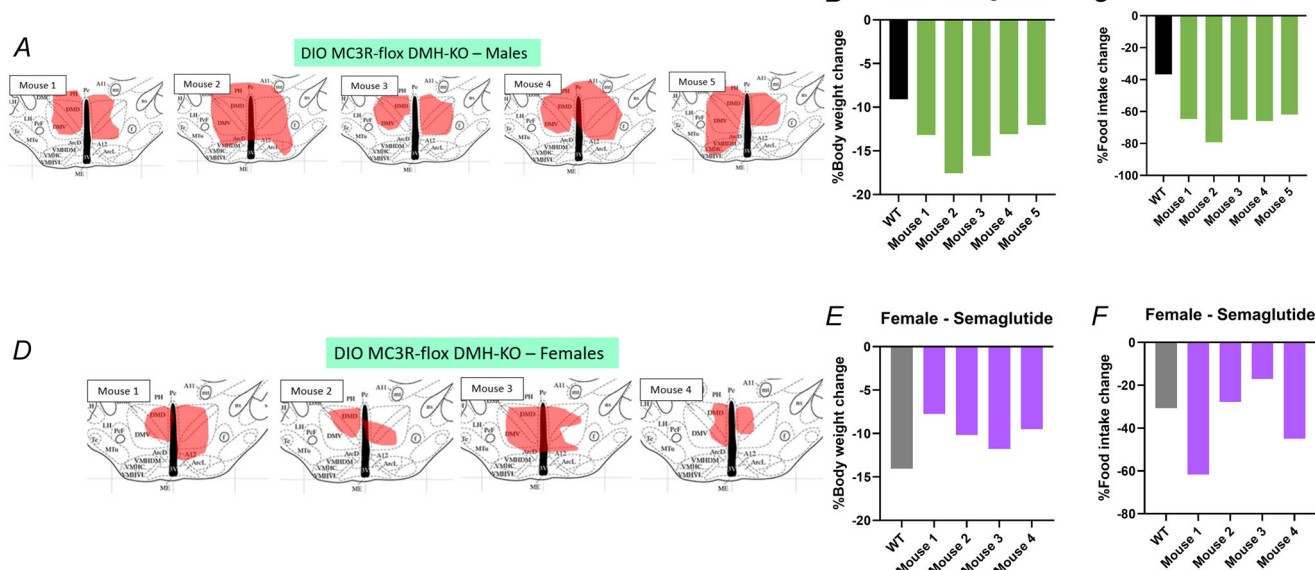

**Figure 10. Viral expression in MC3R floxed mice injected with AAV-Cre into dMH for DIO experiments**
*A*, schematics of the viral expression of all the male MC3R-flox dMH KO mice used in the experiments in Fig. 9. Adapted from Paxinos and Franklin's the Mouse Brain in Stereotaxic Coordinates, using the reference of bregma −1.94 mm. *B*, individual percentage difference of body weight of male mice after semaglutide treatment compared to last day of saline injection. *C*, individual percentage difference of the average food intake of male mice during semaglutide treatment compared to last day of saline injection. *D*, schematics of the viral expression of all the female MC3R-flox dMH KO mice used in the experiments in Fig. 9. Adapted from Paxinos and Franklin's the Mouse Brain in Stereotaxic Coordinates, using the reference of bregma −1.94 mm. *E*, individual percentage difference of body weight of female mice after semaglutide treatment compared to last day of saline injection. *F*, individual percentage difference of the average food intake of female mice during semaglutide treatment compared to last day of saline injection. [Colour figure can be viewed at wileyonlinelibrary.com]

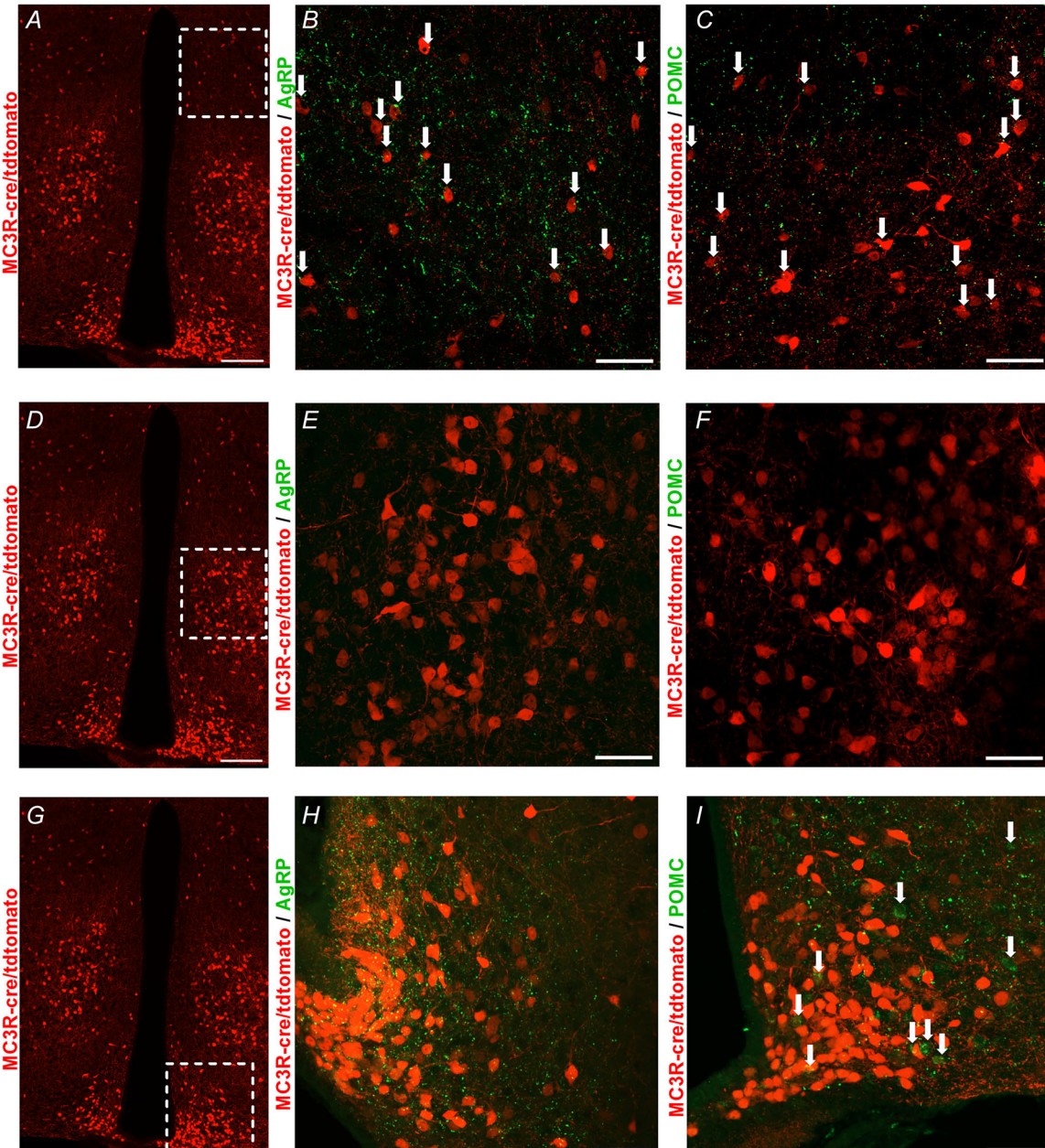

**Figure 11. POMC and AgRP fibres are colocalized with MC3R expressing neurons in DMH**
*A*, MC3R-cre/tdtomato medial hypothalamic section showing the cells expressing MC3R in red. The dotted squared highlights the DMH area. Scale bar = 150 µm. *B* and *C*, DMH images of MC3R-cre/tdtomato mouse. White arrows pointing to colocalization of MC3R expressing cells (red) and (*B*) AgRP fibres (green) or (*C*) POMC fibres (green). Scale bars = 50 µm. *D*) MC3R-cre/tdtomato MH section showing the cells expressing MC3R in red. The dotted squared highlighting the VMH area. Scale bar = 150 µm. *E* and *F*, VMH images of MC3R-cre/tdtomato mouse. Note the lack of AgRP and POMC immunoreactivity in this region. Scale bars = 50 µm. *G*, MC3R-cre/tdtomato MH section showing the cells expressing MC3R in red. The dotted squared highlighting the ARC area. Scale bar = 150 µm. *H* and *I*, ARC images of MC3R-cre/tdtomato mouse. White arrows pointing to colocalization of MC3R expressing cells (red) and (*H*) AgRP fibres (green) or (*I*) POMC fibres (green) and white arrows pointing to POMC cell bodies. Scale bars = 50 µm. [Colour figure can be viewed at wileyonlinelibrary.com]

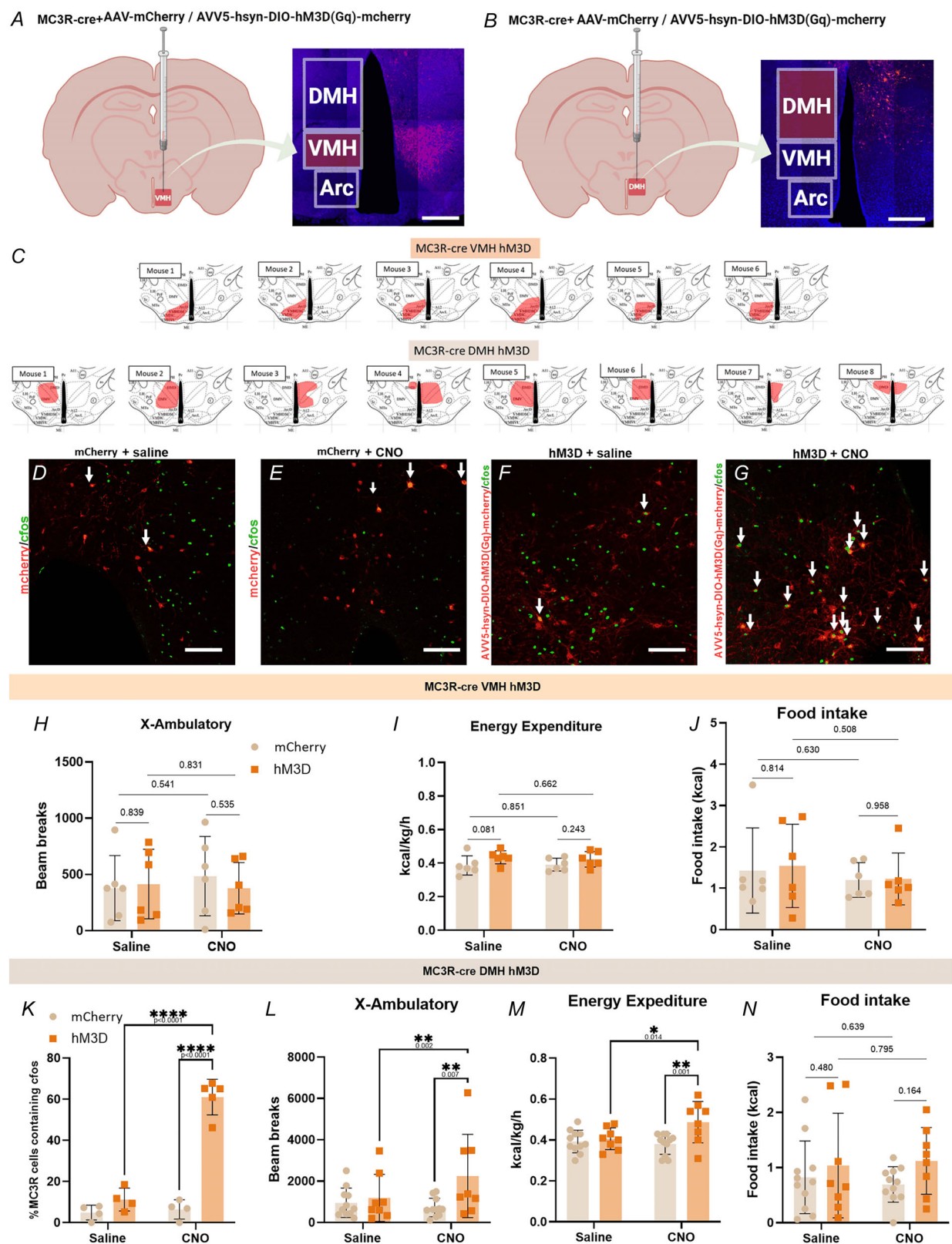

**Figure 12. Activation of MC3R expressing cells in DMH (and not VMH) increases energy expenditure and locomotion**

*A*, schematics of the stereotaxic surgery location of MC3R-cre mice with unilateral injection of hM3Dq or mcherry viral vector in VMH. Scale bar = 300 μm. Created with Biorender. *B*, schematics of the stereotaxic surgery location of

MC3R-cre mice injected with hM3Dq or mcherry viral vector in DMH. Scale bar = 300 μm. Created with Biorender. *C*, schematics of the viral expression of all the MC3R-cre hM3Dq mice used in the experiments in VMH and DMH in Fig. 12. Adapted fromPaxinos and Franklin's the Mouse Brain in Stereotaxic Coordinates, using the reference of bregma −1.94 mm. *D* and *E*, brain section of a control mouse (DIO-mCherry injected) showing co-localization (white arrows) of cfos (green) and the viral vector (red). Scale bars = 100 μm (*D*) administered with saline and (*E*) administered with CNO. *F* and *G*, brain section of a hM3Dq mouse that was administered saline showing co-localization (white arrows) of cfos (green) and the viral vector (red). Scale bars = 100 μm. (*F*) Administered with saline and (*G*) administered with CNO. *H–J*, control and hM3Dq groups with expression in VMH, after saline or 1 mg kg$^{-1}$ CNO administration (*H*) total locomotion (x-ambulatory), (*I*) average of the energy expenditure and (*J*) total food intake [*H*, two-way ANOVA and Fisher's least significant (LSD) test *post hoc* analysis with factors of viral injection condition (mCherry *vs.* hM3Dq) and injection condition (saline *vs.* CNO), main effect of viral injection condition: $F_{1,20} = 0.090$, main effect of injection condition: $F_{1,20} = 0.0823$, interaction between injection condition and viral injection condition: $F_{1,20} = 0.3507$, 3 female and 3 male mice per group; *I*, two-way ANOVA and Fisher's LSD test *post hoc* analysis: main effect of viral injection condition: $F_{1,20} = 4.627$, main effect of injection condition: $F_{1,20} = 0.0321$, interaction between injection condition and viral injection condition: $F_{1,20} = 0.2008$; 3 female and 3 male mice per group; *J*, two-way ANOVA and Fisher's LSD test *post hoc* analysis: main effect of viral injection condition: $F_{1,20} = 0.0423$, main effect of injection condition: $F_{1,20} = 0.6772$, interaction between injection condition and viral injection condition $F_{1,20} = 0.017$, 3 female and 3 male mice per group]. *K*, percentage of MC3R cells that expressed cfos signal, comparing saline and 1 mg kg$^{-1}$ CNO administrations in both control and hM3Dq groups ($n = 4$ mice saline control, $n = 4$ mice saline hM3D, $n = 4$ mice CNO control, $n = 5$ mice CNO hM3D, 2 sections per mouse, two-way ANOVA and Fisher's LSD test *post hoc* analysis with factors of viral injection condition and injection condition; main effect of viral injection condition: $F_{1,13} = 102.0$, main effect of injection condition: $F_{1,13} = 72.39$, interaction between injection condition and viral injection condition, $F_{1,13} = 64.06$). *L–N*, control mCherry and hM3Dq groups with expression in DMH, after saline or 1 mg kg$^{-1}$ CNO administration (*L*) total locomotion (x-ambulatory), (*M*) average of the energy expenditure and (*N*) total food intake (*L*, two-way ANOVA and Fisher's LSD test *post hoc* analysis main effect of viral injection condition: $F_{1,17} = 3.160$, main effect of injection condition: $F_{1,17} = 4.738$, interaction between injection condition and viral injection condition: $F_{1,17} = 11.48$, $n = 5$ female controls and $n = 5$ female hM3D per group, 6 control males and 3 hM3Dq male mice per group; *M*, two-way ANOVA: main effect of viral injection condition: $F_{1,17} = 6.504$, main effect of injection condition: $F_{1,17} = 3.157$, interaction between injection condition and viral injection condition: $F_{1,17} = 5.671$, 5 female and 3 male mice per group; *N*, two-way ANOVA and Fisher's LSD test *post hoc* analysis main effect of viral injection condition: $F_{1,17} = 2.193$, main effect of injection condition: $F_{1,17} = 0.0118$, interaction between injection condition and viral injection condition: $F_{1,17} = 0.2609$, $n = 5$ female per group, 6 control males and 3 hM3Dq male mice per group). Data points represent individual mice for all panels and error bars represent the SD. [Colour figure can be viewed at wileyonlinelibrary.com]

was detected following CNO administration in mice expressing control mCherry virus in DMH (Fig. 12*L–N*).

Because stimulation of DMH MC3R neurons increased locomotion and energy expenditure, we next tested whether MC3R signalling in DMH contributes to locomotion and energy expenditure such that this could contribute to the role of DMH MC3R signalling in energy rheostasis. We performed indirect calorimetry experiments on new cohorts of MC3R floxed or WT male mice targeted with Cre virus in the DMH (Fig. 13). Because the previous effects observed in the energy rheostasis experiments (Figs 2, 7 and 9) were mostly observed after at least 3 days of food intake and body weight measurements, we measured locomotion, energy expenditure and food intake for 3 consecutive days in metabolic cages. In direct contrast with stimulation of DMH MC3R neurons, deletion of MC3R in the DMH reduced overall energy expenditure (Fig. 13*C*). Interestingly, reduced locomotion and energy expenditure was only observed in the light period (Fig. 13*D* and *H*), suggesting an interaction between circadian rhythms and the role of DMH MC3R signalling in metabolic function. No changes in body weight or food intake were observed during testing (Fig. 13*B* and *F*). Thus, DMH MC3R neurons acutely alter energy homeostasis by increasing energy expenditure and locomotor activity, whereas acute activation of MC3R cells in VMH does not alter energy homeostasis.

## MC3R cells in DMH are glutamatergic in a sexually dimorphic manner

Although MC3R is expressed in the DMH (Bedenbaugh et al., 2022), the specific DMH cell types containing MC3R are unknown. Furthermore, although increased MC3R expression has been characterized in male mice in the ARC, and no difference in MC3R expression is observed in the VMH (Bedenbaugh et al., 2022), it is not known whether MC3R expression and distribution are different in the DMH between male and female animals. Given the sexually dimorphic effects of MC3R deletion reported here and in prior studies (Dahir et al., 2024; Gui et al., 2023), as well as the role of DMH MC3R neurons in energy homeostasis (Figs 3, 8 and 9), we next utilized RNAscope *in situ* hybridization to characterize the identity of MC3R containing cells in the DMH in male and female mice. To characterize the DMH cell types expressing MC3R, we first attempted to analyse published single-cell and single-nucleus RNA-sequencing datasets

of the DMH. However, the low mRNA expression of many receptors critical for energy homeostasis (including MC3R and MC4R) did not allow for adequate detection of MC3R-expressing cells (data not shown). Therefore, we instead utilized high-sensitivity RNAscope *in situ* hybridization approaches to map the expression of MC3R containing DMH neurons to previous DMH cell types implicated in energy homeostasis, as well as to characterize the neurochemical identity of DMH MC3R cells. First, there were no differences in the number of

cells expressing MC3R or the intensity of the MC3R signal between female and male animals in the DMH (Fig. 14*H* and *I*). However, to further characterize potential sex differences in the distribution of MC3R in DMH cell types, we next quantified the distribution of MC3R in candidate DMH cell types in male and female mice.

Prior work indicates a critical role for DMH neurons containing the leptin receptor (LepR) in controlling feeding behaviour and circadian rhythms (Faber et al., 2021; Tang et al., 2023). To characterize the overlap

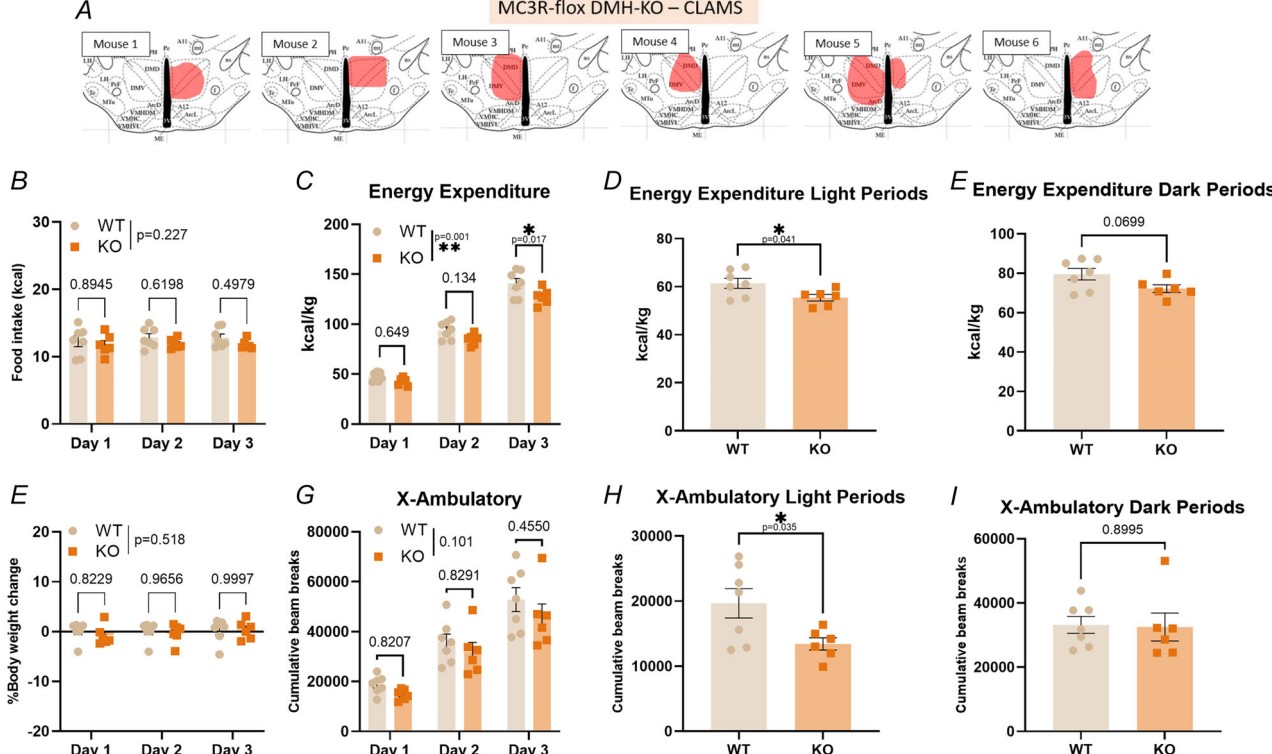

**Figure 13. Deletion of MC3R in DMH reduces energy expenditure and locomotion in mice**
*A*, schematics of the viral expression of all the male MC3R-flox mice used in the experiment shown in Fig. 13. Adapted from Paxinos and Franklin's the Mouse Brain in Stereotaxic Coordinates, using the reference of bregma −1.94 mm. *B*, daily food intake during the 3 days of the experiment (two-way ANOVA and Šídák's test *post hoc* analysis repeated measures, main effect of genotype: $F_{1,11} = 1.634$; main effect of day: $F_{2,22} = 0.4348$, interaction between genotype and day: $F_{2,22} = 0.2039$, $n = 7$ WT male mice and 6 DMH-KO male mice). *C*, cumulative 24 h energy expenditure of WT and MC3R-flox mice (two-way ANOVA and Šídák's test *post hoc* analysis repeated measures, main effect of genotype $F_{1,33} = 12.32$, main effect of day: $F_{2,33} = 395.3$; interaction between genotype and day: $F_{2,33} = 0.8882$, $n = 7$ WT male mice and 6 DMH-KO male mice). *D* and *E*, sum of the energy expenditure for 3 days. (*B*) During the light periods and (*C*) during the dark periods (*D*, unpaired *t* test $F_{6,5} = 2.652$, $n = 7$ WT male mice and 6 DMH-KO male mice; *E*, unpaired *t* test $F_{6,5} = 2.732$ $n = 7$ WT male mice and 6 DMH-KO male mice). *F*, percentage body weight change using as baseline each animal body weight before being in the CLAMS cages (baseline-corrected followed by two-way repeated measures ANOVA with and Šídák's test *post hoc* analysis repeated measures, main effect of genotype: $F_{1,33} = 0.4237$; main effect of day: $F_{2,33} = 0.3407$, interaction between genotype and day $F_{2,33} = 0.1886$, $n = 7$ WT male mice and 6 DMH-KO male mice). *G*, 3-day cumulative 24-h beam breaks (*x*-ambulatory) (two-way ANOVA and Šídák's test *post hoc* analysis repeated measures, main effect of genotype $F_{1,33} = 2,844$; main effect of day: $F_{2,33} = 41.62$; interaction between genotype and day: $F_{2,33} = 0.1118$, $n = 7$ WT male mice and 6 DMH-KO male mice). *H* and *I*, sum of the beam breaks for 3 days (*H*) during light periods and (*I*) during dark periods (*H*, unpaired *t* test, $n = 7$ WT male mice and 6 DMH-KO male mice $F_{6,5} = 6.768$; *I*, unpaired *t* test $F_{5,6} = 2.427$, $n = 7$ WT male mice and 6 DMH-KO male mice). Data points represent individual mice for all panels and error bars represent the SD. [Colour figure can be viewed at wileyonlinelibrary.com]

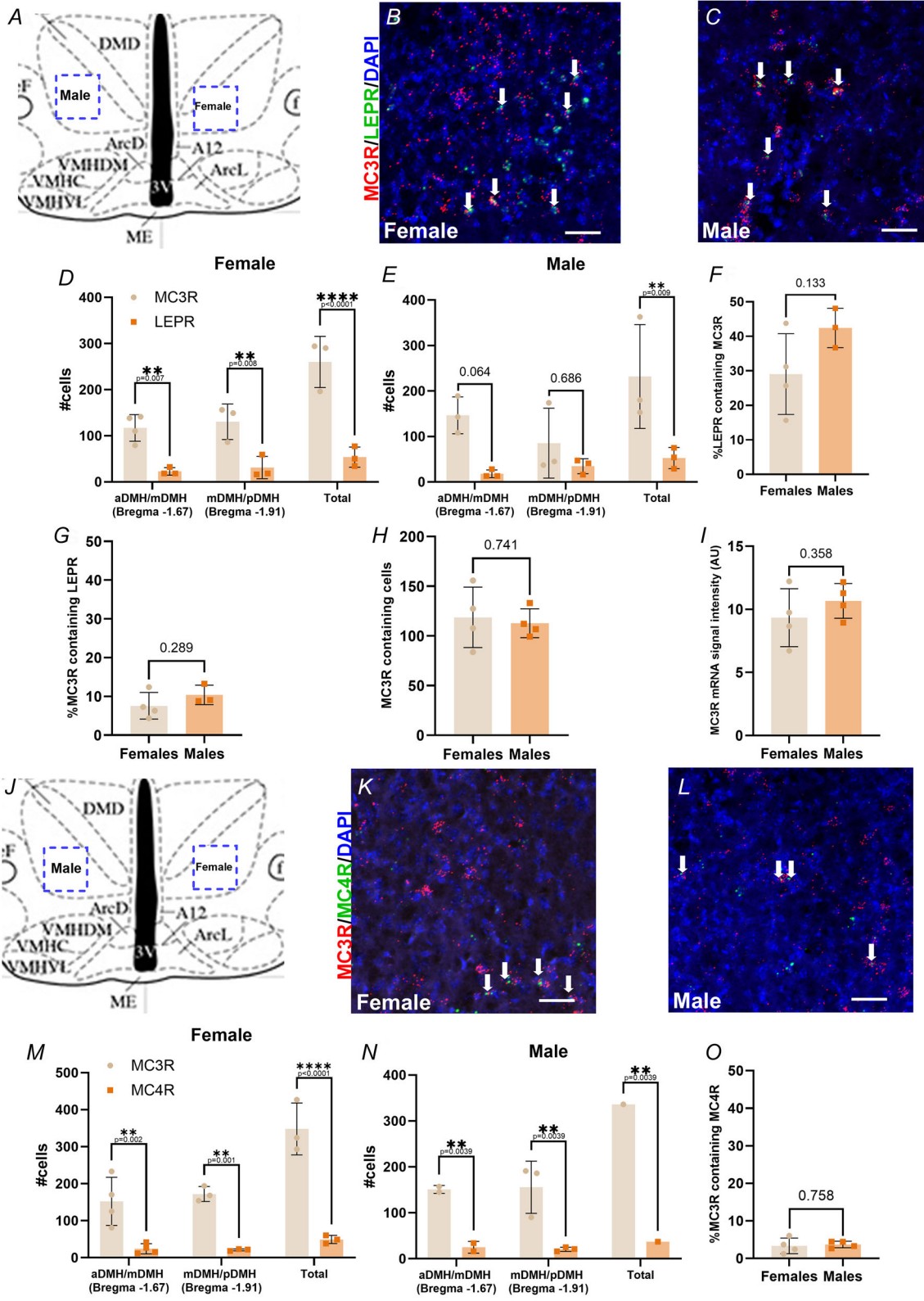

**Figure 14. Colocalization of DMH MC3R neurons with LepR and MC4R neurons**
*A*, schematics of the localization (blue square) of the RNAscope confocal image for MC3R and LEPR mRNA. Adapted from Paxinos and Franklin's the Mouse Brain in Stereotaxic Coordinates, bregma −1.94 mm. *B* and *C*, confocal image of the colocalization (white arrows) of MC3R (red) and LEPR (green) mRNA expression with DAPI (blue) in (*B*) a female mouse brain section and (*C*) a male mouse brain section. Scale bars = 40 μm. *D* and *E*, comparison

between of number of cells in anterior DMH (aDMH), medial DMH (mDMH) and posterior DMH (pDMH) expressing either MC3R or LEPR mRNA (*D*) in female mice and (*E*) male mice [$n = 4$ mice per group in total, 1 or 2 sections per mouse, two-way ANOVA and Šídák's test *post hoc* analysis and Šídák's test *post hoc* analysis, main effect of gene (MC3R *vs*. LepR): $F_{1,13} = 78.14$ for (*D*) and $F_{1,12} = 17.96$ for (*E*)]. *F*, percentage of LEPR cells that co-expressed MC3R in female and male mice ($n = 4$ female mice $n = 3$ male mice, average of 1 or 2 sections per mouse, unpaired *t* test $F_{3,2} = 4.211$). *G*, percentage of MC3R cells that co-expressed LEPR in female and male mice ($n = 4$ female mice $n = 3$ male mice, average of 1 or 2 sections per mouse, unpaired *t* test $F_{3,2} = 1.876$). *H*, measurement of the number of MC3R containing cells in males and females in DMH ($n = 4$ mice per group, average of 3 sections per mouse, unpaired *t* test $F_{3,3} = 4.405$). *I*, comparison of MC3R mRNA signal intensity between male and female mice in DMH ($n = 4$ mice per group, average of 3 sections per mouse, unpaired *t* test $F_{3,3} = 2.790$). *J*, schematics of the localization (blue square) of the RNAscope confocal image for MC3R and MC4R mRNA. Adapted from Paxinos and Franklin's the Mouse Brain in Stereotaxic Coordinates, bregma −1.94 mm. *K* and *L*, confocal image of the colocalization (white arrows) of MC3R (red) and MC4R (green) mRNA expression with DAPI (blue) in (*K*) a female mouse brain section and (*L*) a male mouse brain section. Scale bars = 40 μm. *M* and *N*, comparison between of number of cells in anterior DMH (aDMH), medial DMH (mDMH) and posterior DMH (pDMH) expressing either MC3R or LEPR mRNA (*M*) in female mice and (*N*) male mice [$n = 4$ mice per group in total, 1 or 2 sections per mouse, two-way ANOVA and Šídák's test *post hoc* analysis, main effect of gene (MC3R *vs*. MC4R): $F_{1,14} = 105.8$ for (*M*) and $F_{1,8} = 32.92$ for (*N*)]. *O*, percentage of MC3R cells that co-expressed MC4R in female and male mice ($n = 4$ mice per group, average from 1 or 2 sections per mouse, unpaired *t* test $F_{3,3} = 5.720$). Data points represent the average from each mouse across all quantified sections and error bars represent the SD. [Colour figure can be viewed at wileyonlinelibrary.com]

between DMH MC3R neurons and DMH LepR neurons, we co-localized mRNA for MC3R and LepR in the DMH of male and female mice. Both MC3R and LepR expressing cells were observed throughout the DMH (Fig. 14*A–E*). However, less than 10% of DMH MC3R neurons co-localized with LepR mRNA, indicating that the vast majority of DMH MC3R neurons do not express the leptin receptor (Fig. 14*G*). However, ∼30–40% of DMH LepR neurons did contain MC3R (Fig. 14*F*), indicating that, although the majority of MC3R neurons do not contain LepR, MC3R signalling may still directly regulate the activity of a subset of DMH LepR neurons. Similar co-expression patterns of LepR and MC3R were observed in both male and female mice (Fig. 14*F* and *G*).

Central melanocortin signalling is mediated by both MC3R and MC4R, and DMH MC4R neurons have previously been implicated in energy homeostasis (Chen et al., 2019; Han et al., 2023). Therefore, to quantify the expression of MC3R and MC4R in DMH, we performed dual RNAscope *in situ* hybridization for both MC3R and MC4R in male and female mice. MC3R expression vastly outnumbered MC4R expression in DMH at a ratio of almost 8:1 (Fig. 14*J–N*). Furthermore, less than 4% of DMH MC3R neurons co-express MC4R in both male and female mice (Fig. 14*O*). Thus, the majority of melanocortin signalling in DMH is probably mediated by MC3R, although further functional studies are required to confirm this observation.

The DMH consists of a mixture of inhibitory GABA neurons and excitatory glutamate neurons. To determine the neurochemical distribution of DMH MC3R neurons, we co-localized MC3R mRNA with either vGAT (GABA marker) or vGLUT2 (glutamate marker). In both males and females, ∼60% of DMH MC3R neurons co-localized with vGAT (Fig. 15*A–F*). However, although the total number of MC3R positive cells and the total amount of

MC3R mRNA transcripts in DMH did not differ between male and female mice (Fig. 14*H* and *I*), the percentage of MC3R cells that expressed vGLUT2 was significantly higher in males compared to females (Fig. 15*G–M*). A greater number of vGLUT2 positive cells was observed in the DMH of male *vs*. female animals, suggesting that the increased percentage of MC3R cells containing vGLUT2 in males is driven by an overabundance of vGLUT2 neurons in male mice (Fig. 15*M*). Thus, although the total number of MC3R neurons is similar between male and female mice in DMH, the distribution of MC3R in vGAT and vGLUT2 neurons differs between male and female animals.

## Discussion

Emerging evidence indicates that MC3R has a unique function compared to other components of the central melanocortin system (i.e. AgRP, POMC and MC4R) and other receptors implicated in energy homeostasis (i.e. leptin receptor, GLP1R, etc.). By contrast to the deletion of POMC and MC4R, the deletion of MC3R has minimal effects on food intake or body weight in mice (Ghamari-Langroudi et al., 2018) or rats (You et al., 2016) provided *ad libitum* access to a standard chow diet. However, MC3R KO mice are hypersensitive to a diverse array of anorexic stimuli including stress-related stimuli (Sweeney et al., 2021), anorexic drugs (Dahir et al., 2024; Ghamari-Langroudi et al., 2018; Sweeney et al., 2021) and tumour-associated anorexia (Marks & Cone, 2003; Marks et al., 2003). Paradoxically, these mice also exhibit an exaggerated response to orexigenic and anabolic stimuli, demonstrating increased weight gain following HFD and ovariectomy (Ghamari-Langroudi et al., 2018). Together, these findings suggest that MC3R controls the

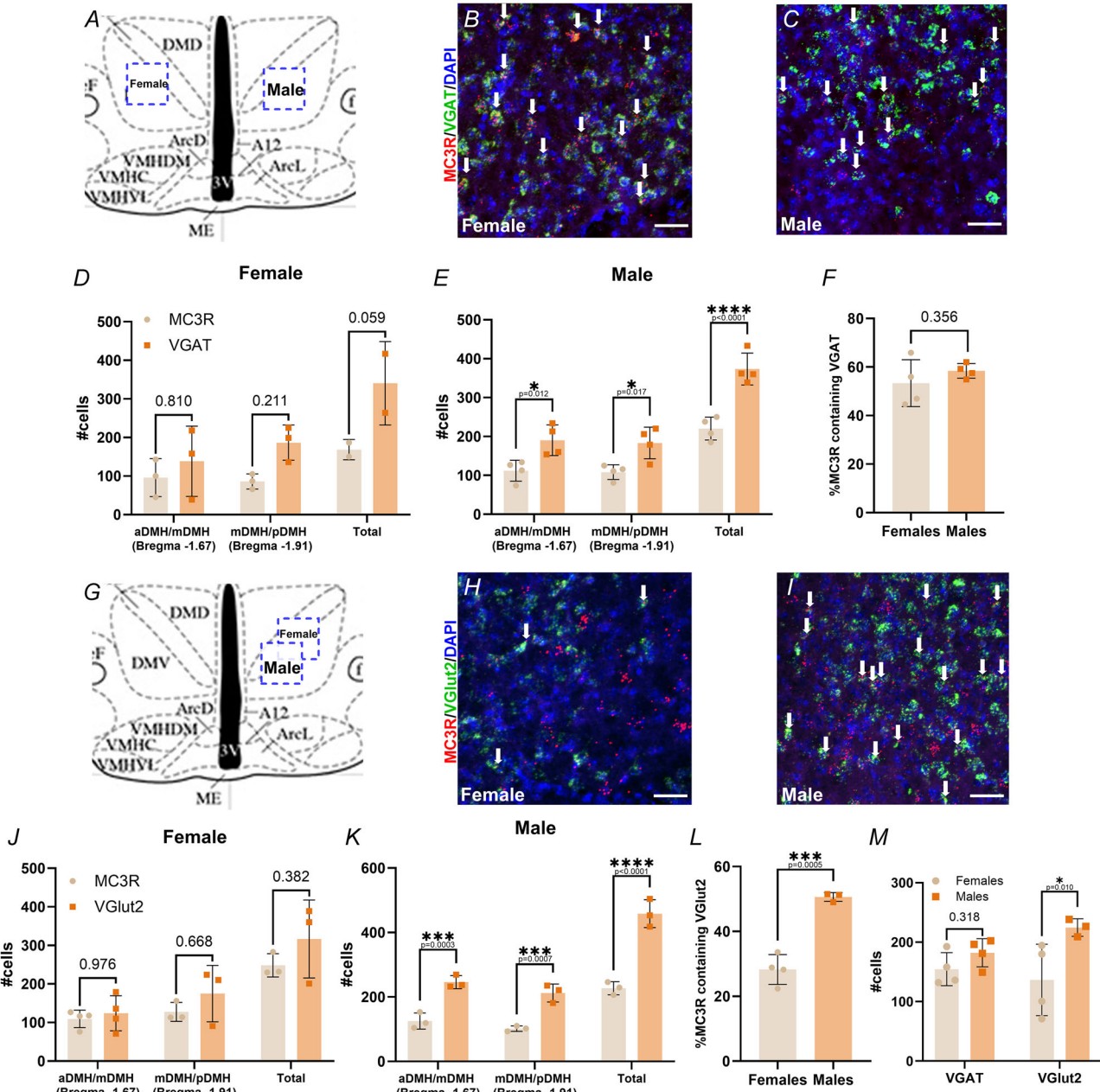

**Figure 15. Colocalization of DMH MC3R neurons with vGAT and vGLUT2 neurons**

*A*, schematics of the localization (blue square) of the RNAscope confocal image for MC3R and VGAT mRNA. Adapted from, bregma −1.94 mm. *B* and *C*, confocal image of the colocalization (white arrows) of MC3R (red) and VGAT (green) mRNA expression with DAPI (blue) in (*B*) a female mouse brain section (*C*) a male mouse brain section. Scale bars = 40 μm. *D* and *E*, comparison between of number of cells in anterior DMH (aDMH), medial DMH (mDMH) and posterior DMH (pDMH) expressing either MC3R or VGAT mRNA (*D*) in female mice and (*E*) male mice [*n* = 3 female mice, *n* = 4 male mice in total, average of 1 or 2 sections per mouse, two-way ANOVA and Šídák's test *post hoc* analysis, main effect of gene (MC3R *vs*. VGAT): $F_{1,10} = 10.97$ for (*D*) and $F_{1,18} = 55.04$ for (*E*)]. *F*, percentage of MC3R cells that co-expressed LEPR of female and male mice (*n* = 4 mice per group in total, average of 1 or 2 sections per mouse, unpaired *t* test $F_{3,3} = 10.33$). *G*, schematics of the localization (blue square) of the RNAscope confocal image for MC3R and VGlut2 mRNA. Adapted from Allen Brain Atlas – Mouse Brain, bregma −1.94 mm. *H* and *I*, confocal image of the colocalization (white arrows) of MC3R (red) and VGlut2 (green) mRNA expression with DAPI (blue) in *H*) a female mouse brain section *I*) a male mouse brain section. Scale bars = 40 μm. *J* and *K*, comparison between of number of cells in anterior DMH (aDMH), medial DMH (mDMH), and posterior DMH (pDMH) expressing either MC3R or VGlut2 mRNA (*J*) in female mice and (*K*) male mice [*n* = 4 female mice, *n* = 3 male mice in total, average of 1 or 2 sections per mouse, two-way ANOVA and Šídák's test *post hoc* analysis, main effect of gene (MC3R *vs*. VGLUT2): $F_{1,14} = 3.087$ for (*J*) and $F_{1,12} = 154.2$ for (*K*)]. *L*,

percentage of MC3R cells that co-expressed MC4R of female and male mice ($n = 4$ female mice, $n = 3$ male mice in total, average of 1 or 2 sections per mouse, unpaired $t$ test $F_{3,2} = 11.56$). *M*, total number of cells in female and male DMH expressing VGAT or VGlut2 mRNA [$n = 4$ female mice, $n = 3$ male mice in total, 1 or 2 sections per mouse, two-way ANOVA and Šídák's test *post hoc* analysis with factors of sex (male *vs.* female) and gene (VGAT *vs.* VGLUT2), main effect of sex: $F_{1,11} = 8.893$; main effect of gene: $F_{1,11} = 0.3889$; interaction between sex and gene: $F_{1,11} = 2.434$]. Data points represent the average from each mouse across all quantified sections and error bars represent the SD. [Colour figure can be viewed at wileyonlinelibrary.com]

magnitude of metabolic responses to both orexigenic and anorexic challenges (referred to here as altered energy rheostasis). However, because prior studies examining the role of MC3R in energy rheostasis were performed in global constitutive KO mouse models, it is not known whether these effects are the result of secondary changes associated with MC3R deletion or mediated by MC3R action on neural circuitry in adult animals. Furthermore, the specific brain regions mediating the role of MC3R in energy rheostasis are largely unknown.

Here, similar to prior findings in global MC3R KO mice (Ghamari-Langroudi et al., 2018; Lam et al., 2021; Sweeney et al., 2021), we show that adult-specific deletion of MC3R in the MH does not change basal feeding or body weight gain when mice are provided *ad libitum* access to regular chow diet (Fig. 1). By contrast, viral-mediated deletion of MC3R in MH recapitulates both the enhanced weight gain on a HFD (Fig. 3*E* and *H*) and the enhanced anorexic response to GLP1R stimulation (Fig. 3*A-D*) observed in global MC3R KO mice in a sexually dimorphic manner. Thus, despite the widespread expression of MC3R throughout the brain, many of the core features of energy rheostasis can be recapitulated by selectively deleting MC3R within a small subset of neurons in the MH. Together, these data indicate a functional role for medial hypothalamic MC3R signalling in energy rheostasis by controlling the magnitude of metabolic responses to both orexigenic and anorexic challenges, establishing a unique role for central MC3R signalling in energy homeostasis.

It is important to note that MC3R signalling is controlled by both an endogenous agonist ($\alpha$-MSH) and an endogenous antagonist (AgRP). Thus, the behavioural effects associated with MC3R deletion will probably depend on the endogenous levels of $\alpha$-MSH and AgRP in each behavioural state. Here, we utilized *in situ* hybridization to test the hypothesis that the levels of the melanocortin agonist ($\alpha$-MSH) and antagonist (AgRP) are oppositely altered following positive rheostasis (HFD administration) and negative rheostasis (semaglutide administration), leading to disparate effects of MC3R deletion in these two behavioural states. Our data indicate that acute HFD exposure reduces AgRP mRNA expression in male mice, whereas semaglutide administration reduces POMC mRNA expression in female mice. Although the effects of HFD and semaglutide on AgRP and POMC mRNA

expression were more prominent in specific sexes, similar trends were observed in both sexes (Fig. 4). Thus, positive rheostasis (HFD exposure) is associated with reduced expression of the MC3R antagonist (AgRP), whereas negative rheostasis (semaglutide administration) is associated with reduced expression of the MC3R agonist (POMC derived-$\alpha$-MSH). Based on these findings, we propose that the decreased AgRP levels observed following HFD exposure leads to overall activation (i.e. greater levels of POMC than AgRP) of secondary MC3R sites to prevent against excessive weight gain, whereas the reduced POMC expression associated with semaglutide administration leads to inhibition of secondary MC3R sites (i.e. lower levels of POMC than AgRP) to prevent against excessive weight loss. As a result, loss of MC3R may produce enhanced weight loss following semaglutide administration and increased weight gain following HFD administration. Further work is required to confirm this hypothesis and to determine the mechanism(s) mediating the sexually dimorphic effects of energy rheostatic stimuli on central melanocortin pathways.

Prior work on the role of MC3R in feeding has primarily focused on the function of MC3R in the ARC (Ghamari-Langroudi et al., 2018; Sweeney et al., 2021), whereas its role in other medial hypothalamic regions is less well understood. Here, we demonstrate that, in addition to the ARC, the DMH probably also contributes to the role of MC3R in energy rheostasis. For example, deletion of MC3R primarily localized to the DMH enhances the anorexic effects of GLP1R stimulation in male mice (Fig. 7*D*) and increases body weight gain and feeding on HFD (Fig. 8*F* and *H*). Although viral expression was sometimes observed in the nearby VMH, altered energy rheostasis was observed in mice without expression in the VMH (Figs 7*A*, *D*, 10*A* and *D*), suggesting that VMH may have a less prominent role in energy rheostasis. Instead, recent reports indicate that the function of MC3R signalling in VMH may be more specialized for glucose homeostasis (Sutton et al., 2021), at the same time as exerting a more minor role in energy homeostasis (Begriche et al., 2011). In some cases, we also observed sparse labelling in the posterior hypothalamus (Figs 7*A*, *D*, 10*A* and *D*) and we cannot completely exclude a role for this structure in energy rheostasis. Along these lines, further work is required to map the precise subregions of DMH (i.e. anterior, medial and posterior) mediating the role of DMH MC3R

signalling in energy rheostasis. However, the functional neuroanatomical experiments presented here indicate that additional medial hypothalamic structures outside of the ARC also contribute to the role of MC3R in energy rheostasis.

Consistent with an important role for DMH MC3R signalling in energy rheostasis, mRNA expression of MC3R is reduced in DMH following semaglutide administration in male mice (Fig. 5*Q–T*), and selective stimulation of DMH MC3R neurons, but not nearby VMH MC3R neurons, acutely increases energy expenditure and locomotor activity without altering food intake (Fig. 12). Conversely, deletion of MC3R in the DMH reduces locomotor activity and energy expenditure specifically in the light period, suggesting an important role for functional MC3R signalling in DMH in linking circadian rhythms with energy metabolism (Fig. 13). The function of MC3R neurons in the DMH contrasts with the established role for DMH leptin receptor and DMH GLP1R neurons in suppressing feeding (Faber et al., 2021; Kim et al., 2024; Rupp et al., 2023; Tang et al., 2023), indicating functional specificity within distinct DMH cell types involved in energy homeostasis. Indeed, RNAscope analysis indicates that the majority of DMH MC3R neurons do not express LepR, which co-expressed with GLP1R in most DMH neurons (Kim et al., 2024) (Fig. 14*G*). Similarly to DMH MC3R cells, MC3R cells in the lateral hypothalamus also promote locomotor activity via interactions with mesolimbic dopamine neurons (Pei et al., 2019). Thus, MC3R-containing cells in multiple regions of the medial hypothalamus (LH and DMH) can increase locomotion, providing one mechanism by which MC3R neurons may alter energy rheostasis. Although we did not observe acute effects on feeding following stimulation of DMH MC3R neurons, it is possible that more sustained, long-term activation of DMH MC3R neurons alters feeding behaviour or that activation of DMH MC3R neurons only regulates feeding in the context of energy rheostatic challenges (as observed in MC3R KO mice). Consistent with these possibilities, deletion of MC3R in the DMH does not acutely affect feeding in *ad libitum* fed conditions or in the first 24 h following behavioural manipulations. Instead, deletion of MC3R only alters feeding after multiple days of HFD exposure or semaglutide administration (Figs 8*D*, *H*, 9*G* and *I*). Based on these findings, we conclude that DMH also contributes to MC3R-mediated effects on energy rheostasis, primarily by modulating energy expenditure and locomotor activity. We thus propose that MC3R acts via multiplexed neural circuits including the ARC and DMH to control energy rheostasis (see graphical abstract), analogous to other receptors involved in energy homeostasis, such as the leptin receptor (Balthasar et al., 2004; Cowley et al., 2001; Xu et al., 2018) and GLP1R (Fortin et al., 2020; Gabery et al., 2020; He et al., 2019;

Kim et al., 2024; Secher et al., 2014), which mediate their effects by acting on multiple neural pathways. Further work is required to map the activity of DMH MC3R neurons in response to orexigenic and anorexic stimuli and to determine how MC3R signals in DMH neurons.

It is notable that sex differences exist with respect to the energy rheostasis phenotypes reported here in male and female MC3R-KO mice, mRNA expression of the melanocortin ligands (Fig. 4) and medial hypothalamic MC3R levels following energy rheostatic challenges (Fig. 5). For example, although we observed an enhanced anorexic response to semaglutide in male mice with viral deletion of MC3R in dMH, we did not observe an enhanced anorexic response to semaglutide in male mice with MC3R deletion throughout the MH (an effect that was observed in female mice). Although we do not currently know the reason for this difference, this may be a result of differences in the distribution of MC3R in arcuate cell types in male and female mice. For example, MC3R is expressed in both orexigenic and anorexic neurons (Ghamari-Langroudi et al., 2018; Lam et al., 2021; Sweeney et al., 2021) in the ARC, and the expression pattern of MC3R in the ARC differs between male and female animals (Bedenbaugh et al., 2022; Sweeney et al., 2021). Because MC3R is expressed in almost twice the number of cells in the ARC in male *vs.* female mice (Bedenbaugh et al., 2022; Sweeney et al., 2021), it is possible that MC3R is expressed to a greater extent in additional anorexic population in the ARC (such as arcuate POMC, vGLUT2 or BNC2 neurons) (Fenselau et al., 2017; Tan et al., 2024) in male mice, partially occluding the anorexic effect associated with MC3R deletion in DMH. Further work is required to test this hypothesis and to fully map the functional implications of sexually dimorphic MC3R expression in the ARC. Importantly, although sex differences exist in the propensity of MC3R deletion to alter energy rheostasis, MC3R deletion consistently results in increased body weight and/or feeding in response to a HFD, and enhanced weight loss and/or anorexia following semaglutide. Thus, although sex differences exist in the role of MC3R in energy rheostasis, the overall patterns of behavioural change associated with MC3R deletion are consistent with those previously reported in global MC3R KO mice (Dahir et al., 2024; Ghamari-Langroudi et al., 2018; Sweeney et al., 2021). Furthermore, global MC3R KO mice also exhibit sexually dimorphic feeding and behavioural phenotypes. For example, female MC3R KO mice (but not males) demonstrate an enhanced response to forms of stress-induced anorexia (Sweeney et al., 2021) and reduced sucrose preference that is not observed in male MC3R KO mice (Lippert et al., 2014). Conversely, male MC3R KO mice have an enhanced anorexic response to leptin, and peptide YY that is not observed in female mice (Dahir et al., 2024). Thus, the sexually dimorphic

effects associated with MC3R deletion are remarkably complex and are observed across many studies. Further work is required to determine the specific circuits and mechanism(s) mediating the sexually dimorphic effects of MC3R deletion on behaviour.

In addition to the sexually dimorphic effects of MC3R deletion on behavioural control, the neuroanatomical distribution of MC3R is drastically different in male and female mice (Bedenbaugh et al., 2022), with increased MC3R expression observed in the ARC in male animals and no difference in MC3R expression in the LH or VMH between male and female mice (Bedenbaugh et al., 2022). Conversely, female mice have higher expression of MC3R in the anteroventral-periventricular nucleus, ventral premammillary nucleus and the bed nucleus of the stria terminalis (Bedenbaugh et al., 2022). Although the total amount of MC3R containing cells and overall MC3R expression is similar in the DMH of male and female mice, the distribution of MC3R in glutamatergic neurons is different between the sexes (Fig. 15*G–M*). Here, we demonstrate that the prevalence of vGLUT2 containing cells is higher in the DMH in male mice, whereas the number of vGAT expressing cells in DMH are similar in male and female animals (Fig. 15*M*). As a result, a higher percentage of MC3R containing DMH cells express vGLUT2 in male animals (Fig. 15*L* and *M*). Futher work is required to precisely map the function of sexually dimorphic DMH MC3R circuits, as well as to determine how these sex differences contribute to the role of MC3R in energy rheostasis. However, the results presented here support the existence of sexually dimorphic MC3R circuits controlling energy rheostasis and highlight the significance of MC3R circuits outside of the ARC in controlling energy rheostasis.

### Study limitations

Although we characterized the effects of MC3R deletion in dMH on energy expenditure, we do not have energy expenditure data for all the experiments in this study, and we thus do not know the effects of MC3R deletion on energy expenditure following HFD administration or semaglutide treatment. This limitation could explain some of the discrepancies between the effects of MC3R deletion on body weight and food intake (i.e. changes in body weight without changes in food intake). Furthermore, as outlined in the associated diagrams of viral expression for each experiment, the viral spread varied on a mouse-to-mouse basis, which may have increased the variability associated with the data presented here, at the same time as resulting in a small sample size for a limited number of behavioural experiments (i.e. the low number of successful viral hits for female MH group). Currently, viral-mediated gene deletion is the only way to delete MC3R expression in adult animals in many brain regions because Cre specific inducible mouse lines do not exist for many hypothalamic brain regions (i.e. DMH). Further technological developments are required to specifically delete MC3R selectively in the DMH in adult animals in a manner that is not subject to the variability associated with viral mediated gene delivery (i.e. a transgenic mouse with a DMH specific inducible Cre-recombinase).

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

## Additional information

### Data availability statement

All original data are available from the corresponding author upon reasonable request.

### Competing interests

PS owns stock in Courage Therapeutics. All the other authors declare that they have no competing interests.

### Author contributions

All work related to this paper was performed in the Sweeney lab at the University of Illinois Urbana-Champaign. I.C.P.P. and P.S. conceived the study, designed the experiments, acquired and analysed the data and wrote and revised the text. J.B., E.P., C.N., S.C. and D.C. acquired and analysed data and contributed to revising the manuscript. All authors approved the final version of the manuscript submitted for publication, are accountable for all aspects of the work and qualify for authorship. All individuals who qualify for authorship are listed as authors.

### Funding

This work was supported by the University of Illinois, the National Institute of Health-National Institute of Diabetes and Digestive and Kidney Diseases Grant R00DK127065 (P.S.), the Foundation for Prader-Willi Research (PS) and Brain and Behaviour Research Foundation Grant 100000874 (PS).

### Acknowledgements

We thank all members of the Sweeney lab for helpful comments on earlier drafts of the manuscript.

### Keywords

energy homeostasis, feeding behaviour, melanocortins, MC3R

## Supporting information

Additional supporting information can be found online in the Supporting Information section at the end of the HTML view of the article. Supporting information files available:

**Peer Review History**

