## [Peer Review History · The Journal of Physiology]

Neuroanatomical dissection of the MC3R circuitry regulating energy rheostasis

Ingrid Camila Possa-Paranhos, Jared Butts, Emma Pyszka, Christina Nelson, Dajin Cho, Samuel Congdon, and Patrick Sweeney

DOI: 10.1113/JP286699

Corresponding author(s): Patrick Sweeney (sweenp@illinois.edu)

The following individual(s) involved in review of this submission have agreed to reveal their identity: Aaron G Roseberry (Referee #1); Jon M Resch (Referee #2)

Review Timeline:

Submission Date:	08-Apr-2024
Editorial Decision:	28-May-2024
Revision Received:	03-Sep-2024
Editorial Decision:	07-Oct-2024
Revision Received:	31-Oct-2024
Accepted:	11-Nov-2024

Senior Editor: David Wyllie

Reviewing Editor: Valentina Mosienko

Transaction Report:

Dear Dr Sweeney,

Re: JP-RP-2024-286699 "Neuroanatomical dissection of the MC3R circuitry regulating energy rheostasis" by Ingrid Camila Possa-Paranhos, Jared Butts, Emma Pyszka, Christina Nelson, Dajin Cho, and Patrick Sweeney

Thank you for submitting your manuscript to The Journal of Physiology. It has been assessed by a Reviewing Editor and by 2 expert referees and we are pleased to tell you that it is potentially acceptable for publication following satisfactory major revision.

REVISION CHECKLIST:

We look forward to receiving your revised submission.

Yours sincerely,

David Wyllie
Senior Editor
The Journal of Physiology

REQUIRED ITEMS

- Author photo and profile. First or joint first authors are asked to provide a short biography (no more than 100 words for one author or 150 words in total for joint first authors) and a portrait photograph. These should be uploaded and clearly labelled together in a Word document with the revised version of the manuscript. See Information for Authors for further details.
- The reference list must be in alphabetical order, rather than numbered, to comply with our Journal format.
- Your manuscript must include a complete Additional Information section, including competing interests; funding; author contributions and acknowledgements.
- Please upload separate high-quality figure files via the submission form.
- Papers must comply with the Statistics Policy: https://jp.msubmit.net/cgi-bin/main.plex?form_type=display_requirements#statistics.

In summary:

- If $n \leq 30$, all data points must be plotted in the figure in a way that reveals their range and distribution. A bar graph with data points overlaid, a box and whisker plot or a violin plot (preferably with data points included) are acceptable formats.
- If $n > 30$, then the entire raw dataset must be made available either as supporting information, or hosted on a not-for-profit repository, e.g. FigShare, with access details provided in the manuscript.
- 'n' clearly defined (e.g. x cells from y slices in z animals) in the Methods. Authors should be mindful of pseudoreplication.
- All relevant 'n' values must be clearly stated in the main text, figures and tables.
- The most appropriate summary statistic (e.g. mean or median and standard deviation) must be used. Standard Error of the Mean (SEM) alone is not permitted.
- Exact p values must be stated. Authors must not use 'greater than' or 'less than'. Exact p values must be stated to three significant figures even when 'no statistical significance' is claimed.

- Please include an Abstract Figure file, as well as the Figure Legend text within the main article file. The Abstract Figure is a piece of artwork designed to give readers an immediate understanding of the research and should summarise the main conclusions. If possible, the image should be easily 'readable' from left to right or top to bottom. It should show the physiological relevance of the manuscript so readers can assess the importance and content of its findings. Abstract Figures should not merely recapitulate other figures in the manuscript. Please try to keep the diagram as simple as possible and without superfluous information that may distract from the main conclusion(s). Abstract Figures must be provided by authors no later than the revised manuscript stage and should be uploaded as a separate file during online submission labelled as File Type 'Abstract Figure'. Please also ensure that you include the figure legend in the main article file. All Abstract Figures should be created using BioRender. Authors should use The Journal's premium BioRender account to export high-resolution images. Details on how to use and access the premium account are included as part of this email.

Ethics Concerns:

Please, indicate methods of euthanasia including if terminal dose of anaesthetic has been used. Section on generation of MC3R-floxed mice need to be extended including strain of mice used and exact procedures done to generate the model, and correct reference to a study which initially described the generation of this mouse line should be included.

Reviewing Editor's comments:

Your study has been reviewed by two experts in the field - both raised concerns in relation to a rather challenging approach targeting DMH using stereotaxic brain injection. Please, make sure that you address all the comments raised by the reviewers, considering increasing number of animals used in the study. Given the novelty of the mouse line used in the study, please, provide detailed information on the generation and general characterization of this mouse line in the appropriate section of the manuscript.

Comments to ensure the paper complies with the Statistics Policy:

If you submit a revised manuscript, please ensure you comply with our Statistics Policy - report SD, not SEM and exact p values unless <0.001

Senior Editor's comments:

Two expert referees and and Reviewing Editor have assessed your manuscript and made many comments regarding the data that you report. For this work to be considered further, substantial revisions, including a considerable amount of new experimental data will be required. To address all the points raised will inevitably take time and you may decide to submit this work elsewhere or transfer to Experimental Physiology for their consideration. If you do revise the work, then please be aware we do not allow Supplemental Data files and all essential data needs to be included in the main manuscript.

Referee #1:

In this manuscript, the authors test the role of MC3Rs expressed in the medial hypothalamus on their role in the response to various challenges to energy homeostasis. The authors use a floxed-MC3R mouse with AAV-Cre injections to test whether KO of MC3Rs in distinct parts of the medial hypothalamus alters the response to challenges, including semaglutide injection, acute HFD exposure and fasting/food restriction. They show that the MH MC3Rs mediate some, but not all of the effects previously seen in the whole mouse MC3R KO. They also show some apparent changes and differences in MC3R expression and which cell types express MC3Rs between sexes and following energetic challenges.

Overall, this manuscript provides new information that should be of benefit to the field. It helps to narrow down which MC3Rs may mediate some of the effects previously observed in the whole animal KO. There are some issues that should be addressed to help improve the manuscript overall, which are described below.

1. First, I want to list some strengths of the manuscript overall (since reviews are almost always focusing exclusively on the negative). The introduction was excellent. It was really great at laying out the background and rationale for the studies and how they fit into our larger understanding of the function of MC3Rs. The comparison and differences between sexes are also important and presented well.
2. The main issue with the manuscript is the big variability in the size and location of the injections. Although the data are pretty consistent and tight overall, there are a few spots where this could be an issue. In some spots, especially for the whole MH experiments, some of the injections covered a really large space, and appeared to extend into the LH and other areas as well. Specifically, for the experiments in Figure 4, there appeared to be a bit of variability in the male HFD FI data (Fig 4H), which could be related to the variety of injection sizes. Have the authors explored this and done any additional analyses to see if these relate to each other?
- 2b. Related to above, there was no discussion of the prior studies from the Olson lab showing effects of MC3Rs in the VMH and LH. The VMH results should be discussed in light of the lack of effects seen here for the VMH. And with some of the injections, the results of their LH studies might be relevant as well. So both of those should be discussed more in the manuscript. There are also other papers showing sex differences in MC3R related function that could also be included in the discussion.
3. I think the title should be altered. It isn't really descriptive of what is included in the manuscript, so I think revising it to better reflect what's in this manuscript would be beneficial. As is, it is really broad and making it more specific would be beneficial.
4. The details of the floxed-MC3R mice need to be explained better. Are these completely new mice, or have they been published before? The reference in the methods that refers to the creation of these mice does not include these mice and only used the full KO mice. If these are a new reagent, more details and characterization of these mice need to be included.
5. I think the figures could be organized a little bit better and more consistent. Figures 3 & 4 could be combined into 1 figure just like Figure 1. For Figure 1, the semaglutide and HFD withdrawal data should be plotted on the same graph as the acute data. I think this is nice and interesting data that should be in the body of the paper and not supplemental info, and it would be fairly straightforward to put it on the same graph.
6. This paper is really focused on acute effects, but I think a discussion of potential longer-term effects of MC3R KO in these areas is warranted. Some of the data suggests that longer treatments might reveal additional effects (i.e. the females on HFD in Figure 1-they look like if they go longer that a difference might become apparent). Doing longer term experiments isn't necessary for this manuscript but a discussion of this point would strengthen the paper overall.
7. The discussion is a bit too much of just a general recap of the results. I think paring this down and adding more discussion of the points mentioned above (and potentially expanding what is already discussed) would also strengthen the manuscript.
8. Were the RNAScope experiments done in both sexes? There was no discussion of the sex of the mice in these experiments. With the small #s in these experiments it will be difficult to compare sexes, but this should at least be mentioned.

8b. For the RNAScope shown in Figure 2, is panel N actually representative or the best image, because there is no apparent signal for MC3R here. With the really low numbers of cells in the DMH in this figure, I think the potential differences need to be qualified, because this could just be due to the relatively small number of mice tested and the low staining here.

-the numbers for some experiments are also a bit low-n=4 or 5 for many experiments. The data appear to be really consistent so this may not be a big deal, but increasing the total # of mice/group would help strengthen the data and the authors' arguments overall.

9. The text at the end of section 2.7 needs to be edited. It says that the data is consistent with prior data showing that semaglutide has no effect in females, but figure 1 showed effects in females, and the last sentence of the paragraph needs to be fixed as well.

10. There are a few issues with the text in some spots:

-During the results section, it would be good to mention/describe what the treatments are for each panel/data being discussed (i.e. 5 day semaglutide) so people don't have to constantly refer to the figure, which isn't quite as straightforward as it could be. This would just help the reader overall.

-affect is used in place of effect in a number of places. Affect is a verb (affected food intake) and effect is a noun (there was an effect of the treatment)-in these instances at least.

-line 225 a needs to be alpha

11. Very minor question worth thinking about or discussing: for the males in Figure 7, the total # of MC3R neurons that are vGAT and vGlut2 together is greater than 100%. Does this suggest that there are some neurons that have both vGAT and vGlut2?

Referee #2:

The manuscript by Possa-Paranhos et al. investigates the role of medial hypothalamic MC3Rs in energy rheostasis. The authors report sex differences in their model where semaglutide has an augmented effect on female KO mice and HFD causes exacerbated weight gain in KO male mice. More specific Mc3r KO in the DMH recapitulated exacerbated weight gain in males on HFD and chemogenetic activation of DMH Mc3r neurons induced increased energy expenditure and activity. Finally, the authors find a sexually dimorphic increase of Vglut2+Mc3r expressing neurons in males, which they propose as a potential mechanism for the effects observed in the KO studies.

The study investigates several important aspects of energy balance using a novel conditional Mc3r knockout approach. However, many of the results are hard to interpret based on experimental approach and/or presentation of the data in figures. Further, some effects are inconsistent from experiment to experiment, which are not addressed/explained. This drives increasing concern over the AAV-Cre approach used to knockout Mc3r.

1) While the goal of specifically knocking out Mc3r in the hypothalamus without complications of developmental compensation is understood, the approach of injecting AAV-Cre into lox-Mc3r mice also has the inherent issue that each individual mouse has a different degree of Mc3r knockout. This fact makes interpretation of the data difficult, particularly negative results and sex differences, because the specific cell types of importance in relation to Mc3r expression are still unknown. Have the authors considered also trying to use a Cre mouse strain that is mostly confined to the hypothalamus to answer some of these questions, such as an Nkx2.1-Cre line?

2) Figure 1B-E aims to demonstrate the knockout of Mc3r in the medial hypothalamus, but the comparison between 1c and 1d is not very convincing. Also, what are the units for the y-axis Mc3r mRNA in 1E (cells containing Mc3r mRNA?). If so, the WT mice average 150, which seems low for the entire medial hypothalamus.

3) The notion of sex differences after removal of Mc3r from the medial hypothalamus suggests that Mc3r+ cells should be

counted in males and females for each nucleus within the region the authors are targeting with their AAV-Cre injections. Further, an assessment of whether there are disparities in Mc3r mRNA levels in response to energy rheostatic challenges between males and females would add important additional evidence.

4) Figure 1J-M is confusing as presented because it looks like semaglutide does not reduce food intake or body weight in male mice. This is clearly not the case, but you have to compare Figure 1F-I and Figure 1J-M to make this assessment. It will help readers if the authors provide body weight and food intake time courses showing the different stages of the experiment (saline, semaglutide, and withdrawal) together in one graph easily demonstrating that semaglutide has effects on all mice, but the effect is exacerbated in female Mc3r KO mice.

5) Figure S2A is very confusing as it shows the last day of saline injections and then withdrawal days 1-5, skipping the semaglutide treatment days (or at least this is what I think it is showing). It would be clearer if the authors just presented the body weight on the same graph as mentioned in the previous comment. The representation of food intake as bar graphs showing cumulative intake over 5 day is also confusing. Why not just daily intake instead of cumulative intake over 5 days? Overall, it is far too difficult to navigate and interpret the semaglutide data as presented in Figure 1, Figure S2, and beyond.

6) How do the authors reconcile the lack of effect of semaglutide in male medial hypothalamus Mc3r KO mice, with the augmented effect of semaglutide with DMH-specific Mc3r KO?

7) The HFD tests run in several of the experiments are only for 5 days, which is a relatively short time. Alternatively, the authors knockout Mc3r in already obese mice. Have the authors provided longer access to HFD in their medial hypothalamus Mc3r knockout model to assess how prone they are to developing obesity? Would the groups still be different if given longer access to HFD, i.e, would the Mc3r mice develop obesity faster/to a greater degree? Showing that medial hypothalamus Mc3r KO causes mice to become more prone to obesity would be an important addition.

8) Given the observed reduction in AgRP mRNA levels in response to a high-fat diet (HFD), is there any direct evidence suggesting that AgRP neurons directly synapse onto DMH MC3R neurons (e.g., through monosynaptic tracing)?

9) How do the authors reconcile the effects of DMH Mc3r KO on feeding during semaglutide and/or HFD treatment with absence of feeding effects from DMH Mc3r neuron chemogenetic studies? It stands to reason that if DMH Mc3r KO affects feeding, then activating the neurons directly should also affect feeding? What about chemogenetic inhibition of DMH Mc3r neurons?

10) Figure 7 suggests that there are very few (< 50) Lepr-expressing neurons in the DMH. This is inconsistent with prior work suggesting a role for DMH lepr neurons in regulating thermogenesis (Zhang et al., J Neuro, 2011; Rezai-Zadeh et al., Mol Metab, 2014) and/or hunger (Garfield et al., Nat Neuro 2016). Further explanation is needed about which part of the DMH is being analyzed. Given the low expression of Mc3r and Lepr, it may be better to use Mc3r-Cre-mediated reporter (via AAV injection or cross to a reporter line) and inject the mice with leptin for pSTAT3 immunofluorescence. Then compare co-expression of reporter and pSTAT3.

11) If the increased amount of Vglut2+Mc3r-expressing neurons in the DMH of male mice is the proposed mechanism for observed sex differences, why weren't there more pronounced sex differences in the chemogenetic studies?

END OF COMMENTS

Reviewing Editor's comments:

Your study has been reviewed by two experts in the field - both raised concerns in relation to a rather challenging approach targeting DMH using stereotaxic brain injection. Please, make sure that you address all the comments raised by the reviewers, considering increasing number of animals used in the study. Given the novelty of the mouse line used in the study, please, provide detailed information on the generation and general characterization of this mouse line in the appropriate section of the manuscript.

Response: Thank you for the helpful comments on our manuscript. We have substantially revised the manuscript, increasing the sample size for many of the panels, and providing additional experimental data in response to reviewer comments. We have also expanded the methods section to further describe the MC3R floxed mouse line used in this study. This line has been previously used and validated in two prior publications, which are cited in the methods section of the paper.

Comments to ensure the paper complies with the Statistics Policy:

If you submit a revised manuscript, please ensure you comply with our Statistics Policy - report SD, not SEM and exact p values unless <0.001

Response: SD is reported, and exact p values have been added to all the analysis in the revised manuscript.

Senior Editor's comments:

Two expert referees and Reviewing Editor have assessed your manuscript and made many comments regarding the data that you report. For this work to be considered further, substantial revisions, including a considerable amount of new experimental data will be required. To address all the points raised will inevitably take time and you may decide to submit this work elsewhere or transfer to Experimental Physiology for their consideration. If you do revise the work, then please be aware we do not allow Supplemental Data files and all essential data needs to be included in the main manuscript.

Response: Thank you for the helpful comments on our manuscript. We have revised our manuscript according to the reviewer comments, providing substantial additional experimental data. We have also revised the figures to not included supplemental data files and have included all necessary data in the main

text figures.

Referee #1:

In this manuscript, the authors test the role of MC3Rs expressed in the medial hypothalamus on their role in the response to various challenges to energy homeostasis. The authors use a floxed-MC3R mouse with AAV-Cre injections to test whether KO of MC3Rs in distinct parts of the medial hypothalamus alters the response to challenges, including semaglutide injection, acute HFD exposure and fasting/food restriction. They show that the MH MC3Rs mediate some, but not all of the effects previously seen in the whole mouse MC3R KO. They also show some apparent changes and differences in MC3R expression and which cell types express MC3Rs between sexes and following energetic challenges.

Overall, this manuscript provides new information that should be of benefit to the field. It helps to narrow down which MC3Rs may mediate some of the effects previously observed in the whole animal KO. There are some issues that should be addressed to help improve the manuscript overall, which are described below.

1. First, I want to list some strengths of the manuscript overall (since reviews are almost always focusing exclusively on the negative). The introduction was excellent. It was really great at laying out the background and rationale for the studies and how they fit into our larger understanding of the function of MC3Rs. The comparison and differences between sexes are also important and presented well.

Response: Thank you for the supportive comments.

2. The main issue with the manuscript is the big variability in the size and location of the injections. Although the data are pretty consistent and tight overall, there are a few spots where this could be an issue. In some spots, especially for the whole MH experiments, some of the injections covered a really large space, and appeared to extend into the LH and other areas as well. Specifically, for the experiments in Figure 4, there appeared to be a bit of variability in the male HFD FI data (Fig 4H), which could be related to the variety of injection sizes. Have the authors explored this and done any additional analyses to see if these relate to each other?

Response: In the revised manuscript we now show the behavioral phenotypes of each mouse directly next to the image of the viral spread for each mouse (see figs 3, 8, and 10). In general, although there is variation in the viral spread between mice and in the magnitude of the behavioral phenotypes for some experiments, the differences in the behavioral phenotype were not obviously correlated with the size of the viral spread (i.e. the variability in feeding in the males on HFD in figure 3C was not clearly linked to the spread shown in 3A: i.e.

see mouse 12 in 3A and 3C). As mentioned in the discussion, further work is required in future studies to dissect the specific medial hypothalamic regions and cell types in these regions mediating the role of MC3R in energy rheostasis.

2b. Related to above, there was no discussion of the prior studies from the Olson lab showing effects of MC3Rs in the VMH and LH. The VMH results should be discussed in light of the lack of effects seen here for the VMH. And with some of the injections, the results of their LH studies might be relevant as well. So both of those should be discussed more in the manuscript. There are also other papers showing sex differences in MC3R related function that could also be included in the discussion.

Response: Thank you for this suggestion. We have revised the discussion section to further discuss the results from these studies. We have also expanded the discussion of previous findings related to sex differences in MC3R function.

3. I think the title should be altered. It isn't really descriptive of what is included in the manuscript, so I think revising it to better reflect what's in this manuscript would be beneficial. As is, it is really broad and making it more specific would be beneficial.

Response: Thank you for this suggestion. We have revised the title to be “Medial hypothalamic MC3R signaling regulates energy rheostasis in adult mice”.

4. The details of the floxed-MC3R mice need to be explained better. Are these completely new mice, or have they been published before? The reference in the methods that refers to the creation of these mice does not include these mice and only used the full KO mice. If these are a new reagent, more details and characterization of these mice need to be included.

Response: This mouse line has been previously published (Gui et al., Cell Reports, 2023; Dahir et al., 2024, Journal of Clinical Investigation), and the creation of the line and validation of the line is described in the initial publications describing this mouse. We have edited the methods section to include this updated information.

5. I think the figures could be organized a little bit better and more consistent. Figures 3 & 4 could be combined into 1 figure just like Figure 1. For Figure 1, the semaglutide and HFD withdrawal data should be plotted on the same graph as the acute data. I think this is nice and interesting data that should be in the body of the paper and not supplemental info, and it would be fairly straightforward to put it on the same graph.

Response: Thank you for these suggestions. We have updated the text figures according to your suggestions.

6. This paper is really focused on acute effects, but I think a discussion of potential longer-term effects of MC3R KO in these areas is warranted. Some of the data

suggests that longer treatments might reveal additional effects (i.e. the females on HFD in Figure 1-they look like if they go longer that a difference might become apparent). Doing longer term experiments isn't necessary for this manuscript but a discussion of this point would strengthen the paper overall.

Response: In the revised version of the manuscript we have repeated the HFD experiment, providing one month of daily feeding and body weight change in WT and MH MC3R KO mice on HFD. We have also increased the sample size of the female mice in this experiment. Our new data show that both male and female MC3R MH KO mice consume more food in the first five days of HFD. Further, male mice continue to gain more weight and consume more food than WT mice in the subsequent month on HFD. However, we did not detect a significant effect of HFD on weight gain over one month in female MH MC3R KO mice vs female WT mice. Given the major variation in the amount of weight gain on HFD observed in female WT mice, this may have obscured our ability to detect an effect in female MH MC3R KO mice on HFD weight gain.

7. The discussion is a bit too much of just a general recap of the results. I think paring this down and adding more discussion of the points mentioned above (and potentially expanding what is already discussed) would also strengthen the manuscript.

Response: Thank you for this suggestion. We have revised the discussion according to your suggestions.

8. Were the RNAScope experiments done in both sexes? There was no discussion of the sex of the mice in these experiments. With the small #s in these experiments it will be difficult to compare sexes, but this should at least be mentioned.

Response: In the original version of the manuscript, the RNAScope characterization of the response to energy rheostasis was only done in male mice. In the revised manuscript we repeated these experiments in female mice.

8b. For the RNAScope shown in Figure 2, is panel N actually representative or the best image, because there is no apparent signal for MC3R here. With the really low numbers of cells in the DMH in this figure, I think the potential differences need to be qualified, because this could just be due to the relatively small number of mice tested and the low staining here.

Response: The MC3R mRNA signal is relatively low in DMH, but these images are representative of the amount of signal typically observed in DMH in each of these conditions, showing the significantly lower signal intensity observed after semaglutide administration (Previously Figure 2N, now Figure 5R) compared to mice with ad libitum access to regular chow (Figure 5Q) or high fat diet (Figure 5S).

-the numbers for some experiments are also a bit low-n=4 or 5 for many experiments. The data appear to be really consistent so this may not be a big deal, but increasing the total # of mice/group would help strengthen the data and the authors' arguments overall.

Response: In the revised manuscript we have increased the sample size for many of the experimental comparisons (i.e. food intake and body weight response of WT and MH MC3R KO mice to both regular chow and HFD administration, and characterizing the RNAscope response to energy rheostasis challenges in both male and female mice).

9. The text at the end of section 2.7 needs to be edited. It says that the data is consistent with prior data showing that semaglutide has no effect in females, but figure 1 showed effects in females, and the last sentence of the paragraph needs to be fixed as well.

Response: This has been corrected in the revised manuscript.

10. There are a few issues with the text in some spots:

-During the results section, it would be good to mention/describe what the treatments are for each panel/data being discussed (i.e. 5 day semaglutide) so people don't have to constantly refer to the figure, which isn't quite as straightforward as it could be. This would just help the reader overall.

Response: Thank you for this suggestion, changes were made to make it more clear in the text.

-affect is used in place of effect in a number of places. Affect is a verb (affected food intake) and effect is a noun (there was an effect of the treatment)-in these instances at least.

Response: corrected

-line 225 a needs to be alpha

Response: corrected

11. Very minor question worth thinking about or discussing: for the males in Figure 7, the total # of MC3R neurons that are vGAT and vGlut2 together is greater than 100%. Does this suggest that there are some neurons that have both vGAT and vGlut2?

Response: That is an interesting observation. This is possible since we did not directly compare vGAT and vGLUT2 in the same sections. Due to this experimental limitation, we cannot exclude the possibility that some cells could be expressing both VGAT and VGlut2 mRNA, as seen in other cell types that

release both GABA and glutamate (Root et al., *Cell Rep*, 2014). However, since the average of both the percentage of MC3R cells that co-express VGAT or VGlut2 in males are close values around 50% (58.41% and 50.61%, respectively), the values being above 100% total could be just because these were not performed within the exact same sections (ie; one section had the MC3R and VGAT probes, another section had the MC3R and VGlut2 probes).

Referee #2:

The manuscript by Possa-Paranhos et al. investigates the role of medial hypothalamic MC3Rs in energy rheostasis. The authors report sex differences in their model where semaglutide has an augmented effect on female KO mice and HFD causes exacerbated weight gain in KO male mice. More specific Mc3r KO in the DMH recapitulated exacerbated weight gain in males on HFD and chemogenetic activation of DMH Mc3r neurons induced increased energy expenditure and activity. Finally, the authors find a sexually dimorphic increase of Vglut2+Mc3r expressing neurons in males, which they propose as a potential mechanism for the effects observed in the KO studies.

The study investigates several important aspects of energy balance using a novel conditional Mc3r knockout approach. However, many of the results are hard to interpret based on experimental approach and/or presentation of the data in figures. Further, some effects are inconsistent from experiment to experiment, which are not addressed/explained. This drives increasing concern over the AAV-Cre approach used to knockout Mc3r.

1) While the goal of specifically knocking out Mc3r in the hypothalamus without complications of developmental compensation is understood, the approach of injecting AAV-Cre into lox-Mc3r mice also has the inherent issue that each individual mouse has a different degree of Mc3r knockout. This fact makes interpretation of the data difficult, particularly negative results and sex differences, because the specific cell types of importance in relation to Mc3r expression are still unknown. Have the authors considered also trying to use a Cre mouse strain that is mostly confined to the hypothalamus to answer some of these questions, such as an Nkx2.1-Cre line?

Response: Thank you for these suggestions. We agree that there is inherent variation in the magnitude of the viral expression and deletion on a mouse to-mouse basis. However, since the overall focus of this manuscript was to determine if the effects of MC3R deletion on energy rheostasis are regulated by MC3R signaling in adult animals and to begin to map the relevant brain regions involved, we believe that viral deletion approaches are the most reasonable strategy to address this question. This rationale was based on similar approaches which were utilized to map the role of MC4R in adult animals and the relevant brain regions that are involved in MC4R mediated effects on feeding. We

acknowledge, however, that viral deletion approaches are not without limitations but believe that these limitations (i.e. mouse-by-mouse variation in viral targeting, etc) are associated with all viral injection approaches. In the revised manuscript we aim to more transparently show the link between viral expression and the behavioral phenotypes by showing the viral expression and corresponding behavioral phenotypes for each mouse used in the manuscript (i.e. new figures 3, 8, and 10) so that the reader can directly observe the relationship between viral spread and behavioral changes. Although there is some mouse-to-mouse variation in viral spread, this approach demonstrates that MC3R regulates energy rheostasis in adult mice (i.e. the role of MC3R in energy rheostasis is not due to secondary effects associated with global MC3R deletion), and that the medial hypothalamus is one brain region in which MC3R signals in adults to regulate energy rheostasis.

It is true that our approach does not provide information on the relevant cell types mediating the effects of MC3R deletion on behavior. However, we want to note that the MC3R system is remarkably complicated. For example, at least a dozen different arcuate cell types contain MC3R (Sweeney et al., 2021; Lam et al., 2021), and it is likely that many distinct cell types also contain MC3R in VMH and DMH. Thus, we believe that determining the relevant cell types involved in energy rheostasis is well beyond the scope of one paper, and it will take years of work from multiple groups to map the function of MC3R in individual cell types within individual brain regions.

Thank you for the suggestion of using the Nkx2.1 mouse line to limit deletion to mostly hypothalamus. We agree that this mouse line could be helpful for mostly limiting the deletion to hypothalamus. However, Nkx2.1 is expressed in multiple parts of hypothalamus (Murcia-Ramón et al., *Brain Structure and Function*, 2020), multiple cell types in hypothalamus (Yee et al., *J Comp Neuro*, 2015), and in additional extra-hypothalamic sites such as cortex and amygdala (which also contain MC3R, Bedendaugh et al., *Journal of Comparative Biology*, 2022). Thus, this line will not provide the information concerning the important cell types involved. As previously mentioned, solving the involved cell types will require deleting MC3R from neurochemically distinct cell types via Cre-loxp recombination and/or CRISPR-Cas9 approaches, which will likely require many years of work and is beyond the scope of this paper.

2) Figure 1B-E aims to demonstrate the knockout of Mc3r in the medial hypothalamus, but the comparison between 1c and 1d is not very convincing. Also, what are the units for the y-axis Mc3r mRNA in 1E (cells containing Mc3r mRNA?). If so, the WT mice average 150, which seems low for the entire medial hypothalamus.

Response: We have replaced the original image from 1c and 1d with improved high-resolution images that more clearly show the difference in MC3R mRNA

expression between WT and MH MC3R KO animals. The original quantification of these images was performed by counting the cells in a specific subregion of hypothalamus and the same region of interest with viral expression in WT and KO animals . In the revised figure 1 we now count the total number of MC3R cells detected in the entire medial hypothalamus. This was done by counting the cells that contained MC3R mRNA expression for all the regions in the medial hypothalamus (DMH, VMH, and Arc) and comparing the WT to the KO mice.

3) The notion of sex differences after removal of Mc3r from the medial hypothalamus suggests that Mc3r+ cells should be counted in males and females for each nucleus within the region the authors are targeting with their AAV-Cre injections. Further, an assessment of whether there are disparities in Mc3r mRNA levels in response to energy rheostatic challenges between males and females would add important additional evidence.

Response: Thank you for this important suggestion. Prior studies (Sweeney et al., 2021; Bedenbaugh et al., 2022) have determined that MC3R mRNA expression and the number of MC3R positive cells is higher in males (approximately 2-fold higher) than females in the arcuate nucleus. Prior reports also indicate no significant difference in the number of MC3R cells in VMH between males and females (Bedenbaugh et al., 2022). These prior findings are cited multiple times in the results and discussion section of the revised manuscript. Here, we also determined that the number of MC3R containing cells and the amount of MC3R mRNA expression was not different in the DMH between males and females (Fig. 14 and 15; although the distribution of MC3R in vGAT vs vGLUT2 cells was different).

We agree with your important suggestion that we should perform the mRNA quantification of AgRP, POMC, and MC3R in both males and females following energy rheostatic challenges (the original manuscript only contained data from male mice for this experiment; i.e. original figure 2). In the revised manuscript we have repeated all of these experiments in female mice (new figure 4 and 5), allowing for a direct comparison of the effects of energy rheostasis challenges on AgRP, POMC, and MC3R in male and female mice. Interesting, we also find sex differences in the effects of energy rheostatic challenges on the mRNA levels of MC3R, AgRP, and POMC, further highlighting the sexually dimorphic nature of this system.

4) Figure 1J-M is confusing as presented because it looks like semaglutide does not reduce food intake or body weight in male mice. This is clearly not the case, but you have to compare Figure 1F-I and Figure 1J-M to make this assessment. It will help readers if the authors provide body weight and food intake time courses showing the different stages of the experiment (saline, semaglutide, and withdrawal) together in one

graph easily demonstrating that semaglutide has effects on all mice, but the effect is exacerbated in female Mc3r KO mice.

Response: Thank you for this suggestion. In the revised manuscript, we have updated the figures to include the saline treatment, semaglutide treatment, and withdrawal periods together in the same graph.

5) Figure S2A is very confusing as it shows the last day of saline injections and then withdrawal days 1-5, skipping the semaglutide treatment days (or at least this is what I think it is showing). It would be clearer if the authors just presented the body weight on the same graph as mentioned in the previous comment. The representation of food intake as bar graphs showing cumulative intake over 5 day is also confusing. Why not just daily intake instead of cumulative intake over 5 days? Overall, it is far too difficult to navigate and interpret the semaglutide data as presented in Figure 1, Figure S2, and beyond.

Response: Thank you for this suggestion. As described in comment 4 above, this data has been revised to include the semaglutide and WD feeding and BW graphs on the same graphs. As requested, we have also replotted all the food intake data to show daily food intake instead of cumulative food intake.

6) How do the authors reconcile the lack of effect of semaglutide in male medial hypothalamus Mc3r KO mice, with the augmented effect of semaglutide with DMH-specific Mc3r KO?

Response: Although we do not currently know the reason for this difference it is important to note here that the sexually dimorphic anatomy and function of MC3R is remarkably complicated, with multiple potential reasons for this difference. For example, MC3R is expressed in both orexigenic and anorexic neurons (Ghamari-Langroudi et al., Science Advances 2018; Sweeney et al., Science Translational Medicine., 2021; Lam et al., Nature 2022) in the arcuate nucleus of hypothalamus, and this expression pattern differs between male and female animals (Sweeney et al., 2021; Bedenbaugh et al., 2022). Since MC3R is expressed in nearly twice the number of cells in the arcuate nucleus in male vs female mice (Sweeney et al., 2021; Bedenbaugh et al., 2022), one possibility is that in male mice MC3R is expressed to a greater extent in additional anorexic population in the arcuate nucleus than in female mice (such as arcuate POMC, vGLUT2, or BNC2 neurons). As a result, in male mice deletion of MC3R in the arcuate nucleus may produce an orexigenic effect that occludes the anorexic effect of MC3R deletion in DMH. We have expanded the discussion to outline potential reasons for the sex differences in this manuscript (see lines 586-621 in the discussion), and to note that further work is required to solve this question.

7) The HFD tests run in several of the experiments are only for 5 days, which is a

relatively short time. Alternatively, the authors knockout Mc3r in already obese mice. Have the authors provided longer access to HFD in their medial hypothalamus Mc3r knockout model to assess how prone they are to developing obesity? Would the groups still be different if given longer access to HFD, i.e, would the Mc3r mice develop obesity faster/to a greater degree? Showing that medial hypothalamus Mc3r KO causes mice to become more prone to obesity would be an important addition.

Response: Thank you for this important suggestion. We agree that this is an important experiment to perform. Therefore, in the revised version of the manuscript we have repeated the HFD experiment in MC3R MH KO mice and WT littermate control animals, providing one month of daily feeding and body weight change in WT and MH MC3R KO mice on HFD (new figure 3I-L). Our new data show that both male and female MC3R MH KO mice consume more food in the first five days of HFD, despite consuming the same amount of food over the same period when fed a regular chow diet. Further, male mice continue to gain more weight and consume more food than WT mice in the subsequent month on HFD. Female MH MC3R KO mice also consumed more HFD during one month of HFD exposure than WT animals. However, we did not detect a significant effect of HFD on weight gain over one month in female MH MC3R KO mice vs female WT mice, although a trend towards increased body weight gain was observed. Given the major variation in the amount of weight gain on HFD observed in female WT mice, this may have obscured our ability to detect an effect in female MH MC3R KO mice on HFD weight gain.

8) Given the observed reduction in AgRP mRNA levels in response to a high-fat diet (HFD), is there any direct evidence suggesting that AgRP neurons directly synapse onto DMH MC3R neurons (e.g., through monosynaptic tracing)?

Response: We do not have any direct evidence that AgRP neurons directly synapse onto DMH MC3R neurons. We want to note that although rabies tracing could help to determine if AgRP neurons are synaptically connected to DMH MC3R neurons, since the melanocortin receptors signal via peptidergic transmission (i.e. alpha-MSH and AgRP action on MC3R), rabies tracing would not provide information concerning if there are functional effects of AgRP peptide (or alpha-MSH peptide) on DMH MC3R neurons. In the revised manuscript we provide additional immunohistochemical evidence indicating strong AgRP and POMC peptide expression in the DMH, with many instances of overlap between AgRP/POMC peptides and DMH MC3R neurons (see new figure 11). Although this data does not prove functional connections, it does demonstrate that melanocortin peptides are present near DMH MC3R cells, indicating that functional signaling is at least anatomically possible. As noted in the discussion, further work is required to determine if functional connections exist between AgRP/POMC neurons and DMH MC3R neurons, and to determine how melanocortins signal in these cells. These studies will require the development of

electrophysiological approaches in our lab, which will require multiple years of additional work and are beyond the scope of this manuscript.

9) How do the authors reconcile the effects of DMH Mc3r KO on feeding during semaglutide and/or HFD treatment with absence of feeding effects from DMH Mc3r neuron chemogenetic studies? It stands to reason that if DMH Mc3r KO affects feeding, then activating the neurons directly should also affect feeding? What about chemogenetic inhibition of DMH Mc3r neurons?

Response: As outlined in the discussion (i.e. lines 570-579), we believe there are a few different possibilities for this difference. Firstly, our data indicate that acute chemogenetic activation of DMH MC3R neurons does not alter feeding. It is, however, possible that long term activation of these cells may alter feeding behavior. This possibility is consistent with our data showing that multiple days of HFD or semaglutide treatment is required for effects of MC3R deletion on feeding to occur. Secondly, given that deletion of MC3R only alters feeding in certain conditions (i.e following semaglutide or HFD administration), and does not alter feeding under *ad libitum* regular chow feed conditions, it is possible that stimulation of DMH MC3R neurons only alters feeding under certain conditions (such as in conjunction with semaglutide or after multiple days of HFD). Finally, this difference may be due to technical limitations associated with the chemogenetic stimulation experiments used in this paper. Since the goal of the chemogenetic approaches in this manuscript was to localize specific MH MC3R populations involved in energy homeostasis, we targeted a small volume of virus (i.e. 30-50nl) unilaterally to limit viral spread to one specific region of MH. While this approach is useful for pinpointing the role of VMH MC3R neurons vs DMH MC3R neurons, it only allows for the targeting and stimulation of a small population of neurons. Thus, it is possible that a larger number of targeted cells are required to produce acute effects on feeding.

We did not utilize chemogenetic inhibition experiments in this manuscript for a few different reasons. Since we do not know how MC3R signals in hypothalamic neurons, it is difficult to interpret how chemogenetic inhibition approaches would relate to functional MC3R signaling in these cells. For example, MC4R has been reported to signal via many different signaling pathways (i.e. Gs, Gq, non-G protein dependent pathways) and has either excitatory or inhibitory effects on different populations of neurons. A similarly complex signaling mechanism may also exist for MC3R, although there is very little data concerning how MC3R signals in vivo. We elected to use chemogenetic stimulation approaches here as a strategy to pinpoint specific populations of MC3R neurons in MH which are capable of altering energy homeostasis since this approach only requires unilateral injection and small viral injection volumes which can be localized to individual regions. Further, since all our viral deletion approaches are loss of function strategies, we wanted to include a gain of

function approach (i.e. chemogenetic stimulation). In the revised manuscript we further probe the effects of MC3R deletion in DMH (loss of function studies) on energy homeostasis by performing metabolic profiling on a new cohort of DMH MC3R KO mice and littermate controls (i.e. energy expenditure and locomotion in metabolic cages; new figure 13). In direct contrast to activation of DMH MC3R neurons, we find that deletion of MC3R in DMH reduces energy expenditure and locomotion.

10) Figure 7 suggests that there are very few (< 50) LepR-expressing neurons in the DMH. This is inconsistent with prior work suggesting a role for DMH lepr neurons in regulating thermogenesis (Zhang et al., J Neuro, 2011; Rezai-Zadeh et al., Mol Metab, 2014) and/or hunger (Garfield et al., Nat Neuro 2016). Further explanation is needed about which part of the DMH is being analyzed. Given the low expression of Mc3r and LepR, it may be better to use Mc3r-Cre-mediated reporter (via AAV injection or cross to a reporter line) and inject the mice with leptin for pSTAT3 immunofluorescence. Then compare co-expression of reporter and pSTAT3.

Response: We apologize for not making this figure clearer in the previous version of the manuscript. In the prior version of this paper we analyzed the average number of cells in a single 40um section containing DMH (rather than the sum of all the DMH containing sections; hence the lower cell numbers compared to the literature cited, in which they typically sum the number of cells across the entire DMH). Additionally, we only quantified one side of DMH, while the other papers mentioned summed the number of cells on both sides. In the revised manuscript we have added Allen Brain Atlas diagrams to the side of each of the RNAscope images to show where the example image is taken from. Further, we have quantified the number of cells throughout most of the DMH (bregma -1.67 to -1.91), and the values of the different areas of DMH are available in Figure 14. These new analysis provided a similar number of DMH LepR cells as previous studies which used RNAscope to quantify LepR expression in DMH (Tang et al., *Sci Adv*, 2023 and Kyu Sik Kim et al. *Science*, 2024).

We also want to note that most of the papers you mentioned which previously quantified LepR cells in DMH utilized LepR-GFP or LepR-cre lines crossed to reporters. These mouse lines will label every cell that contains LepR at any point in development, and thus may be expected to label more cells than RNAscope, which only labels adult and time specific expression of LepR. Thus, our RNAscope analysis likely excluded some LepR cells that would be included with transgenic reporter lines.

11) If the increased amount of Vglut2+Mc3r-expressing neurons in the DMH of male mice is the proposed mechanism for observed sex differences, why weren't there more pronounced sex differences in the chemogenetic studies?

Response: We did not intend to propose that this neuroanatomical difference is the mechanism for the observed behavioral difference associated with deletion of MC3R in DMH and have made this clearer in our revised manuscript (see lines 613-621 in discussion). We apologize if this seemed to be the case in the previous version of the manuscript and have updated the text accordingly. Instead, this data indicates that the distribution of MC3R in DMH is different in male and female mice, although the behavioral significance of this difference is unknown.

There are multiple possibilities why we do not see more pronounced sex differences in chemogenetic assays. Firstly, similar to our previous comments, sex differences may emerge with long-term chemogenetic stimulation and/or with chemogenetic stimulation in the context of energy rheostatic challenges (i.e. semalgutide or HFD administration). Further, the chemogenetic stimulation experiments used here target all DMH MC3R cell types, and do not specifically target the vGLUT2 subset that is more highly expressed in males. Thus, to dissect potential sex differences between Vglut2+/MC3R+ DMH neurons it may be necessary to specifically manipulate this subset of DMH cells in males and females. In future studies, we may utilize double transgenic mice (i.e. MC3R-Cre; Vglut2-flp) and dual recombinase contingent virus' (i.e. Cre-on; Flp-on) to selectively manipulate this subclass of neurons. However, these studies will require substantial expansion of our mouse colony and the development of new viral vector tools which we believe are beyond the scope of this current paper.

Dear Dr Sweeney,

Re: JP-RP-2024-286699R1 "Neuroanatomical dissection of the MC3R circuitry regulating energy rheostasis" by Ingrid Camila Possa-Paranhos, Jared Butts, Emma Pyszka, Christina Nelson, Dajin Cho, Samuel Congdon, and Patrick Sweeney

Thank you for submitting your manuscript to The Journal of Physiology. It has been assessed by a Reviewing Editor and by 2 expert referees and we are pleased to tell you that it is acceptable for publication following satisfactory revision.

REVISION CHECKLIST:

We look forward to receiving your revised submission.

Yours sincerely,

David Wyllie
Senior Editor
The Journal of Physiology

REQUIRED ITEMS

- Papers must comply with the Statistics Policy: https://jp.msubmit.net/cgi-bin/main.plex?form_type=display_requirements#statistics.

In summary:

- If $n \leq 30$, all data points must be plotted in the figure in a way that reveals their range and distribution. A bar graph with data points overlaid, a box and whisker plot or a violin plot (preferably with data points included) are acceptable formats.
 - If $n > 30$, then the entire raw dataset must be made available either as supporting information, or hosted on a not-for-profit repository, e.g. FigShare, with access details provided in the manuscript.
 - 'n' clearly defined (e.g. x cells from y slices in z animals) in the Methods. Authors should be mindful of pseudoreplication.
 - All relevant 'n' values must be clearly stated in the main text, figures and tables.
 - The most appropriate summary statistic (e.g. mean or median and standard deviation) must be used. Standard Error of the Mean (SEM) alone is not permitted.
 - Exact p values must be stated. Authors must not use 'greater than' or 'less than'. Exact p values must be stated to three significant figures even when 'no statistical significance' is claimed.
-

Reviewing Editor's comments:

The referees considered the revisions to be mainly satisfactory. One of the reviewers still raises concerns regarding the

used experimental approach and its limitations. In the revised version of the manuscript, please, include a throughout discussion on the limitations of the used approach and make sure that statistical analysis is appropriate for the data. Please, make sure that all figures are cited throughout the manuscript.

Senior Editor's comments:

Thank you for the significant revisions to your manuscript following the first round of reviews. It is acknowledged that you have addressed many of these but some concerns remain. These are highlighted in Referee 1's report which details these and suggests further revisions, some of which can be addressed with text edits. Notwithstanding, both referees note the potential contribution of this work to the field and as such I invite you to further revise your manuscript to allow clarification of the points raised.

Referee #1:

In this revised manuscript, the authors have addressed a number of the concerns with the prior submission. They have added some additional new data, and have modified the graphs to show all individual data points throughout the entire paper. There are still some outstanding issues present in the manuscript, however, and the addition of the new data has also raised some additional new concerns. Unfortunately, the concerns with the fidelity of the AAV-Cre injection approach still severely weaken the manuscript overall. Specific comments are listed below.

1. The AAV injection approach is still problematic. I sympathize with the authors on this issue, as it is the best approach to address this question, but it raises significant concern. The huge variability in the size and location of the injections, with some mice showing really big injection areas vs others showing very small areas and some being entirely unilateral raises a lot of concerns. One particularly concerning issue here is that there are some individual mice with relatively little injections in the dmH area tested in later figures, but these mice still show the same phenotype. Combined with the small sample sizes and variability in some of the figures, it just raises a lot of concern about the overall interpretation of the data.

2. There are some questions with the statistical methods used. In a number of figures, it was stated that a 2-way ANOVA with multiple comparisons was used, but it's a little unclear what this means. Is this a repeated measures 2-way ANOVA (which appears to be the appropriate analysis) or a simple 2-way ANOVA that is corrected for multiple comparisons (which wouldn't be appropriate)? There is also no mention of whether the main effects are significant vs significant interactions, which would be required for post-hoc comparisons, and no description of what post-hoc analyses were used. A detailed description of all statistical methods for each analysis should be included, and all of the relevant data should be provided (F values, df, etc).

2a. Related to above, for some of the panels, it is stated that there is a significant difference, which appears to mean that there is a significant main effect of WT vs KO, yet there are no individual points that are actually significantly different. Is this actually physiologically relevant then? Clarifying the stats as described above might help with this point.

3. The sample size for some groups is still quite low, with some being an n=4 or 5. These experiments are just not powered strongly enough to draw any conclusions from the data, so appropriate sample sizes are needed to be able to make accurate conclusions from the data. I understand the difficulty of these experiments and how challenging they all are, but without sufficient sample sizes, accurate conclusions just can't be drawn from the data.

4. The data overall don't appear to be quite consistent across the experiments, and this isn't really explained well. For example, in Figure 7, female mice lost more weight with semaglutide (KO) but had no change in food intake, while males had no change in weight, but differences in FI? What does this mean. Similarly the male data in Figure 7 doesn't correlate with the data from Figure 2, and there seems to be a few other inconsistencies like this that aren't really explained.
5. The IHC data is a bit concerning. Were appropriate controls done to ensure that this is authentic specific staining? This is a concern because there is little to no staining for AgRP or POMC cell bodies in the Arc at all in these images. It is difficult to stain AgRP in cell bodies, but anti-POMC antibodies robustly label POMC cell bodies in the Arc, so it's a big concern that these don't show up in these figures.
6. I think the title of Section 2.5 should be edited to reflect the data a bit better. It is true for part of what is discussed in that section, but the fact that DMH KO doesn't affect regular chow or fasting/food-restriction and refeeding contradicts what the title says.
7. Figures 3 and 8 (and maybe another?) are never actually referred to in the text of the manuscript. I understand why these are included, but they should at least be referred to in the text.
8. The presentation of the panels in Figure 4 appears incomplete. The top panels don't show all treatment groups (although this may just be an issue of mislabeling a panel), but even if it is mislabeled, there are no panels showing fasting in the females, though it is mentioned in the text. It's also unclear why both POMC and AgRP are shown in the same images for females but not males. It might make sense to separate the female data to make them consistent.
9. The sex of the animals used in the DREADD experiments isn't listed anywhere, and the use of only 1 sex in these experiments should be noted and discussed.
10. The source of the relevant drugs used in the experiments should also be provided (i.e. semaglutide).

Referee #2:

The authors have adequately addressed all comments/concerns.

END OF COMMENTS

Referee #1:

In this revised manuscript, the authors have addressed a number of the concerns with the prior submission. They have added some additional new data, and have modified the graphs to show all individual data points throughout the entire paper. There are still some outstanding issues present in the manuscript, however, and the addition of the new data has also raised some additional new concerns. Unfortunately, the concerns with the fidelity of the AAV-Cre injection approach still severely weaken the manuscript overall. Specific comments are listed below.

1. The AAV injection approach is still problematic. I sympathize with the authors on this issue, as it is the best approach to address this question, but it raises significant concern. The huge variability in the size and location of the injections, with some mice showing really big injection areas vs others showing very small areas and some being entirely unilateral raises a lot of concerns. One particularly concerning issue here is that there are some individual mice with relatively little injections in the dMH area tested in later figures, but these mice still show the same phenotype. Combined with the small sample sizes and variability in some of the figures, it just raises a lot of concern about the overall interpretation of the data.

Response: We appreciate your concern with the variability in the viral targeting approaches. However, we would suggest that the presence of phenotypic differences in mice with limited viral deletion of MC3R serves to strengthen the argument that MC3R expression in medial hypothalamus is important for energy rheostasis, since deletion of MC3R from only a small number of cells in MH in adult mice is sufficient to produce differences in feeding behavior and/or body weight. It is true that the mouse-to-mouse variation is a limitation of the study, which is why we show the viral spread for every mouse and have added a separate limitations section at the conclusion of the discussion section in the revised manuscript to outline the caveats associated with the viral deletion approach utilized in this study.

2. There are some questions with the statistical methods used. In a number of figures, it was stated that a 2-way ANOVA with multiple comparisons was used, but it's a little unclear what this means. Is this a repeated measures 2-way ANOVA (which appears to be the appropriate analysis) or a simple 2-way ANOVA that is corrected for multiple comparisons (which wouldn't be appropriate)? There is also no mention of whether the main effects are significant vs significant interactions, which would be required for post-hoc comparisons, and no description of what post-hoc analyses were used. A detailed description of all statistical methods for each analysis should be included, and all of the relevant data should be provided (F values, df, etc).

Response: In this updated version, we added the detailed statistical methods used in the figure legends for each figure. The 2-way ANOVA analysis was

repeated measurements for all the daily food intake and daily body weight measurements. The details of the F values were added to the figure legends of each figure and the post-hoc analysis for the main effect and/or interactions is now described in the methodology in “Data analysis and Statistics” and in the figure legends.

2a. Related to above, for some of the panels, it is stated that there is a significant difference, which appears to mean that there is a significant main effect of WT vs KO, yet there are no individual points that are actually significantly different. Is this actually physiologically relevant then? Clarifying the stats as described above might help with this point.

Response: Yes, we meant that the main effect of the KO is significantly different from the WT. This has been made clearer in the updated text and figure legends where we now outline all relevant statistical information.

3. The sample size for some groups is still quite low, with some being an n=4 or 5. These experiments are just not powered strongly enough to draw any conclusions from the data, so appropriate sample sizes are needed to be able to make accurate conclusions from the data. I understand the difficulty of these experiments and how challenging they all are, but without sufficient sample sizes, accurate conclusions just can't be drawn from the data.

Response: The sample sizes of 4-5 mice per group are primarily for the RNAscope experiments in the paper. This is common for many papers in the field that utilize RNAscope to characterize the phenotypic identify of cells (see Chen et al., Cell Metabolism, 2023; Biglari et al., Nature Neuroscience, 2021; and Kosse et al., Nature, 2024 as a few examples with similar analysis approaches for RNAscope analysis). For the *in vivo* behavioral experiments, the sample size is 4 mice for only a few panels in the paper (i.e. female MH MC3R KO group). For all other behavioral experiments, the sample sizes are between 6 and 12 mice/group, which is consistent with most related studies in the field. We have further discussed the low sample sizes for some of the panels as a limitation of the study in the new limitations section of the manuscript.

4. The data overall don't appear to be quite consistent across the experiments, and this isn't really explained well. For example, in Figure 7, female mice lost more weight with semaglutide (KO) but had no change in food intake, while males had no change in weight, but differences in FI? What does this mean. Similarly the male data in Figure 7 doesn't correlate with the data from Figure 2, and there seems to be a few other inconsistencies like this that aren't really explained.

Response: In figure 7, since the female mice lost more weight without eating less, this suggests that semaglutide produces a greater increase in energy expenditure

in MC3R dMH KO mice than WT animals. In the male's, deletion of MC3R in dMH enhances the anorexic effect to semaglutide, but the mice do not lose more weight. It is possible that semaglutide does not increase energy expenditure in male MC3R dMH KO mice as efficiently as it does in WT animals, which could explain this difference. Unfortunately, we do not have energy expenditure data in this assay so we do not know the effect of semaglutide on energy expenditure in dMH KO mice, which may also be different between male and female mice. This limitation is further discussed in the new limitations section of the manuscript.

Figure 2 presents data from deletion of MC3R across the MH, while figure 7 only presents data from the dorsal part of MH. Therefore, the difference in the targeted cells could explain the differences in the effect of MC3R deletion on feeding and BW responses to HFD and semaglutide in figures 2 and 7. The possible reasons for the differences between the male MH and dMH data is further described in the discussion section (lines 594-609)

5. The IHC data is a bit concerning. Were appropriate controls done to ensure that this is authentic specific staining? This is a concern because there is little to no staining for AgRP or POMC cell bodies in the Arc at all in these images. It is difficult to stain AgRP in cell bodies, but anti-POMC antibodies robustly label POMC cell bodies in the Arc, so it's a big concern that these don't show up in these figures.

Response: Thank you for pointing this out. There was a mistake on the labelling of the arcuate example images that POMC and AgRP images were switched. You can see that the cell bodies are present on the now new panel 11I for POMC (which was previously labelled incorrectly as AgRP), and as you mentioned it is hard to see cell bodies with AgRP IHC, which is what is seen in the new panel 11H (which was previously labelled as POMC).

6. I think the title of Section 2.5 should be edited to reflect the data a bit better. It is true for part of what is discussed in that section, but the fact that dMH KO doesn't affect regular chow or fasting/food-restriction and refeeding contradicts what the title says.

Response: Thank you for this suggestion. We have revised section 2.5 to indicate that the increased weight gain on HFD (positive energy rheostasis) following MC3R deletion does not require the arcuate nucleus.

7. Figures 3 and 8 (and maybe another?) are never actually referred to in the text of the manuscript. I understand why these are included, but they should at least be referred to in the text.

Response: Thank you for this suggestion. In the revised results, Figure 3 is cited on line 180 and figure 8 is cited on lines 317, 320, and 321.

8. The presentation of the panels in Figure 4 appears incomplete. The top panels don't show all treatment groups (although this may just be an issue of mislabeling a panel), but even if it is mislabeled, there are no panels showing fasting in the females, though it is mentioned in the text. It's also unclear why both POMC and AgRP are shown in the same images for females but not males. It might make sense to separate the female data to make them consistent.

Response: Thank you for this suggestion. We have updated the figure to make the AgRP and POMC example images the same for both male and females and have updated the labeling in the revised figure 4.

The goal for the fasting experiment was to test if differences in transcription could be observed using the mean intensity parameters, since fasting is known to increase AgRP mRNA. Therefore, we only performed this experiment once on male mice, mainly as a positive control for the technique we were using.

9. The sex of the animals used in the DREADD experiments isn't listed anywhere, and the use of only 1 sex in these experiments should be noted and discussed.

Response: For the DREADD experiment, we used both female and male animals and the number for each sex are described in the figure legends for both the VMH and DMH cohorts.

10. The source of the relevant drugs used in the experiments should also be provided (i.e. semaglutide).

Response: The semaglutide and CNO information was added to the methodology.

Dear Dr Sweeney,

Re: JP-RP-2024-286699R2 "Neuroanatomical dissection of the MC3R circuitry regulating energy rheostasis" by Ingrid Camila Possa-Paranhos, Jared Butts, Emma Pyszka, Christina Nelson, Dajin Cho, Samuel Congdon, and Patrick Sweeney

We are pleased to tell you that your paper has been accepted for publication in The Journal of Physiology.

Yours sincerely,

David Wyllie
Senior Editor
The Journal of Physiology

If you would like to receive our 'Research Roundup', a monthly newsletter highlighting the cutting-edge research published in The Physiological Society's family of journals (The Journal of Physiology, Experimental Physiology, Physiological Reports, The Journal of Nutritional Physiology and The Journal of Precision Medicine: Health and Disease), please click this link, fill in your name and email address and select 'Research Roundup':

<https://www.physoc.org/journals-and-media/membernews>

- You can help your research get the attention it deserves! Check out Wiley's free Promotion Guide for best-practice recommendations for promoting your work at: www.wileyauthors.com/eoo/guide. You can learn more about Wiley Editing Services which offers professional video, design, and writing services to create shareable video abstracts, infographics, conference posters, lay summaries, and research news stories for your research at: www.wileyauthors.com/eoo/promotion.

The Corresponding Author will receive an email from Wiley with details on how to register or log-in to Wiley Authors Services where you will be able to place an order

Journal of Physiology staff comments:

Reviewer 1 suggests that the statistical information in the figures could be moved to a supplementary table. However this would be against journal policies, so I wouldn't worry too much about it. For more information please see:

https://jp.msubmit.net/cgi-bin/main.plex?form_type=display_requirements#suppinfo

Reviewing Editor's comments:

Many thanks for revising the manuscript and addressing reviewer's comments.

Senior Editor's comments:

Thank you for the further revisions. I am happy to accept this work and thank you for all your efforts in responding to comments by referees. The office may be in contact as the manuscript is not formatted in JPhysiol style to ease of generating a proof.

Referee #1:

The authors have sufficiently addressed all of my comments.

One potential suggestion is to put all of the statistics in a separate table (even as supplementary info) instead of being in each figure legend, which makes the legends pretty unwieldy, but I will leave that up to the authors and editors.

END OF COMMENTS